# Chromosomal instability promotes cell migration and invasion via EFEMP1 secretion into extracellular vesicles

Siqi Zheng[1], Ruifang Tian [1], Marsudi Siburian[1], Anna Haider Rubio[1], Yuanyuan Liu[1], Rene Wardenaar [1], Marjan Shirzai [1], Laura Kempe [1], Emma Dijkstra [1,2], Eliza Warszawik[3], Maria Suarez Peredo Rodriguez[1], Klaas Sjollema[2], Petra L Bakker[1], Patrick van Rijn[3], Michaela Borghesan [1], Judith TML Paridaen[1], Stefano Santaguida [4,5] & Floris Foijer [1✉]

## Abstract

**Triple-negative breast cancer (TNBC) is characterised by high rates of chromosomal instability (CIN) and rewired intercellular communication driven by both soluble factors and extracellular vesicles (EVs). To assess how CIN might affect EV-mediated signalling in TNBC, we studied the EV landscape of TNBC cell lines with induced CIN. We find that CIN leads to increased secretion of EVs and that these EVs promote cell migration of recipient cells. EVs are enriched for extracellular matrix (ECM) proteins, including EFEMP1. Indeed, modulation of EFEMP1 levels in EVs significantly alters migration behaviour of EV-treated cells. We show that EFEMP1 expression is regulated by STAT1, that EVs from STAT1-deficient cells no longer promote migration, and that this can be rescued by overexpression of EFEMP1 in STAT1-null cells. Xenografting TNBC cells with EFEMP1-enriched cells promotes migration in zebrafish embryos, suggesting that EFEMP1 expression is a factor that promotes metastasis. Together, our results identify a CIN-associated EV program in triple-negative breast cancer and highlight EFEMP1 as a potential therapeutic target to impair EV-driven tumour cell migration.**

Subject Categories Cancer; Cell Adhesion, Polarity & Cytoskeleton; DNA Replication, Recombination & Repair

## Introduction

Triple-negative breast cancer (TNBC) is defined by the loss of the estrogen receptor (ER), progesterone receptor, and human epidermal growth factor 2 (HER2) and is associated with poor prognosis (Dietze et al, 2015). TNBC frequently displays chromosomal instability (CIN), an increased frequency of chromosome segregation errors during mitosis (Bianchini et al, 2016; Vargas-Rondón et al, 2017, 2020). CIN in TNBC is associated with cancer cell evolution, immune evasion, an altered tumour microenvironment (TME), increased metastasis and thus, a poor outcome (Hoevenaar et al, 2020; Gao et al, 2016; Bakhoum and Cantley, 2018; Li et al, 2021).

Extracellular vesicles (EVs), such as exosomes and ectosomes, secreted by cancer cells and other cells in the TME, are known to play an important role in the communication between cancer cells and the TME, thus shaping the TME (Bhome et al, 2022). Indeed, EVs, can serve as biomarkers for cancer and genomic instability, e.g. when measured in peripheral blood (Minciacchi et al, 2015; Willms et al, 2018; Bao et al, 2021; Martins et al, 2023). The molecular cargo of EVs has been extensively characterized, revealing roles for various lncRNAs, microRNAs and proteins in tumour progression (Minciacchi et al, 2015; Willms et al, 2018; Martins et al, 2023; Bao et al, 2021). CIN is increasingly associated with altered EV biogenesis and cargo, e.g. by promoting the release of EVs that remodel the TME and promote invasion and metastasis (Zheng et al, 2023; Fordjour et al, 2022; Adams et al, 2021; Bao et al, 2021; Martins et al, 2023). While this has positioned EVs as potential mediators of CIN-driven tumour progression and metastasis, the molecular mechanisms of how CIN shapes EV composition and function, and how these CIN-induced EVs act on specific cellular compartments within the TME, remain largely unresolved.

In this study, we investigate how a CIN phenotype in cells influences the secretion and content of EVs and the impact of these EVs on their neighbouring cells. For this, we compare the composition of EVs secreted by CIN[HIGH] and CIN[LOW] TNBC cells and the effect of their secreted EVs on CIN[LOW] TNBC cells. We find that CIN leads to increased secretion of EVs and that EVs secreted from CIN[HIGH] TNBC cells promote migration and invasion of

[1]European Research Institute for the Biology of Ageing, University Groningen, University Medical Center Groningen, NL-9713 AV Groningen, the Netherlands. [2]UMCG Microscopy Centre, University Groningen, University Medical Center Groningen, NL-9713 AV Groningen, the Netherlands. [3]Department of Biomaterials & Biomedical Technology, University Groningen, University of Medical Center Groningen, NL-9713 AV Groningen, the Netherlands. [4]Department of Experimental Oncology at IEO, European Institute of Oncology IRCCS, Via Adamello 16, 20139 Milan, Italy. [5]Department of Oncology and Hemato-Oncology, University of Milan, Milan, Italy. ✉E-mail: f.foijer@umcg.nl

isogenic CIN^LOW TNBC cells. We identify the extracellular matrix protein EFEMP1, also known as Fibulin-3 and a known prognostic factor in TNBC (McHenry and Prosperi, 2023; Noonan et al, 2018; Hu et al, 2011) to be enriched in EVs secreted by CIN^HIGH TNBC cells. The induction of EFEMP1 expression is linked to induced CIN and relies on the presence of Signal Transducer and Activator of Transcription 1 (STAT1), a crucial factor in breast cancer (Banik et al, 2021). Overexpression of EFEMP1 rescues the delayed migration observed in TNBC cells lacking STAT1 and restores the invasive potential of EVs originating from STAT1-expressing cells, thereby promoting the invasive phenotype. In conclusion, we have uncovered a CIN-driven pathway involving EFEMP1 and STAT1 that promotes cell migration within the TME through a paracrine mechanism, potentially contributing to the increased metastatic potential of CIN^HIGH TNBC.

# Results

## Chromosomal instability promotes extracellular vesicle production and release in TNBC cancer cells

To compare the effects between EVs released by cells with low and high rates of CIN, we use two frequently used and well-characterized TNBC cell lines, BT549 and MDA-MB-231. To induce CIN^HIGH phenotypes, we use the MPS1 inhibitor reversine (REV) (Santaguida et al, 2010), a widely used compound for this purpose (Bosco et al, 2018; Hiruma et al, 2016; Garribba et al, 2023; Ippolito et al, 2021). CIN phenotypes were quantified by time-lapse imaging according to established protocols (Crozier et al, 2022; Huis In't Veld et al, 2019; Thu et al, 2018) and confirmed that MPS1 inhibition increased the rate of mitotic abnormalities in a dose-dependent manner in both BT549 and MDA-MB-231 TNBC cells (Fig. 1A). Cells were treated with 500 nM of REV for 72 h prior to EV isolation or harvesting of target cells, unless indicated otherwise.

To determine whether CIN affects EV production and secretion, we first quantified CD63, an established marker of EVs, by immunofluorescence in BT549 and MDA-MB-231 cells (Fig. 1B,C) and found that CD63 levels were significantly increased in both BT549 and MDA-MB-231 cells upon the induction of a CIN phenotype (Fig. 1D,E). We then isolated EVs secreted into the medium by REV-treated and DMSO-treated BT549 cells by ultracentrifugation and assessed EV shape by transmission electron microscopy, which confirmed their integrity (Théry et al, 2018) (Fig. EV1A). Western blot analysis indicated that our isolates were enriched for the EV markers CD63 and CD81 (Fig. 1F). Calnexin (endoplasmic reticulum) and beta-actin were used as negative controls (Théry et al, 2018).

To exclude that our EV preparations contained co-isolated soluble or aggregated proteins, we subjected ultracentrifugation-isolated EVs to further concentration using size-exclusion chromatography (SEC) with qEV columns (Tkach et al, 2022). Ponceau S staining of both ultracentrifuge inputs and SEC outputs showed minimal protein contamination across fractions and controls (Fig. EV1B), indicating that ultracentrifugation was sufficient to separate vesicular from soluble components. Furthermore, Western blotting for canonical EV markers CD63 and CD81 (Tkach et al, 2022) across all SEC fractions showed that EVs were concentrated

in fractions 1–5 (Fig. 1G), providing an additional layer of EV concentration when required.

To further characterize EV numbers and size, we used nanoparticle tracking (Théry et al, 2018), which revealed that induction of CIN by REV significantly increased EV release, both for BT549 cells as well as MDA-MB-231 cells, while EV size remained unaltered with EV sizes ranging from 40-200 nm (Figs. 1H and EV1C,D). Taken together, these data indicate that MPS1 inhibition enhances EV secretion in both BT549 and MDA-MB-231 cells.

## CIN^HIGH TNBC cell-derived EVs promote invasion and migration in a paracrine manner

Next, to compare uptake of different EVs into recipient cells, EVs released under CIN^LOW (DMSO) and CIN^HIGH (REV-treated) conditions were isolated, labelled with PKH26, and incubated with BT549 recipient cells. As negative controls, BT549 cells were incubated with PBS or EVs at 4 °C (Fig. EV1E, upper 2 panels). Indeed, in line with increased EV secretion, BT549 cells treated with CIN^HIGH EVs showed a significant increase in cellular PKH26 fluorescence (Fig EV1E,F). As a complementary approach, we fluorescently labelled EVs of BT549 cells using an EV-specific pHluorin_M153R reporter (Sung et al, 2020) and then induced CIN. We then isolated the labelled EVs and confirmed uptake of the EVs by BT549 recipient cells by time-lapse imaging (Movie EV1). To determine functional consequences of these EVs, we then exposed BT549 cells for 24 h to EVs isolated from either CIN^LOW or CIN^HIGH cells and quantified EdU incorporation as a readout of cell proliferation. We observed no differences in EdU incorporation between cells treated with CIN^LOW or CIN^HIGH EVs, nor cells that received no EVs, indicating that BT549-isolated EVs and MDA-MB-231-isolated EVs do not influence proliferation of recipient cells (Figs. 2A,B and EV2A,B). As tumour-derived EVs have previously been associated with increased invasiveness and metastasis (Sun et al, 2021; Adams et al, 2021), we then determined the impact of EVs isolated from CIN^LOW and CIN^HIGH BT549 cells and MDA-MB-231 cells on migration and invasion of recipient BT549 and MDA-MB-231 cells, respectively using a trans-well assay in combination with uncoated or Matrigel-coated surfaces (Fig. 2C) (Vasudevan et al, 2020). We found a significant increase in migration and invasion when recipient BT549 (Fig. 2D,E) and MDA-MB-231 (Fig. 2F,G) cells were treated with CIN^HIGH EVs compared to treatment with CIN^LOW EVs. In contrast, the cells from which the EVs were isolated, showed impaired EdU incorporation (Fig. EV2C–F) and reduced migration and invasion compared to vehicle-treated cells (Fig. EV2G,H), suggesting that ongoing CIN impairs proliferation and migration of TNBC cells, but promotes the secretion of migration-stimulating EVs.

We next asked whether the pro-migratory effects of CIN^HIGH EV isolates were truly driven by EVs. For this, we disrupted endolysosomal/exocytic trafficking in EV-recipient cells by transiently silencing expression of RAB27A (Fig. EV2I,J), a standard strategy to attenuate EV pathway activity and to test EV dependence of phenotypes in cancer models (Ostrowski et al, 2010; Wang et al, 2025; Jamshidiha et al, 2022). RAB27A siRNA but not control siRNA treatment of BT549 recipient cells completely abrogated the increase in migration induced by EVs isolated from CIN^HIGH cells (Fig. EV2K,L), indicating that RAB27A-dependent

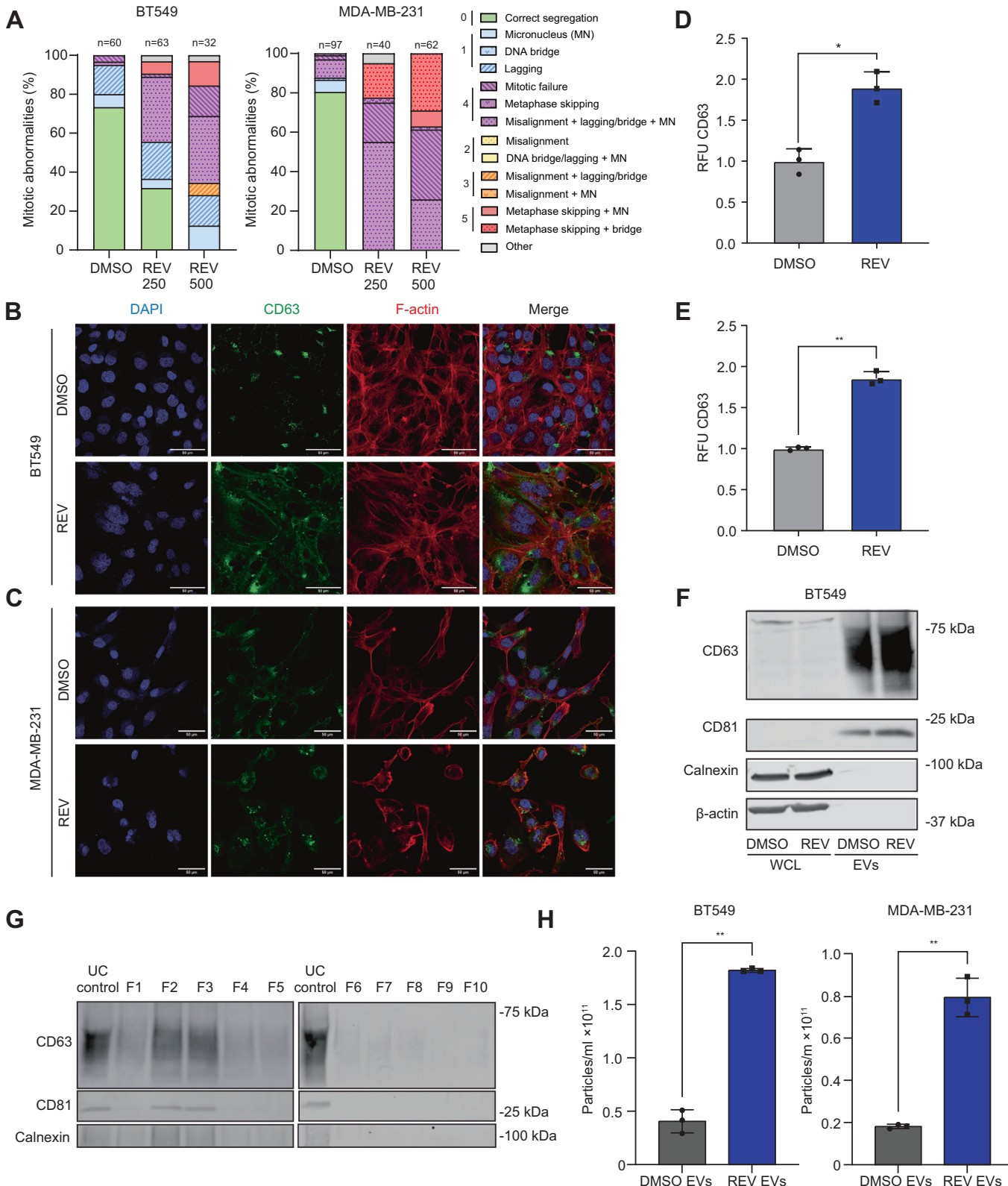

◀ **Figure 1. Chromosomal instability promotes extracellular vesicle production and release in TNBC cells.**

(A) Quantitative analysis of mitotic aberrations in BT549 and MDA-MB-231 cells after exposure to 250 nM and 500 nM reversine (REV). (B, C) CD63 and F-Actin immunofluorescence labelling in BT549 (B) and MDA-MB-231 (C) cells treated with 500 nM reversine for 72 h, compared to DMSO controls. DAPI was used to label nuclei. (D, E) Quantification of CD63 immunofluorescence intensity in BT549 (D) and MDA-MB-231 (E) cells in presence and absence of CIN phenotypes. Error bars represent the standard deviation (SD) of the mean. $N = 3$ independent experiments; each data point represents the mean of the replicates; Statistical significance was determined with a paired two-tailed t-test. *$p < 0.05$; **$p < 0.01$ (D) $p = 0.0451$; (E) $p = 0.0034$. RFU is relative fluorescence units, REV refers to 500 nM reversine unless indicated otherwise. (F) Western blot for EV markers for CD63 and CD81.Calnexin and beta-Actin serve as a loading control for whole lysates and negative control for EVs. (G) Western blots of SEC fractions show EV markers (CD63, CD81) enriched in fractions 1–5 and absent in fractions 6–10. (H) Concentrations of EVs isolated from BT548 (left panel) and MDA-MB-231 (right panel) cells as determined by nanoparticle tracking analysis. $p = 0.0019$ (left), and $p = 0.0087$ (right). Error bars represent the standard deviation (SD) of the mean. $N = 3$ independent experiments; Statistical significance was determined with a paired two-tailed t-test. Source data are available online for this figure.

vesicle trafficking is required for the migration phenotype of EV-recipient cells, further substantiating that the migration effects are driven by EVs.

To assess whether the EV-driven migration phenotype can be generalized to other cancer cell lines or non-cancer cells, we treated near-diploid HCT116 colon cancer cells, non-transformed near-diploid RPE1 and non-transformed diploid BJ cells with EVs isolated from DMSO and REV-treated BT549 cells and vice versa. Indeed, HCT116 cells treated with EVs isolated from REV-treated BT549 cells and BT549 cells treated with EVs isolated from REV-treated HCT116 cells displayed increased migration in scratch assays indicating that the migration phenotype also occurs in cell lines of another cancer type (Fig. EV3A). However, this was not true for non-transformed cell lines as recipient RPE1 cells treated with EVs isolated from CIN^HIGH BT549 cells showed modestly reduced migration (Fig. EV3B, left bars) while recipient BJ cells showed no effects on migration when treated with CIN^HIGH BT549 EVs (Fig. EV3C, left bars). Similarly, EVs isolated from REV-treated RPE1 cells, or REV-treated BJ-cells did not affect migration behaviour of recipient BT549 cells (Fig. EV3B,C, right bars).

To identify factors that mediate the effect on migration observed in TNBC cells, we assessed protein contents of CIN^HIGH and CIN^LOW BT549 EVs using label-free mass spectrometry (Dataset EV1, Fig. 2H, the latter showing a selection of all identified peptides). Overlaying our mass spec results with proteins annotated in the EV Exocarta database (Keerthikumar et al, 2016; Mathivanan et al, 2012), revealed substantial overlap, confirming our isolations were highly enriched for EVs (Fig. EV3D). A general gene ontology (GO) analysis validated that the content of our EVs was enriched for 'exosomes' (Fig. EV3E). Furthermore, 'Integrin binding' and 'Cell adhesion', were among the topmost significantly enriched categories for CIN^HIGH EVs in GO and KEGG (Kyoto Encyclopedia of Genes and Genomes) analyses (Luo and Brouwer, 2013; Kanehisa et al, 2021; Ge et al, 2020) (Fig. EV3F). Given its known association with migration and invasion phenotypes (Noonan et al, 2018), we next decided to further investigate Fibulin3 (FBLN3, Fig. 2H). Fibulin3 is encoded by the EFEMP1 gene, which is the name we will use in the rest of this study. As a first validation, we quantified EFEMP1 levels in CIN^LOW and CIN^HIGH BT549 cells and EVs isolated from these cells, which revealed that EFEMP1 levels were increased in CIN^HIGH BT549s and EVs secreted by these cells, compared to their DMSO-treated counterparts (Fig. EV3G).

We conclude that EVs isolated from CIN^HIGH TNBC and colon cancer cells, but not those isolated from non-cancer cells, can promote migration and invasion in a paracrine and RAB27-dependent manner and that EFEMP1, an EV-enriched factor, is a candidate driver of this phenotype.

## EFEMP1-enriched EVs promote migration and invasion in recipient TNBC cells

To investigate the role of increased EFEMP1 levels in CIN^HIGH EVs in migration and invasion observed in TNBC cells upon EV treatment, we manipulated EFEMP1 expression in BT549 (Fig. EV4A,B) and MDA-MB-231 (Fig. EV4C,D) TNBC cells using overexpression and shRNA constructs. We then determined the impact of EFEMP1 modulation on the migratory and invasive potential of these cells directly. Trans-well assays revealed that overexpression of EFEMP1 promoted migration and invasion of both BT549 as well as MDA-MB-231 cells (Fig. EV4E–H). Similarly, scratch assays (Jonkman et al, 2014) showed that EFEMP1 overexpression promoted invasiveness of BT549 cells (Fig. EV4I). Conversely, downregulation of EFEMP1 impaired migration and invasion of both cell types in migration and scratch assays (Fig. EV4J–N). After having evaluated the effects of EFEMP1 expression levels on TNBC cells directly, we next asked whether EFEMP1 contributes to the EV-mediated effects we observed on recipient TNBC cells. To this end, we isolated EVs of EFEMP1-overexpressing BT549 and MDA-MB-231 cells (EFEMP1^OEX EVs) and treated their respective parental cells with these. As expected, overexpression of EFEMP1 led to increased EFEMP1 protein levels in both BT549 and MDA-MB-231 as well as in EVs isolated from these (Fig. EV5A,B). Indeed, we found that BT549 derived EFEMP1^OEX EVs promoted migration and invasion of recipient BT549 cells (Fig. 3A,B). This was also true for MDA-MB-231 cells treated with MDA-MB-231 EFEMP1^OEX EVs (Fig. EV5C). Conversely, treating parental BT549 and MDA-MB-231 cells with EVs isolated from EFEMP1^KD BT549 (Fig. EV5D) and EFEMP1^KD MDA-MB-231 cells (Fig. EV5E), respectively, reduced migration and invasion potential (Figs. 3C,D and EV5F), jointly indicating that EFEMP1 in EVs modulates migration and invasion of recipient cells in a dose-dependent manner.

To explore the contribution of EFEMP1 to the migration and invasion phenotypes induced by CIN^HIGH EVs, we induced CIN in control and EFEMP1^KD cells (Fig. EV5G), isolated EVs, and determined their effects on recipient cell migration and invasion. While CIN^HIGH control EVs promoted migration and invasion as observed previously, this effect was largely abolished in CIN^HIGH EFEMP1^KD EVs (Fig. 3E,F), indicating that EFEMP1 plays a key role in the CIN^HIGH EV-mediated migration and invasion phenotypes observed in BT549 cells.

To better understand the mechanism underlying the invasion and migration phenotype instigated by EFEMP1 EVs, we performed RNA sequencing analysis on BT549 cells treated with control and

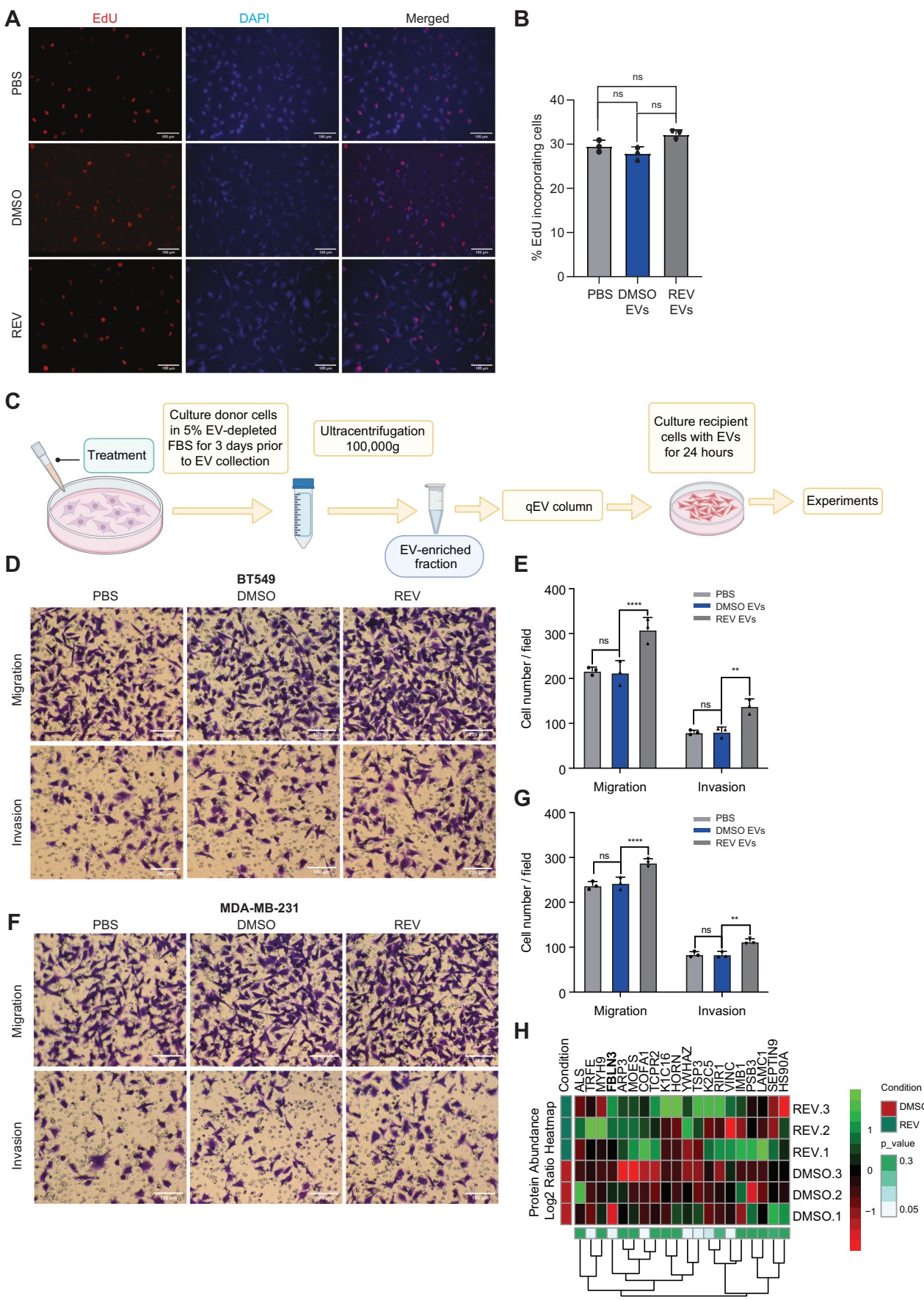

Figure 2.   Paracrine effects of EVs isolated from CIN^HIGH BT549 and MDA-MB-231 cells on cell proliferation, migration, and invasion.

(A) Representative images of EdU incorporation in BT549 cells treated with EVs from DMSO or REV treated cells. Scale bar: 100 µm. (B) Quantification of EdU incorporation as shown in (A) for three biological replicates. Statistical analysis was performed using Two-way ANOVAS ($*p < 0.05$; $**p < 0.01$; $***p < 0.001$; $****p < 0.0001$). (C) Schematic outline of the treatment of BT549 cells with EVs and experimental setup for downstream analyses. Image created using a licenced BioRender account. (D, E) Representative images (D) and quantification (E) of cell migration and invasion of BT549 cells treated with PBS, DMSO (CIN^LOW) or REV (CIN^HIGH) using trans-well assays. Scale bar: 100 µm. Error bars represent the standard deviation (SD) of the mean. $N = 3$ independent experiments; each data point represents the mean of the replicates; Statistical significance was determined with a two-way ANOVA. $p = 0.000089$, $****p < 0.0001$, and $p = 0.0060$, $***p < 0.001$. Experiments were performed as biological triplicates. (F, G) Representative images (F) and quantification (G) of cell migration and invasion of MDA-MB-231 cells treated with PBS, DMSO (CIN^LOW) or REV (CIN^HIGH) using trans-well assays. Scale bar: 100 µm. Error bars represent the standard deviation (SD) of the mean. $N = 3$ independent experiments; each data point represents the mean of within-experiment replicates; Statistical significance was determined with a two-way ANOVA. $p = 0.000126$, $***p < 0.0001$, and $p = 0.0053$, $**p < 0.001$. Experiments were performed as biological triplicates. (H) Heatmap of a selection of peptides identified in the EVs isolates from BT549 cells treated with REV compared to controls. The full list can be found in Dataset EV1. Source data are available online for this figure.

EFEMP1^KD EVs. Among the 290 genes showing differential expression between cells treated with control EVs and EFEMP1^KD EVs, 29 have a known role in migration and/or invasion (Fig. EV6A) (Shi et al, 2023). Notably, the dysregulated genes are enriched in pathways such as 'ECM receptor interaction' and 'Focal adhesion' (KEGG pathways, Fig. EV6B) (Luo and Brouwer, 2013; Kanehisa et al, 2021; Ge et al, 2020). Specifically, genes involved in 'cell adhesion' and 'cadherin binding' (GO pathways, Fig. EV6C) exhibit reduced expression levels in cells treated with EFEMP1^KD EVs compared to those treated with control EVs. To further validate transcriptional modulation of genes involved in cell adhesion by EFEMP1 in EVs, we quantified transcript levels of several genes that have previously been linked to EMT and adhesion (Hałas-Wiśniewska et al, 2025; Bendas and Borsig, 2012; Nieto et al, 2016; Bakir et al, 2020; Ostrowska-Podhorodecka et al, 2021). While TWIST1, CDH2 (N-cadherin) and VIM were upregulated, CDH1 (E-cadherin) was downregulated in BT549 cells treated with EVs isolated from EFEM1-overexpressing cells (Fig. EV6D). This pattern is in line with a canonical 'cadherin switch' associated with vimentin-dependent cytoskeletal reorganization, integrin activation, and increased focal-adhesion turnover, facilitating cell motility (Hałas-Wiśniewska et al, 2025; Ostrowska-Podhorodecka et al, 2021). The converse was true for BT549 cells treated with EVs isolated from EFEMP1^KD cells (Fig. EV6E). In line with this, BT549 cells treated with EFEMP1-enriched EVs displayed enhanced adhesion across several adhesion substrates (Fig. EV6F,G), whereas cells treated with EFEMP1-depleted EVs showed decreased adhesion (Fig. EV6F,H). Together, these data support a model in which CIN-induced EFEMP1-containing EVs promote epithelial to mesenchymal transition, promoting integrin and cadherin-mediated adhesion and thus increased migration (Khalili and Ahmad, 2015; Ahmad et al, 2015; Liu et al, 2019).

## STAT1 is a regulator of EFEMP1 expression in CIN^HIGH cells

We next asked how EFEMP1 expression is modulated following acute CIN. As we recently identified STAT1 as a central node in CIN-induced inflammatory signalling (preprint: Schubert et al, 2021; Hong et al, 2022), we first determined whether EFEMP1 is a STAT1 target gene using publicly available ChIP-seq data (Inoue et al, 2017; Kazachenka et al, 2018; Satoh and Tabunoki, 2013). Indeed, STAT1 binds to multiple sites across the human EFEMP1 locus (Fig. EV7A) (Satoh and Tabunoki, 2013). This is also true for B2M (Fig. EV7B), another established STAT1-target gene

(Neerincx et al, 2013; Satoh and Tabunoki, 2013) but not for ACTB (beta-Actin) (Fig. EV7C) (Robertson et al, 2016; Satoh and Tabunoki, 2013). This strongly supports that STAT1 drives EFEMP1 expression. Furthermore, plotting EFEMP1 and STAT1 expression levels for more than 1000 cancer cell lines confirmed a positive correlation between the expression of both genes across all DepMap included cancer cell lines (Fig. 4A), as well as all DepMap included breast cancer cell lines (Fig. 4B; dataset 23Q4; (Tsherniak et al, 2017)). Together, these data indicate that EFEMP1 is a bona fide STAT1 target.

To further test a possible epistatic relation between STAT1 and EFEMP1 in a CIN background, we generated STAT1^KO BT549 cells (Fig. EV7D) and compared expression levels of several inflammatory genes (IL-6, IL-8, CXCL1, and CXCL10), and furthermore expression levels of EFEMP1, STAT1, STAT3, CD63 and CYLD in STAT1-proficient and -deficient backgrounds, with and without CIN. While expression of the cytokines was significantly upregulated in a STAT1-independent manner, EFEMP1, STAT3, CD63 and CYLD were not upregulated in STAT1-deficient BT549 cells, indicating that the expression of the latter is STAT1-regulated (Fig. 4C). In fact, EFEMP1, STAT3 and CD63 levels were even downregulated under CIN conditions. Also, note that STAT1-deficient cells show much lower basal expression of EFEMP1, further evidence that EFEMP1 is a bona fide STAT1 target gene (Fig. EV7D).

We then isolated EVs of STAT1-proficient and -deficient BT549 cells that were either treated with DMSO (CIN^LOW) or MPS1 inhibitor (CIN^HIGH), transferred these to recipient BT549 cells and quantified migration and invasion using trans-well assays. While CIN^HIGH EVs isolated from STAT1-proficient cells induced migration and invasion as expected, loss of STAT1 alleviated this phenotype, suggesting that STAT1-mediated EFEMP1 expression is required for this (Fig. 4D,E). To determine whether the decreased potential to induce migration and invasion of STAT1-deficient EVs was indeed dependent on EFEMP1, we overexpressed EFEMP1 in STAT1^KO BT549 cells (Fig. EV7E), isolated EVs and compared migration and invasion between STAT1^KO cells with and without EFEMP1 overexpression. We found that re-expressing EFEMP1 in STAT1^KO BT549 was indeed sufficient to rescue the migration and invasion potential of EVs isolated from these cells (Fig. 4F,G), independent of the EV isolation method (Fig. EV7F–H). Jointly, these findings indicate that EFEMP1 is induced in a STAT1-dependent manner and that overexpression of EFEMP1 in STAT1^KO BT549 cells is sufficient for the secretion of EVs that promote migration and invasion of recipient TNBC cells.

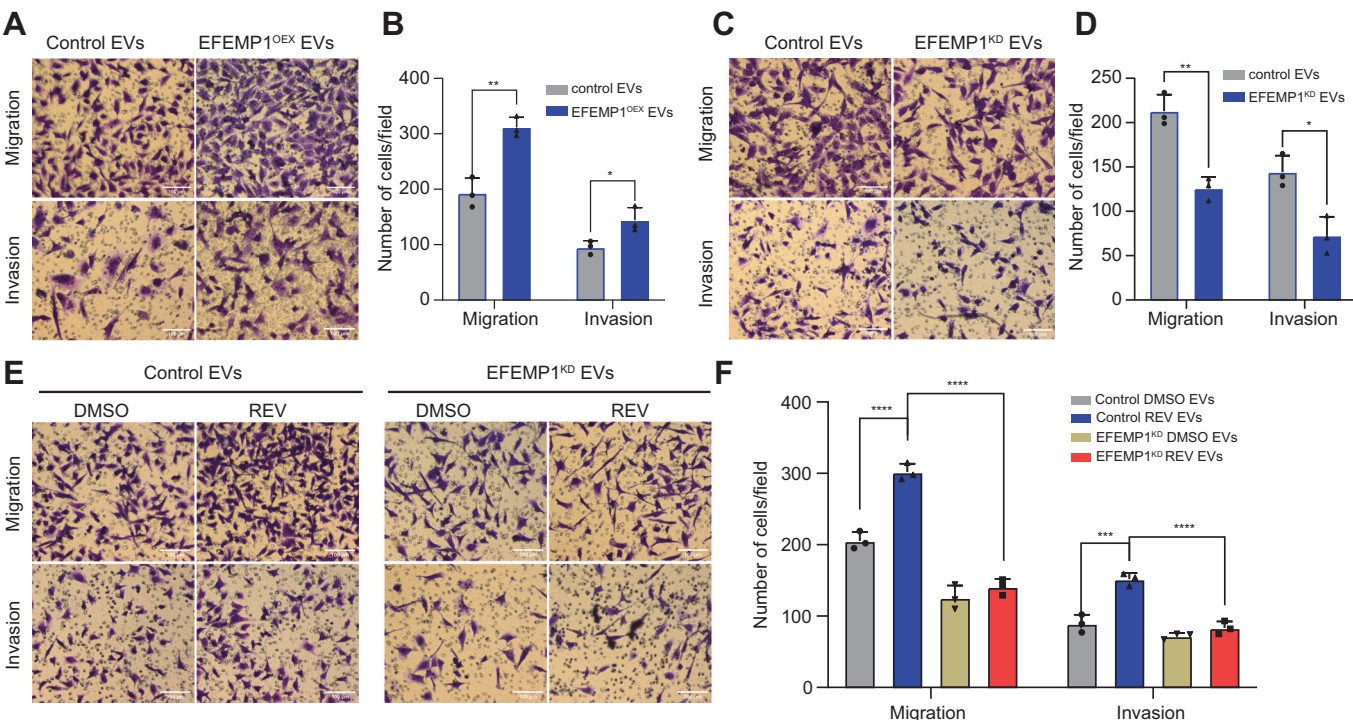

**Figure 3.  EFEMP1-enriched EVs promote cell migration and invasion.**

(A, B) Representative images (A) and quantification (B) of BT549 migration and invasion transwell experiments following EFEMP1 overexpression (EFEMP1$^{OEX}$) compared to control EVs. Statistical significance was determined by a two-sided T-test ($n = 3$ biological replicates), Error bars represent the standard deviation (SD) of the mean, $p = 0.0377$, $p = 0.0242$. *$p < 0.05$. Scale bar: 100 μm. (C, D) Representative images (C) and quantification (D) of BT549 migration and invasion transwell experiments following EFEMP1 knockdown compared to control (scramble) EVs. Statistical significance was determined by a two-sided T-test ($n = 3$ biological replicates), Error bars represent the standard deviation (SD) of the mean. $p = 0.0095$ (left), $p = 0.0118$ (right). *$p < 0.05$, **$p < 0.01$. Scale bar: 100 μm. (E, F) Representative images (E) and quantification (F) of migration and invasion of control and CIN$^{HIGH}$ (REV-treated) BT549 cells using transwell assays for scramble control and EFEMP1$^{KD}$ conditions. Statistical significance was determined by a two-way ANOVA ($n = 3$ biological replicates), Error bars represent the standard deviation (SD) of the mean, $p$-values left to right: $p < 0.0001$; $p < 0.0001$; $p = 0.0001$; and $p < 0.0001$. *$p < 0.05$, **$p < 0.01$, ***$p < 0.001$, ****$p < 0.0001$. Scale bar: 100 μm. Source data are available online for this figure.

## EFEMP1 promotes cell dissemination in vivo and is associated with decreased survival in breast cancer patients

Migration and invasion are important factors for metastasis (van Zijl et al, 2011; Pachmayr et al, 2017; Novikov et al, 2021). To test whether EFEMP1-containing EVs facilitate tumour cell spreading in vivo, we made use of a zebrafish xenograft model. For this, we cultured H2B-mCherry labelled MDA-MB-231 cells and treated them with EVs isolated from MDA-MB-231 cells in which EFEMP1 levels were modulated. Post-EV incubation, cells were injected into the perivitelline space (PVS) of 36 hpf-old zebrafish embryos (Rouhi et al, 2010; Teng et al, 2013; Martinez-Lopez et al, 2021). We then determined which fraction of the injected EV-treated MDA-MB-231 cells migrated to the tail section of the embryos (Figs. 5A and EV8A–C). We compared the spreading of cells treated with control EVs, EVs from EFEMP1$^{OEX}$ and EVs from EFEMP1$^{KD}$ cells (Fig. 5B) and found that while ~50% of embryos injected with cells treated with control EVs showed cells spreading towards the tail of the embryo (Fig. 5C), this fraction was significantly reduced for cells treated with EFEMP1$^{KD}$ EVs, and, conversely, significantly increased for cells treated with EFEMP1$^{OEX}$ EVs. Quantifying the number of spreading cells further

strengthened this observation: while ~30% of the embryos injected with cells treated with control EVs showed more than 4 cells spreading to the tail, this fraction was reduced at least six-fold in embryos injected with cells treated with EFEMP1$^{KD}$ EVs (Fig. 5D). For cells treated with EFEMP1$^{OEX}$ EVs, the proportion of embryos exhibiting the spread of more than 5 cells towards the tail doubled (Fig. 5E). Together, these data indicate that that EFEMP1-derived EVs can modulate cancer cell spreading in vivo.

Finally, to relate our findings to human cancer, we examined public datasets. Using DepMap data (23Q4; (Tsherniak et al, 2017)), we found that EFEMP1 mRNA levels positively correlated with aneuploidy scores across all cancer cell lines, including breast cancer lines (Fig. EV8D,E), consistent with our experimental observation that CIN elevates EFEMP1 expression. Furthermore, stratifying breast cancer survival for EFEMP1 expression levels revealed that high EFEMP1 expression associated with a more adverse outcome, particularly in higher-grade tumours (Fig. EV8F–I), although these effects were relatively modest. However, together these data confirm that EFEMP1 expression is associated with aneuploidy across cancers and with poor survival in breast cancer, also in a human setting.

While further work is required to explore the role of EFEMP1 as a cancer therapy target, altogether our work identifies EFEMP1 as a

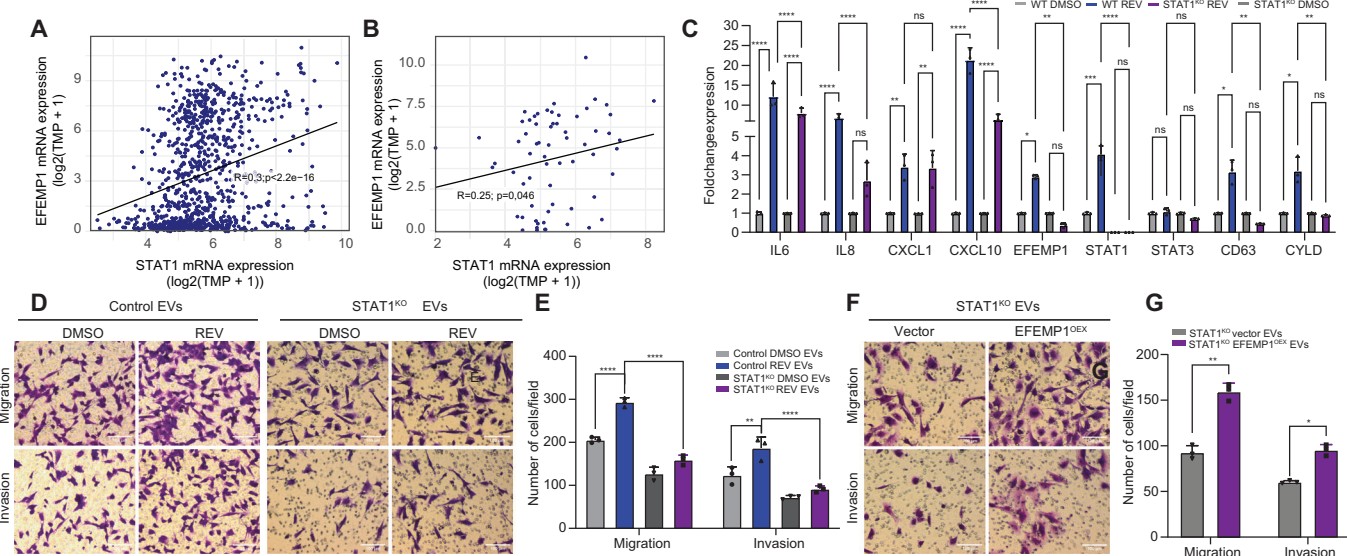

**Figure 4. STAT1 is required for CIN-induced EFEMP1 expression and its downstream effects on migration and invasion.**

(A) Correlation analysis of EFEMP1 and STAT1 mRNA expression using DepMap data. A Pearson correlation (two-tailed) was performed for this purpose; R and P are shown. (B) Correlation analysis of EFEMP1 and STAT1 mRNA expression in DepMap-included breast cancer cell lines. A Pearson correlation (two-tailed) was performed for this purpose; R and P are shown. (C) qPCR quantification of IL-6, IL-8, CXCL1, CXCL10, EFEMP1, STAT1, STAT3, CD63 and CYLD in BT549 WT and STAT1$^{KO}$ cells. Statistical analysis was done using a two-way ANOVA test ($N = 3$; *$p < 0.05$, **$p < 0.01$, ***$p < 0.001$, ****$p < 0.0001$), Error bars represent the standard deviation (SD) of the mean. WT REV vs WT DMSO: IL-6: $p < 0.0001$ (****); IL-8: $p < 0.0001$ (****); CXCL1: $p = 0.0060$ (**); CXCL10: $p < 0.0001$ (****); EFEMP1: $p = 0.1589$ (ns); STAT1: $p = 0.0003$ (***); STAT3: $p = 0.9991$ (ns); CD63: $p = 0.0102$ (*); CYLD: $p = 0.0145$ (*). STAT1$^{KO}$ REV vs STAT1$^{KO}$ DMSO: IL-6: $p < 0.0001$ (****); IL-8: $p = 0.0893$ (ns); CXCL1: $p = 0.0070$ (**); CXCL10: $p < 0.0001$ (****); EFEMP1: $p = 0.8185$ (ns); STAT1: $p > 0.9999$ (ns); STAT3: $p = 0.9760$ (ns); CD63: $p = 0.8688$ (ns); CYLD: $p = 0.9984$ (ns); WT REV vs STAT1$^{KO}$ REV: IL-6: $p < 0.0001$ (****); IL-8: $p < 0.0001$ (****); CXCL1: $p = 0.9998$ (ns); CXCL10: $p < 0.0001$ (****); EFEMP1: $p = 0.0200$ (*); STAT1: $p < 0.0001$ (****); STAT3: $p = 0.9459$ (ns); CD63: $p = 0.0009$ (***); CYLD: $p = 0.0090$ (**). (D, E) Representative images (D) and quantification (E) of cell migration and invasion of BT549 cells co-cultured with EVs isolated from WT of STAT1$^{KO}$ BT549 cells, treated with DMSO or REV, assessed by transwell assays. Statistical significance was determined by a two-way ANOVA test ($n = 3$ biological replicates), Error bars represent the standard deviation (SD) of the mean. p-values from left to right: $p < 0.0001$; $p = 0.0012$, $p < 0.0001$, **$p < 0.01$, ***$p < 0.001$, ****$p < 0.0001$. Scale bar: 100 µm. (F, G) Representative images (F) and quantification (G) of cell migration and invasion of BT549 cells co-cultured with EVs isolated from STAT1$^{KO}$ BT549 cells with or without EFEMP1 overexpression, assessed by transwell assays. Statistical significance was determined by a two-sided T-test ($n = 3$ biological replicates), Error bars represent the standard deviation (SD) of the mean, $p = 0.0012$ (left) and $p = 0.0139$ (right). *$p < 0.05$, **$p < 0.01$. Scale bar: 100 µm. Source data are available online for this figure.

CIN-induced, STAT1-regulated factor that promotes migration and invasion of cancer cells in a paracrine manner. As high EFEMP1 expression is correlated with poor survival of breast cancer patients, future work should explore whether inhibition of EFEMP1 indeed improves breast cancer survival, for instance by reducing metastatic potential.

# Discussion

In this study, we investigated the paracrine effects of chromosomal instability using TNBC, the most aggressive breast cancer subtype, as a model cancer. We find that CIN, induced by the MPS1 inhibitor reversine promotes the secretion of extracellular vesicles, thus revealing a new mechanism by which CIN$^{HIGH}$ cancer cells can influence the tumour microenvironment (TME). In line with our findings, others have reported that centrosome amplification, another driver of CIN, also leads to increased release of EVs (Adams et al, 2021). However, CIN triggers various downstream responses including cell cycle arrest, cell death, and inflammation (Zheng et al, 2023). While our data indicate that ongoing CIN leads to increased secretion of EVs and that these EVs modulate the

TME, but that this CIN also leads to decreased proliferation of the EV-secreting cells, future work should reveal whether other instigators of cell cycle arrest also yield secretion of EVs that promote migration of recipient cells.

We identify EFEMP1 as a key factor secreted by CIN$^{HIGH}$ cancer cells via EVs to modulate the migration and invasion of recipient cells. EFEMP1 has previously been identified as a factor secreted by TNBC cells (McHenry and Prosperi, 2023), and as a biomarker of cancer detected in peripheral blood (Noonan et al, 2018). We find that CIN increases EFEMP1 protein levels up to ~1.5 fold and that this is sufficient to promote the migration of cancer cells in tissue culture. While our findings show that EFEMP1 is associated with EVs, further work should clarify whether EFEMP1 resides inside or on the surface of the CIN-induced EVs. However, our findings do reveal EFEMP1 as a CIN-induced factor and a paracrine modulator of cell migration and invasion. We show that EFEMP1-loaded EVs promote the migration of cancer cells in tissue culture and a zebrafish model and that this effect is most pronounced in CIN$^{LOW}$ TNBC cells treated with EVs isolated from CIN$^{HIGH}$ TNBC cells. However, further work in for instance mouse models and human co-culture models is required to better understand which cell types in the TME are influenced by these EVs in a more human relevant

setting and how this compares to the effects that the EVs have on neighbouring cancer cells. Nevertheless, our finding that CIN-induced expression of EFEMP1 might contribute to metastasis aligns well with the discovery that CIN-driven activation of the cGAS–STING pathway promotes a pro-metastatic TME (Bakhoum and Cantley, 2018; Li et al, 2023, 2021). Our work thus complements these other findings on the role of CIN in metastasis.

In addition to better understanding how CIN phenotypes influence the TME, our work further contributes to a broader understanding of how paracrine signalling of cancer cells can promote an EMT-like adhesion remodelling program in cancer cell invasion and migration.

Transcriptome analysis of TNBC cells treated with EVs isolated from EFEMP1[KD] cells revealed a significant decrease in the expression of genes involved in adhesion, suggesting that EFEMP1 in EVs contributes to an increased adhesion phenotype in recipient cells, which was confirmed with adhesion assays of both EFEMP1[OEX] and EFEMP1[KD] cells. While an increased adhesion phenotype might appear at odds with the concomitant increased migration, this could be due to a dynamic regulation of cell-matrix interactions that enhances detachment and reattachment processes, thus offering a potential explanation for our finding that CIN promotes the migration of EV-recipient cells via EFEMP1. This is consistent with findings of others that exosomes can deposit fibronectin and other ECM proteins promoting adhesion while enabling directional migration in vivo (Sung et al, 2015). Indeed, prior work has identified a role for EFEMP1 in TNBC cell migration through its interaction with KISS1R (Noonan et al, 2018), albeit not yet in the context of CIN and EVs. Further work is required to better understand the molecular mechanism that underlies the CIN-induced EFEMP1-mediated effect on migration and invasion of cells.

We furthermore show that EFEMP1 expression is regulated by STAT1, a key factor in breast cancer progression (Chan et al, 2012), thus revealing a new regulatory pathway that regulates migration and invasion, downstream of CIN. Indeed, STAT1 was previously identified as a key signalling node downstream of CIN that is activated by acute CIN (preprint: Schubert et al, 2021; Hong et al, 2022), but also inactivated in CIN[HIGH] cancers, presumably to prevent immune recognition of CIN[HIGH] cancer cells. As CIN is known to promote metastasis but also leads to downregulation of STAT1 signalling in cancer, it will be interesting to test whether EFEMP1 is upregulated in a STAT1-independent manner in CIN[HIGH] cancers and thus contributes to the migration and invasion of cancer cells, and, via EVs, also to migration and invasion of other cell types in the TME.

Our findings are supported by analyses of publicly available real-life datasets. Using DepMap data, we found that EFEMP1 expression positively correlates with aneuploidy in cancer cell lines. Furthermore, in line with our finding that CIN promotes EFEMP1 expression and that EFEMP1 expression promotes invasion and migration, analyses of TGCA data show that increased expression of EFEMP1 is associated with poor breast cancer patient survival.

Altogether, our work reveals EFEMP1 as a CIN-regulated factor that is secreted via EVs and that promotes cancer cell migration in a paracrine manner. Therefore, EFEMP1 might present a new clinical target of CIN[HIGH] cancers to suppress their increased metastasis rates.

# Methods

### Reagents and tools table

| Reagent/Resource | Reference or Source | Identifier or Catalog Number |
|---|---|---|
| | TCGA Breast Invasive Carcinoma data were accessed via cBioPortal (cBioPortal, n.d.; Cerami et al, 2012) and queried for EFEMP1 and STAT1 to integrate clinical annotations with gene expression. METABRIC expression and clinical data were included as an independent cohort (cBioPortal, 2012). For cell-line validation, DepMap was used to relate EFEMP1/STAT1 expression to aneuploidy scores. | |
| **Experimental models** | | |
| BT549(human) | ATCC | |
| MDA-MB-231(human) | ATCC | |
| 293 T(human) | ATCC | |
| **Recombinant DNA** | | |
| pSpCas9(BB)-2A-Puro V2.0 (PX459) | Addgene | 62988 |
| pSpCas9(BB)-2A-Puro V2.0 (PX459)-STAT1 | This study | |
| Tet-pLKO-puro | Addgene | 21915 |
| FL-fibulin-3 pcDNA4 | Addgene | 29700 |
| pLenti-pHluorin_M153R-CD63-mScarlet | Addgene | 172118 |
| pcDNA | Addgene | 138209 |
| pSPAX2 | Addgene | 12260 |
| pMD2.G | Addgene | 12259 |
| Tet-pLKO-puro-EFEMP1 shRNA1 | This study | |
| Tet-pLKO-puro-EFEMP1 shRNA2 | This study | |
| PIGZ-H2B-Cherry | This study | |
| **Antibodies** | | |
| β-Actin (13E5) Rabbit mAb | Cell Signaling Technology | Cat# 4970, Clone 13E5, RRID:AB_2223172 |
| β-Actin (8H10D10) Mouse mAb | Cell Signaling Technology | Cat# 3700, Clone 8H10D10, RRID:AB_2242334. |
| EFEMP1 (Fibulin-3/ EFEMP1) | Novus Biologicals | Cat# NBP1-77040, polyclonal |
| STAT1 (Stat1) antibody | Cell Signaling Technology | Cat# 9172 (9172S), polyclonal, RRID:AB_2198300 |

| Reagent/Resource | Reference or Source | Identifier or Catalog Number |
|---|---|---|
| CD63 (H5C6) Mouse mAb | Novus Biologicals | Cat# NBP2-42225, Clone H5C6, RRID:AB_2884028 |
| CD81 (1D6) Mouse mAb | Novus Biologicals | Cat# NB100-65805 (: NB100-65805SS is the same catalog line), Clone 1D6, RRID:AB_962702 |
| Vinculin (hVIN-1) Mouse mAb | Sigma-Aldrich | Cat# V9131, Clone hVIN-1, RRID:AB_477629 |
| Calnexin antibody | Novus Biologicals | Cat# NB100-1965 (NB100-1965SS is the same catalog line), RRID:AB_10002123. |
| **Oligonucleotides and other sequence-based reagents** | | |
| qPCR primers | This study | Table EV2 |
| shRNA primers | This study | Table EV2 |
| RAB27A siRNA1 | Dharmacon D-004667-01-0020 | (5'-GGACAAAGUCUGCAAGUUA) |
| RAB27A siRNA2 | Dharmacon D-004667-04-0020 | (5'-GCAACAGCCUCAAGAAUUA) |
| siGENOME non-targeting siRNA Control Pools #2 | Dharmacon D-001210-02-20 | |
| **Chemicals, Enzymes and other reagents** | | |
| RPMI 1640 | Gibco | 11554526 |
| DMEM | Gibco | 31966-021 |
| Fetal bovine serum, FBS | Thermo Fisher Scientific | 11573397 |
| Penicillin/ Streptomycin | Gibco | 15140-122 |
| TrypLE Express | Gibco | 12605-010 |
| PBS | Gibco | 12559069 |
| Formaldehyde 4% | Sigma-Aldrich | 252549 |
| Triton X-100 | Sigma-Aldrich | 9036-19-5 |
| Phalloidin-iFluor 555 | Abcam | ab176756 |
| DAPI | Sigma | D9542-1MG |
| Vivaspin® 20 centrifugal concentrator, 100 kDa MWCO PES | Cytiva | 28-9323-63 |
| qEV single-use EV SEC columns | Izon Science | IC2-35 |
| Vivaspin® 2, 10 kDa MWCO | Cytiva | 28-9322-47 |
| PKH26 Red Fluorescent Cell Linker Kit | Sigma | MINI26-1KT |
| EdU | Lumiprobe | 10540 |
| Click chemistry detection reagent | Lumiprobe | D1330 |
| cOmplete, EDTA-free | Roche | 11873580001 |
| Pierce™ BCA Protein Assay Kit | Thermo Scientific | 23225 |

| Reagent/Resource | Reference or Source | Identifier or Catalog Number |
|---|---|---|
| SuperSignal™ West Femto Maximum Sensitivity Substrate | Thermo Fisher Scientific | 10095983 |
| Trans-well chambers/Millicell inserts | cellQART | 9328002 |
| ECM gel/Matrigel | Corning | 356234 |
| Doxycycline | Sigma-Aldrich | D9891 |
| Proteinase K | Roche | Lot 30032017 |
| BbsI | NEB | R3539S |
| AgeI | NEB | R3552 |
| EcoRI | NEB | Cat# R0101 |
| Polybrene | Sigma-Aldrich | Cat# 107689 |
| 0.45 µm filter | TPP | Y99745 |
| RNeasy Plus kit | Qiage | 74136 |
| LunaScript RT SuperMix | NEB | M3010X |
| iTaq Universal SYBR Green Supermix | Bio-Rad | 1725124 |
| NEBNext Multiplex Oligos for Illumina | NEB | E6440L |
| Human fibronectin | CellSystems | 5050 |
| Mouse laminin | Thermo Fisher | 23017015 |
| Poly-L-lysine, high MW | Sigma-Aldrich | A3890401 |
| Hyaluronic acid, high MW | Calbiochem | 23017015 |
| Rat tail collagen coating solution | Merck Life Science N.V | 122-20 |
| T7 DNA ligase | NEB | M0318 |
| T4 PNK | NEB | M0201 |
| Lipofectamine™ RNAiMAX Transfection Reagent | Thermo Fisher | 13778075 |
| **Software** | | |
| ImageJ v1.53 | ImageJ/NIH (ImageJ.net) | |
| Fiji ImageJ v1.53c | Fiji/ImageJ (ImageJ.net) | |
| ZetaView software v8.04.02 | Particle Metrix | |
| ICY version Xpro | Institut Pasteur | |
| PEAKS Studio | Bioinformatics Solutions Inc. | |
| version Xpro | | |
| Morpheus | Broad Institute; heatmaps | |
| FunRich v3.1.4 | FunRich | |

| Reagent/Resource | Reference or Source | Identifier or Catalog Number |
|---|---|---|
| ShinyGO (v0.80) | South Dakota State University (ShinyGO server) — ShinyGO v0.80 site. | |
| STAGEs (streamlit app; network/ pathway clustering) | Duke-NUS (Streamlit app) | |
| KOBAS (KEGG/ pathway analysis) | KOBAS project | |
| CancerSEA (Cancer Single-cell State Atlas; gene set download portal) | Harbin Medical University (biocc.hrbmu.edu.cn) | |
| Venn diagram web tool (Ghent University webtools Venn) | Ghent University (PSB webtools) | |
| GraphPad Prism | GraphPad Software | |
| Other | | |

## Cell culture

All cell lines were obtained from the American Type Culture Collection (ATCC). BT549 cells were cultured in RPMI 1640 (Gibco, 11554526) supplemented with 10% fetal bovine serum (FBS) (Thermo Fisher Scientific, 11573397) and 1% penicillin/ streptomycin (P/S) (Gibco, 15140-122). MDA-MB-231 and 293FT cells were cultured in Dulbecco's modified Eagle medium (DMEM) (Gibco, 31966-021) with 10% FBS and 1% P/S. Cell lines were grown at 37 °C and 5% $CO_2$. For passaging, cells were detached using Tryple Express (Gibco, 12605-010).

## Immunofluorescence microscopy

For immunofluorescence assays, cells were seeded on glass coverslips in 24-well plates. Following 24 h of treatment, cells were fixed with 4% formaldehyde (Sigma-Aldrich, 252549) for 15 min and washed twice with cold PBS (Gibco). Permeabilization was performed using 0.1% Triton X-100 (Sigma-Aldrich, 9036-19-5) in PBS for 5–10 min at room temperature. After blocking with 3% FBS in PBS for 30 min, cells were incubated overnight with a CD63 antibody (H5C6; Novus Biologicals, NBP2-42225) diluted in the blocking buffer. Thereafter, cells were washed and incubated with a secondary anti-mouse Alexa Fluor 488 antibody (Invitrogen, A-21110) and Phalloidin-iFluor 555 (Abcam, ab176756) for F-actin staining. Nuclei were stained with DAPI (Sigma, D9542-1MG). Imaging was conducted using a Leica Sp8 confocal microscope. CD63 intensity was quantified using ImageJ (v1.53).

## EV isolation

EVs were isolated from the culture medium, supplemented with 5% EV-depleted FBS, as previously described (Théry et al, 2018). Cells were cultured to reach 50–60% confluence before the medium was replaced with fresh medium containing EV-depleted FBS. Cells

were then incubated for an additional 2–3 days. To isolate EVs, we used two different approaches: ultracentrifugation (Merchant et al, 2017; Théry et al, 2018) followed by concentration via EVs isolation tubes (Vivaspin® 20, 100 kDa MWCO Polyethersulfone, Cytiva, 28-9323-63) or size exclusion chromatography. Both methods were compared and used as indicated in the manuscript.

SEC column concentration (qEV) was performed using single-use EV SEC columns (qEV-type, Izon Science) per the manufacturer's instructions. Columns and PBS were equilibrated to 18–24 °C before cap removal. Columns were mounted upright and the loading reservoir was rinsed with PBS. Prior to attaching the reservoir to the column, 5 mL PBS was added to wet the loading frit; flow was allowed to proceed until it reached the frit. The connector was topped up with PBS to avoid air, the reservoir attached firmly, and the column bottom cap removed. Columns were flushed with ≥2 column volumes (CV) of PBS to minimize carryover from storage buffer. With PBS still flowing to the loading frit, the sample was applied directly onto the frit. Collection began immediately to capture the displaced buffer volume (void). Once the sample fully entered the frit, the reservoir was topped up with PBS and sequential fractions of 2.0 mL were collected into pre-labelled low-binding tubes. Based on the column's void profile and our empirical EV marker recovery, fractions 1–5 (2 mL each) were designated EV-enriched (F1–F5) for EVs western blot or pooled ("EV pool"). Fractions 6–10 (2 mL each) were designated protein-enriched (F6–F10) for western blot or pooled ("protein pool"). Pools were kept on ice throughout processing. The pools were reduced in volume using centrifugal filters (10 kDa MWCO) for concentration pre-rinsed with PBS, at 4 °C. EV pools were used immediately or snap-frozen and stored at −80 °C in low-binding tubes. Freeze-thaw cycles were avoided.

## EV labelling, uptake assay, and imaging

EVs were labelled using PKH26 (Sigma, MINI26-1KT) following a quenched-dye workflow. Briefly, same quantities of EVs were used in each group, then 2 µl PKH26 dye was diluted in 500 µl Diluent C (solution A). In parallel, the EV suspension was brought to 500 µl with PBS (solution B). Solutions A and B were combined in a 15 ml conical tube, gently mixed, and incubated for 5 min at room temperature (RT) in the dark. The reaction was quenched by adding 1 ml of 1% BSA (w/v in PBS) and incubating for 1 min at RT. Labelled EVs were isolated using 100 kDa MWCO centrifugal filter units: samples were centrifuged at $4000 \times g$ for 20 min, the retentate was brought to 5 ml with PBS and centrifuged again ($4000 \times g$, 20 min); this PBS wash was repeated once more. The final retentate (PKH26-labelled EVs) was recovered in PBS and used immediately.

For EV uptake experiments, target cells were incubated with labelled EVs under the indicated experimental conditions for 4 h at 37 °C. Cells were then washed twice with PBS, fixed in 4% paraformaldehyde (PFA) for 15 min at RT, and washed again with PBS. For F-actin and nuclear staining, fixed cells were permeabilized and stained with Phalloidin-iFluor 555 (Abcam, ab176756) according to the manufacturer's instructions to visualize F-actin. Nuclei were counterstained with DAPI (Sigma, D9542-1MG), followed by PBS washes. Images were acquired on a Leica TCS SP8 confocal microscope using identical acquisition settings across samples. PKH26 uptake was quantified in ImageJ (v1.53) by

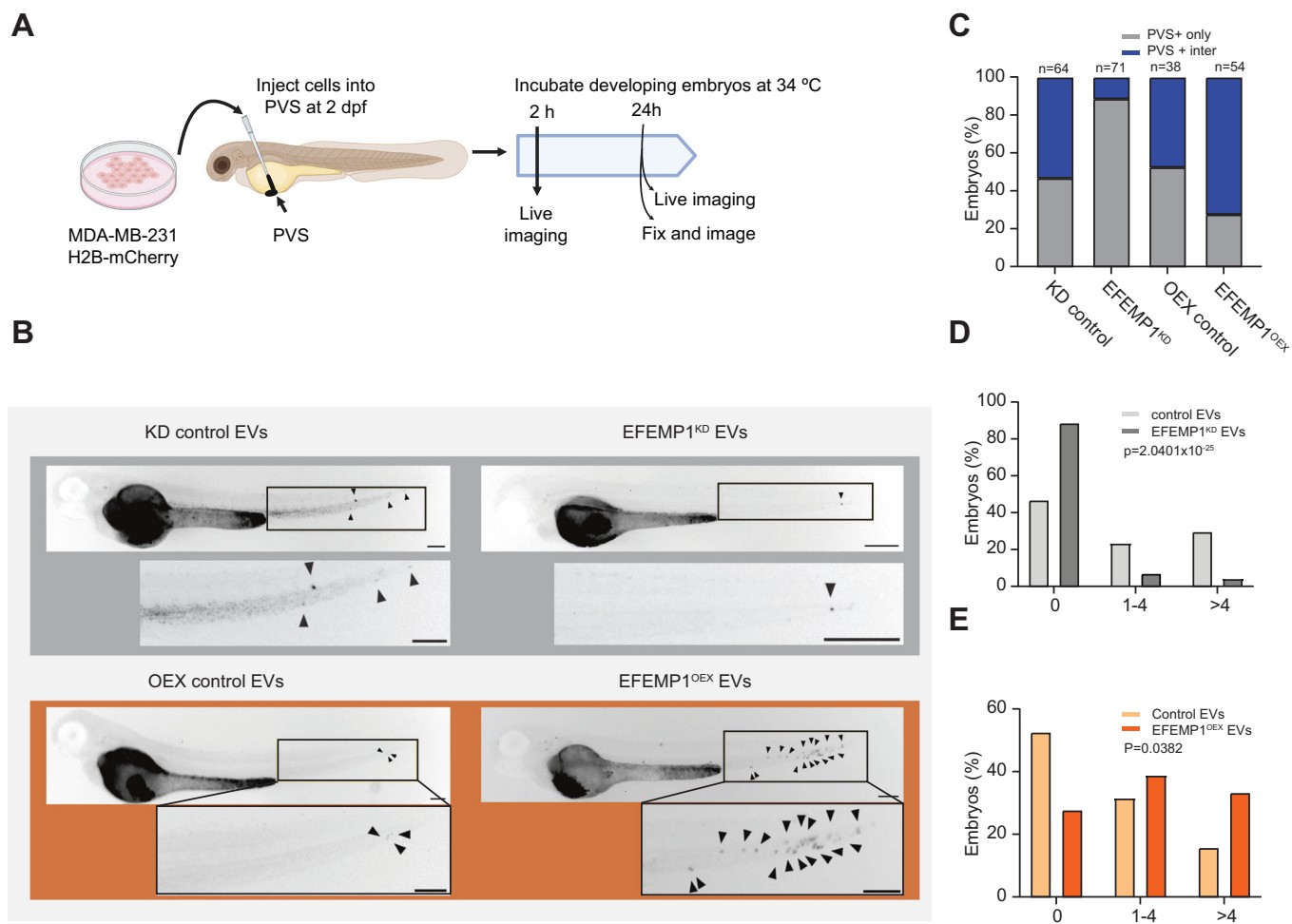

**Figure 5. EFEMP1 promotes MDA-MB-231 cell spreading in a zebrafish xenograft model for metastasis.**

(A) Experimental setup for investigating the invasive behaviour of EV-treated MDA-MB-231 cells in zebrafish embryos. Xenografted embryos were monitored until 1-day post-transplantation (dpt). Image created using a licenced BioRender account. (B) Representative images of embryos xenografted with MDA-MB-231 cells treated with control (left column), EFEMP1$^{KD}$ EVs (right column), or EFEMP1$^{OEX}$ EVs (bottom rows) at 1 dpt. Arrowheads point to migrated MDA-MB-231 cells. Scale bar: 200 μm. (C) Quantification of MDA-MB-231 cell migration treated with EVs as indicated in zebrafish embryos. Significance was tested using a chi-square test (****$p = 3.4084 \times 10^{-30}$). (D) Quantification of EV-treated MDA-MB-231 cell migration toward the zebrafish embryo's tail region. Cells were treated with control or EFEMP1$^{KD}$ EVs as indicated. Significance between groups (control or EFEMP1$^{KD}$ EVs) was tested using a chi-square test and migration patterns (categorized as 0, 1–4, and >4 cells migrated cells). $p = 2.0401 \times 10^{-25}$. (E) Quantification of EV-treated MDA-MB-231 cell migration toward the zebrafish embryo's tail region. Cells were treated with control or EFEMP1$^{OEX}$ EVs as indicated. Significance between groups (control or EFEMP1$^{KD}$ EVs) was tested using a chi-square test and migration patterns (categorized as 0, 1–4, and >4 cells migrated cells). $p = 0.0382$. Source data are available online for this figure.

measuring mean fluorescence intensity per cell (background-subtracted). All image processing and thresholding parameters were applied uniformly to all samples. Where indicated, dye-only controls (PKH26 processed without EVs) were included and carried through the concentration steps to assess non-specific signal.

### EdU proliferation assay

For EdU incorporation assays, cells were plated on coverslips in 24-well plates and treated with EVs for 24 h. For S-phase labelling, cells were pulsed with 10 μM EdU (Lumiprobe, 10540) for 2 h, fixed in 4% formaldehyde (Sigma-Aldrich, 252549), and EdU was detected using click chemistry (Lumiprobe, D1330). Slides were imaged by fluorescence microscopy (Olympus IX51 or BX43), with

image analysis and quantification performed using Fiji (ImageJ v1.53c).

### Western blot assays

Cells or EVs were lysed using RIPA buffer (150 mM Sodium chloride, 50 mM Tris-HCl at pH 8.0, 1% NP-40, 0.5% Sodium deoxycholate, and 0.1% SDS), complemented with a protease inhibitor cocktail (Roche) for 30 min. The cell lysates were combined with 5X SDS buffer, while 4X LDS buffer was utilized for EV lysates. Samples were heated at 95 °C for 5 min to denature proteins.

Protein concentration was quantified using BCA (Pierce™ BCA Protein Assay Kits, 23225, Thermo Scientific™). For this, 20–50 μg of protein was loaded per sample onto 7.5–10% polyacrylamide gels

for electrophoretic separation. Following electrophoresis, proteins were transferred onto PVDF membranes, which were then blocked using Odyssey blocking buffer (LI-COR Biosciences) or 5% non-fat milk in TBS-0.1%T to minimize non-specific binding.

The primary antibodies for Western blots used in this study were EFEMP1 (1:300, Novus Biologicals, NBP1-77040), β-Actin (1:2000, Cell Signaling Technology, 4970 or 3700S), STAT1 (1:1000, Cell Signaling Technology, 9172S), CD63 (1:1000, Novus Biologicals, NBP2-42225), CD81 (1:1000, Novus Biologicals, NB100-65805SS), vinculin (1:10,000, Sigma-Aldrich, V9131-200UL) and Calnexin (1:1000, Novus Biologicals, NB100-1965SS). All antibodies were incubated with the blots overnight at 4 °C.

Secondary antibodies used were IRDye 800CW Goat anti-Rabbit IgG (H + L) (1:15000, LI-COR) and IRDye 680CW Goat anti-Mouse IgG (H + L) (1:15000, LI-COR), or for HRP-conjugated antibodies, Amersham ECL Mouse IgG, HRP-linked whole Ab (1:5000, Cytiva, NA931) and Amersham ECL Rabbit IgG, HRP-linked whole Ab (1:5000, Cytiva, NA934). All of these were incubated for 1 h at room temperature.

Blots incubated with IRDye-conjugated antibodies were imaged using the Odyssey imaging system and analysed using Image Studio Lite software (LI-COR Biosciences). For HRP-conjugated antibodies, protein signals were detected using the ECL Chemiluminescent Kit (Thermo Fisher Scientific) and imaged with an Amersham Imager 600 (GE Healthcare).

## Trans-well assays

To assess cell migration, cells were seeded directly onto trans-well chambers (cellQART, 9328002). For invasion assays, ECM gel (Corning, 356234) was prepared and applied to cool Millicell inserts (cellQART, 9328002) to solidify at 37 °C for two hours (Vasudevan et al, 2020). After a PBS wash, cells suspended in serum-free RPMI were seeded into the upper compartment of both setups. The lower compartment contained RPMI (or DMEM) with 10% FBS as a chemoattractant. Post-incubation, cells on the lower membrane side were fixed, stained with crystal violet, and imaged. Quantitative analysis involved averaging cell counts from three randomly selected fields per membrane at 100x magnification, using an Olympus IX51 fluorescence microscope. Each assay was replicated in triplicate.

## Scratch assays

For scratch assays, cells were detached, counted, and seeded in 6 cm dishes. For gene knockdown, cells were treated with doxycycline in case of inducible shRNAs or transfected with siRNAs as indicated, while gene overexpression was achieved via plasmid transfection, both for 2–3 days. Cells ($3.5 \times 10^5$ for BT549) were then reseeded in 12-well plates. After cell attachment, a 1 mm pipette tip was used to create a straight scratch. Post-scratching, the monolayer was washed with PBS and replenished with an FBS-free medium. Scratch closure was monitored using an IncuCyte machine at 10x magnification, capturing images every 2 h for 24–48 h. Image analysis and quantification were performed using Fiji software (ImageJ 1.53C).

## Zebrafish xenograft experiments

Zebrafish xenograft experiments were performed according to the European animal welfare regulations and standard protocols.

Transparent Casper fish embryos were housed at 31 °C until transplantation. To xenograft MDA-MB-231 cells, an established perivitelline space (PVS) xenograft method was used (Rouhi et al, 2010; Teng et al, 2013; Martinez-Lopez et al, 2021).

Prior to xenografting, MDA-MB-231 WT cells expressing H2B-mCherry were pre-treated with EVs for 3 days and then harvested for microinjection. At 36 hpf, healthy embryos were dechorionated, anaesthetized with tricaine, and arranged in agarose dishes. Cells suspended at $0.12 \times 10^6$ cells/μL were loaded into capillaries with filament and microinjected into the PVS using a pneumatic pico pump (PV820, World Precision Instruments). This process was completed within 1.5 h.

Post-xenograft embryos were transferred to clean E3 medium dishes and examined within 2 h. Criteria for inclusion in the study included embryos with cells solely in the PVS. Embryos displaying obvious differences in initial tumour burden at baseline were excluded from downstream analyses to minimize tumour load-dependent bias in dissemination frequency. Embryos were incubated at 34 °C in E3 medium in individual wells of a 24-well plate for imaging until fixation.

To assess tumour cell dissemination, live embryo images were captured at 2- and 24-h post-transplantation using either a Leica MZ FLIII fluorescence stereomicroscope with a Leica DFC3000G digital camera or a Zeiss Axio Zoom V16 fluorescence stereomicroscope with an HRm digital camera. For higher resolution imaging, fixed xenograft embryos were processed by permeabilization using Proteinase K (10 μg/ml in PBS, Lot: 30032017, Roche) for 15–45 min, depending on embryo age, at 37 °C. Subsequently, the embryos were stained with DAPI (Sigma, D9542-1MG), mounted, and subjected to imaging. The tail regions of embryos harbouring migrated cells were imaged using a Leica SP8x confocal microscope equipped with a 40X oil objective.

## Transmission electron microscopy (TEM)

For TEM, isolated EVs were fixed using 2% paraformaldehyde in 0.1 M sodium cacodylate buffer at pH 7.4 (Théry et al, 2018; Joshi et al, 2020). Grids were prepared with a double layer of Formvar film and carbon coating for enhanced stability. These grids were treated with 1% Alcian blue in 1% acetic acid, followed by thorough rinsing with double-distilled water. 10 μL of the EV suspension was applied to the hydrophilic grid surface and allowed to adhere for 30 min. After washing, the grids were rapidly stained with either 0.5% uranyl acetate or 1% phosphotungstic acid at a neutral pH, crucial for enhancing the EVs' contrast. The samples were incubated at room temperature for 20 min to allow for full interaction between the stain and the EVs and then examined under a Transmission Electron Microscope (Talos200 TFS) (Théry et al, 2018; Joshi et al, 2020).

## Nanoparticle tracking analysis (NTA)

The size and concentration of the EVs were determined using a PMX-130 MONO Laser ZetaView® (Particle Metrix, Ammersee, Germany), with data analysis conducted via ZetaView 8.04.02 software. Calibration was performed using 110 nm polystyrene particles, and measurements were kept consistent at 25 °C. EV samples were diluted in 1×PBS buffer for analysis. Nanoparticle tracking measurements were recorded and analysed at 11 distinct locations per sample. Each experiment was repeated three times.

## CRISPR-Cas9 gene knockouts

To create CRISPR knockout cell lines targeting the human STAT1 gene (details in Expanded View Table EV2), we utilized the pSpCas9(BB)-2A-Puro V2.0 (PX459) plasmid (Addgene plasmid 62988, Feng Zhang) (Ran et al, 2013). sgRNAs were cloned into this Cas9 plasmid using BbsI (NEB). STAT1 knockouts were made by transfecting with these sgRNA plasmids using FuGENE® HD Transfection Reagent (Promega, E2311). Cells underwent puromycin selection (Invitrogen, 0.1–1 µg/ml) for 5 days to select for knockouts. gRNA sequences are provided in Table EV1.

## shRNA knockdowns

For shRNA experiments, we used a Tet-pLKO-puro vector (Addgene plasmid 21915) (Wiederschain et al, 2009). shRNAs were cloned using AgeI and EcoRI restriction enzymes (NEB) and verified by sequencing. Target cells were transduced with these lentiviral constructs and selected using puromycin for 3–7 days. shRNA expression was induced with 1 µg/mL doxycycline (Sigma-Aldrich) in the culture medium. Gene knockdown efficacy was confirmed through quantitative RT-PCR after three days of induction, and protein level changes were assessed by western blotting. The shRNA sequences are provided in Table EV1.

## siRNA knockdowns

Custom siRNAs targeting RAB27A siRNA1 D-004667-01-0020 (5′-GGACAAAGUCUGCAAGUUA), RAB27A siRNA2 D-004667-04-0020 (5′-GCAACAGCCUCAAGAAUUA), and a siGENOME non-targeting siRNA Control Pools #2 (D-001210-02-20) were obtained from Dharmacon. siRNAs were applied at a final concentration of 20 nM for BT549 cell line, unless otherwise specified in the figure legend. A total of 2.5 µl Lipofectamine RNAiMAX (Thermo, 13778075) was used per ml of the final transfection medium. For qPCR, Western blot and trans-well analyses, assays were performed 48 h post-transfection, followed by EVs-treated induction.

## Lentiviral transduction

Lentiviruses were produced by transfecting 293FT cells with the selected vector and essential packaging plasmids (pSPAX2 and pMD2.G, gifts from Didier Trono; Addgene plasmid 12260 and 12259). Post-transfection, the medium was harvested, filtered through a 0.45 µm filter, and used for transducing target cells in the presence of 4–8 µg/ml polybrene (Sigma-Aldrich).

## Protein overexpression

For labelling and tracking the uptake of EVs, we utilized the pLenti-pHluorin_M153R-CD63-mScarlet plasmid (Addgene, Plasmid 172118). This dual-colour fluorescent reporter (Addgene) is specifically designed for identifying CD63-positive exosome secretion and uptake (Sung et al, 2020). EFEMP1 overexpression was achieved by lentiviral transduction (see above) using the FL-fibulin-3 pcDNA4 plasmid (Addgene, Plasmid 29700) (Hu et al, 2009) transfected into 293 T cells using FuGENE® HD Transfection Reagent (Promega, E2311).

## RNA isolation and qRT-PCR

For qPCR analysis, RNA was extracted using the RNA plus isolation kit (Qiagen, 74136). Primer sequences are listed in Table EV1. For cDNA synthesis, 1.5 µg of RNA was added to a LunaScript RT SuperMix Kit (Bioke) in a 20 µl reaction. qPCR was performed using iTaq Universal SYBR Green supermix (Biorad) on a LightCycler® 480 Instrument. Data analysis was conducted using Excel (Microsoft) and Prism software (GraphPad). The qPCR primers are provided in Table EV1.

## Time-lapse imaging

Chromosomal abnormalities were quantified using time-lapse imaging. For this, cell lines were transduced with lentiviral H2B-mCherry constructs. One day prior to imaging, BT549 or MDA-MB-231 cells were seeded into imaging disks (Greiner Bio-One, 627870). Imaging was performed on a DeltaVision Elite microscope (GE Healthcare), fitted with a CoolSNAP HQ2 camera and a 40X, 0.6 NA immersion objective lens (Olympus) for at least 20 h with images captured every 6 min. Each imaging stack consisted of 30 to 40 Z-stacks at 0.5 µm apart. Images were analysed using ICY software (Institut Pasteur). Only mitotic cells were included in the analyses. For live-cell imaging, the cells were pre-treated with the drug one hour before initiating the imaging sessions.

## Mass spectrometry

Coomassie-stained bands were excised in one gel slice that was further cut into small pieces and destained using 70% 50 mM $NH_4HCO_3$ and 30% acetonitrile. Reduction was performed using 10 mM DTT dissolved in 50 mM $NH_4HCO_3$ for 30 min at 55 °C. Next, the samples were alkylated using 55 mM chloroacetamide in 50 mM $NH_4HCO_3$ for 30 min at room temperature and protected from light. Subsequently, samples were washed for 10 min with 50 mM $NH_4HCO_3$ and for 15 min with 100% acetonitrile. The remaining fluid was removed, and gel pieces were dried for 15 min at 55 °C. Tryptic digestion was performed by the addition of sequencing-grade modified trypsin (Promega; 25 µL of 10 ng/mL in 50 mM $NH_4HCO_3$) and overnight incubation at 37 °C. Peptides were extracted using 5% formic acid in water, followed by a second elution with 5% formic acid in 75% acetonitrile. Samples were dried in a SpeedVac centrifuge and dissolved in 20 µL 5% formic acid in water for analysis with LC-MS/MS.

The samples were analysed on a nanoLC-MS/MS consisting of an Ultimate 3000 LC system (Thermo Fisher Scientific, USA) interfaced with a Q Exactive Plus mass spectrometer (Thermo Fisher Scientific). Peptide mixtures were loaded onto a 5 mm × 300 µm i.d. C18 PepMAP100 trapping column with water with 0.1% formic acid at 20 µL/min. After loading and washing for 3 min, peptides were eluted onto a 15 cm × 75 µm i.d. C18 PepMAP100 nanocolumn (Thermo Fisher Scientific). A mobile phase gradient at a flow rate of 300 µL/min and with a total run time 120 min was used: 2–45% of solvent B in 92 min; 45–85% B in 1 min; 85% B during 6 min, and back to 2% B in 0.1 min. Solvent A was 100:0 water/acetonitrile (v/v) with 0.1% formic acid, and solvent B was 0:100 water/acetonitrile (v/v) with 0.1% formic acid. In the nanospray source a stainless-steel emitter (Thermo Fisher Scientific)

was used at a spray voltage of 2 kV with no sheath or auxiliary gas flow. The ion transfer tube temperature was 250 °C. Spectra were acquired in data-dependent mode with a survey scan at $m/z$ 375−1600 at a resolution of 70,000 followed by MS/MS fragmentation of the top 10 precursor ions at a resolution of 17,500. Singly charged ions were excluded from MS/MS experiments, and fragmented precursor ions were dynamically excluded for 20 s.

PEAKS Studio version Xpro (Bioinformatics Solutions, Inc., Waterloo, Canada) software was used to search the MS data against the UniProt reviewed human protein sequence database (19-12-2023, 26,450 entries) with the UniProt reviewed bovine protein database as contaminant database. Search parameters: trypsin digestion with up to 2 missed cleavages; fixed modification carbamidomethylation of cysteine; variable modification oxidation of methionine and deamidation of asparagine and glutamine; precursor mass tolerance of 15 ppm; fragment mass tolerance of 0.02 Da. The false discovery rate was set at 0.1% on the peptide level. Label free quantitation was performed with the PEAKS Q module, with the 3 reversine-treated cell samples and the 3 DMSO-treated cells assigned as separate groups, and with a significance threshold of 13. Quantified results are shown in Dataset EV1.

## Analysis of differentially expressed proteins

To determine differential proteins expression from mass spectrometry results, we utilized Morpheus (https://software.broadinstitute.org/morpheus) for heatmap generation. This analysis was used to identify the top 20 upregulated proteins in EVs derived from REV-treated cells and revealed EFEMP1 as a significantly enriched protein (1.38-fold change, $p = 0.021$). Furthermore, the cellular component enrichment of these differentially expressed proteins was analysed using Funrich 3.1.4. We used ShinyGO v0.80 (http://bioinformatics.sdstate.edu/go/) for gene ontology analyses (Ge et al, 2020). These analyses were complemented by pathway clustering and protein–protein interaction network analyses using STAGEs (https://kuanrongchan-stages-stages-vpgh46.streamlit.app/).

## RNA sequencing sample and library preparation

RNA quality control (QC) involved concentration assessment using Nanodrop and quality and RIN value measurement via HSRNA screentape assay or RNA screentape assay on the Agilent 4200 Tapestation system.

Library preparation employed the Smart-3seq method on an Agilent Bravo Automated Liquid Handling Platform, using 100 ng input per sample. For lower-concentration samples, a pre-bead clean-up (poly-A RNA capturing) with SPRI beads was performed. Indexes utilized were NEBNext Multiplex oligos for Illumina (E6440L) with 12 PCR cycles. Bead clean-up was done twice at a 0.8x ratio.

Post-library preparation QC included concentration measurement using Qubit 1X dsDNA HS Assay Kit and BioTek Synergy H1 Multimode Reader, with molarity assessment via HSD5000 or D5000 screentape assay on the Agilent 4200 Tapestation system.

Libraries were diluted to 4 nM and pooled. Superpool concentration and molarity verification employed the Qubit 1X dsDNA HS Assay Kit, Qubit 4 Fluorometer, and HSD5000 screentape assay.

Sequencing was conducted on the Illumina Nextseq2000 system using a P2 100 cycle kit. Run specifications included a read 1 of 113 bp, read 2 of 7 bp, index 1 of 9 bp, index 2 of 9 bp, loaded molarity of 750 pM, and a phiX percentage of 5%.

## RNA sequencing results analysis

We analysed RNA sequencing data by comparing differentially expressed genes (>2-fold) in EFEMP1[KD] cells to on gene sets related to migration (166 genes) and invasion (97 genes), obtained from the Cancer Single-cell State Atlas (http://biocc.hrbmu.edu.cn/CancerSEA/goDownload) (Yuan et al, 2019). The overlap among these gene sets was visualized using Venn diagrams visualized using an online tool (https://bioinformatics.psb.ugent.be/webtools/Venn/). For the GO gene set enrichment within the RNA sequencing results, we performed KEGG pathway analysis using KOBAS (http://bioinfo.org/kobas/genelist/) (Bu et al, 2021), focusing on pathways with $p < 0.05$.

## Cell adhesion assays

For cell adhesion assays (Khalili and Ahmad, j2019), 96-well plates were first coated with human fibronectin (5 μg/mL; CellSystems; 5050), mouse laminin (5 μg/mL; Thermo Fisher; 23017015), high-molecular-weight poly-l-lysine (50 μg/mL; Sigma-Aldrich; A3890401), or high-molecular-weight hyaluronic acid (g/mL; Calbiochem), followed by rat tail collagen coating solution (Merck Life Science N.V.) (Hu et al, 2008, 2009) for 2 h in incubator. Blocking was performed with 1% BSA in Dulbecco's PBS (Life Technologies; 11500496)). After harvesting of cells, seeding of cells and incubation, plates were washed, fixed, and stained with Crystal Violet, and absorbance was measured at 590 nm as described. This protocol was replicated for three replicates per condition.

## Statistical tests

Data from in vitro experiments were presented as mean ± standard deviation (SD). Comparisons between two groups were conducted using a two-tailed Student's t-test. For multiple group comparisons, we used one-way ANOVA and two-way ANOVA tests followed by Bonferroni's post hoc test to ascertain statistical significance. All analyses were performed using GraphPad Prism 9.5.1 software. Statistical significance was set at $p < 0.05$, with results denoted as *$p < 0.05$, **$p < 0.01$, ***$p < 0.001$. The Pearson correlation coefficient (r value) was calculated to evaluate linear relationships between variables. No predetermined method was employed for sample size determination, and normal distribution or variation of data was not assessed. In vivo study data were expressed as mean ± standard error (SEM). The analysis employed a chi-square test to explore the relationship between treatment groups and migration types in the study of zebrafish results. This statistical method evaluated the observed frequencies across the treatment conditions (control and EFEMP1[KD], or control and EFEMP1[OEX]) and migration categories (0, 1–4, >4).

## Public data analyses

The Breast Invasive Carcinoma dataset was obtained from TGCA using cBioPortal (cBioPortal, n.d.; (Cerami et al, 2012). A search

was conducted on EFEMP1 and STAT1 in cBioPortal to integrate clinical and gene expression data, providing valuable insights into the significance of these genes in breast cancer. The study also integrates information from the METABRIC dataset, which comprises gene expression data produced through microarray technology along with associated clinical data. This dataset was merged with the microarray expression data for additional analysis (cBioPortal, 2012). To validate findings in publicly available human cancer cell line data, the Cancer Dependency Map (DepMap) platform was used. EFEMP1 and STAT1 expression data was compared in connection with the aneuploidy score.

## Data availability

RNA-sequencing data was uploaded to Array Express under accession number E-MTAB-14473. The label-free quantitative proteomics dataset of extracellular vesicles from BT549 cells following 3-day reversine treatment has been deposited in the PRIDE repository under accession PXD073786.

The source data of this paper are collected in the following database record: biostudies:S-SCDT-10_1038-S44318-026-00766-4.

## Peer review information

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

## Acknowledgements

We are grateful to the members of the Foijer lab, van Vugt lab and other labs at ERIBA for fruitful discussion. Siqi Zheng and Ruifang Tian were supported by personal fellowships from the Chinese Scholar Council (CSC). This work was further supported by a Dutch Cancer Society (KWF) and Dutch Research Council (NWO/ZonMw) Vici grant to Foijer (2015-RUG-7822 and 09150182210049, respectively). pLenti-pHluorin_M153R-CD63-mScarlet was a gift from Alissa Weaver (Addgene plasmid 172118; http://n2t.net/addgene:172118; RRID:Addgene_172118). We thank Nancy Halsema, Diana Spierings, Rianna Arjaans, Laura Kempe and Jennefer Beenen for help with RNA sequencing. We are grateful to Karina Köpke for help with imaging. We thank Ben Giepmans, Kim Kats and Anouk Wolters for assistance with TEM imaging. Mass spectrometry was performed at the Interfaculty Mass Spectrometry Center at RUG, with help of Marcel P. de Vries, Hjalmar Permentier and Dr.

Karin Wolters. We thank Joop de Vries, Willem Woudstra, and Hélder A. Santos for providing access to nanoparticle quantification devices. Part of the work has been performed at the UMCG Imaging and Microscopy Center (UMIC), which is sponsored by the Netherlands Electron Microscopy Infrastructure (NEMI; NWO 184.034.014).

## Author contributions

**Siqi Zheng**: Conceptualization; Formal analysis; Funding acquisition; Validation; Investigation; Visualization; Methodology; Writing—original draft; Writing—review and editing. **Ruifang Tian**: Formal analysis; Investigation; Writing—review and editing. **Marsudi Siburian**: Investigation. **Anna Haider Rubio**: Investigation. **Yuanyuan Liu**: Investigation; Methodology. **Rene Wardenaar**: Software; Formal analysis; Visualization; Writing—review and editing. **Marjan Shirzai**: Software; Formal analysis. **Laura Kempe**: Investigation. **Emma Dijkstra**: Investigation. **Eliza Warszawik**: Investigation; Methodology. **Maria Suarez Peredo Rodriguez**: Investigation; Methodology. **Klaas Sjollema**: Methodology. **Petra L Bakker**: Supervision; Investigation. **Patrick van Rijn**: Resources; Supervision. **Michaela Borghesan**: Investigation; Methodology. **Judith TML Paridaen**: Supervision; Methodology. **Stefano Santaguida**: Conceptualization; Supervision; Writing—review and editing. **Floris Foijer**: Conceptualization; Supervision; Funding acquisition; Investigation; Visualization; Writing—original draft; Project administration; Writing—review and editing.

Source data underlying figure panels in this paper may have individual authorship assigned. Where available, figure panel/source data authorship is listed in the following database record: biostudies:S-SCDT-10_1038-S44318-026-00766-4.

## Disclosure and competing interests statement

FF is Chief Scientific Officer of iPsomics B.V. He does not stand financial benefits from this role and the work in this study is unrelated to this role. SS is a consultant for Menarini. The other authors do not declare competing interests.

# Expanded View Figures

**Figure EV1.  Biophysical characterization of EVs.**

(**A**) Transmission electron microscopy (TEM) images showing EV morphology when isolated from BT549 treated cells. Scale bar: 200 nm. (**B**) Ponceau S staining of ultracentrifuge inputs and SEC fractions. (**C, D**) Nanoparticle tracking analysis of EVs isolated from BT549 cells (**C**) and MDA-MB-231 cells (**D**), to determine EV size distribution and EV concentrations. (**E**) Immunofluorescence images of BT549 cells labelled for PKH26 and F-actin incubated for 4 h with indicated EV preparations, compared to DMSO controls. Nuclei were counterstained with DAPI. (**F**) Quantification of PKH26 fluorescence intensity (RFU) in BT549 cells in the presence or absence of 500 nM REV-induced EVs. *$p < 0.05$. RFU, relative fluorescence units. Error bars represent the standard deviation (SD) of the mean. $N = 3$ independent experiments; each data point represents the mean of the replicates; Statistical significance was determined with a paired two-tailed t-test. $p = 0.0165$; *$p < 0.05$. Source data are available online for this figure.

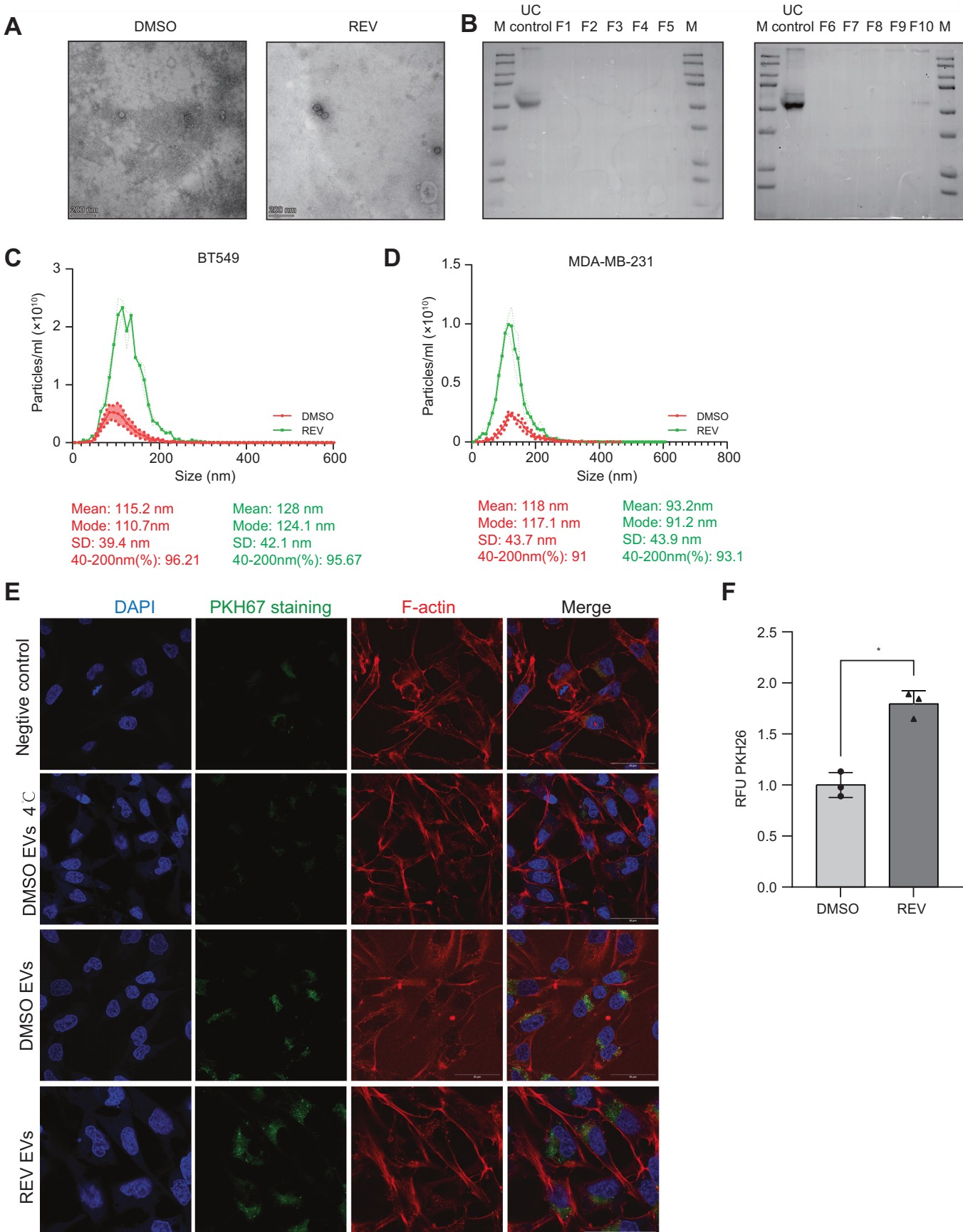

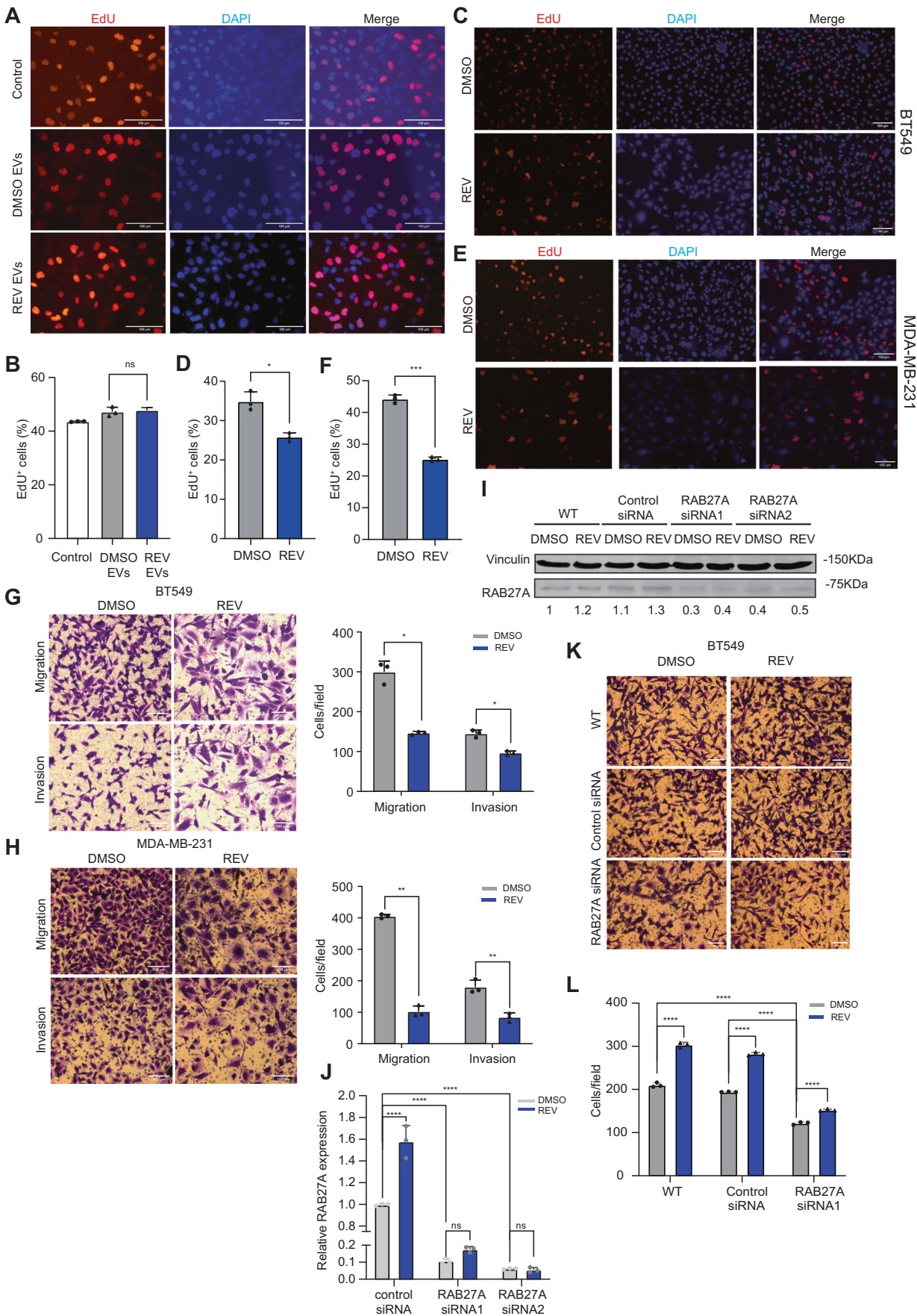

◀

**Figure EV2. CIN drives pro-migratory EVs and requires RAB27A in recipient cells.**

(A, B) Representative images of EdU incorporation in MDA-MB-231 cells treated with EVs from PBS, DMSO or REV treated cells (A) and quantification for three biological replicates (B). Error bars represent the standard deviation (SD) of the mean. $N = 3$ independent experiments; each data point represents the mean of the replicates; Statistical significance was determined with a one-way ANOVA. (C, D) Representative immunofluorescence images of EdU incorporation in BT549 EV-donor cells treated with DMSO or REV prior to EV isolation (C), quantified for three biological replicates (D) Error bars represent the standard deviation (SD) of the mean. $N = 3$ independent experiments; each data point represents the mean of the replicates; Statistical significance was determined with a paired two-tailed t-test. $p = 0.0405$, *$p < 0.05$. (E, F) Representative immunofluorescence images of EdU incorporation in MDA-MB-231 EV-donor cells treated with DMSO or REV prior to EV isolation (E), quantified for three biological replicates (F). Error bars represent the standard deviation (SD) of the mean. $N = 3$ independent experiments; each data point represents the mean of within-experiment replicates; Statistical significance was determined with a paired two-tailed t-test. $p = 0.0004$, ***$p < 0.001$. (G, H) Migration and invasion of BT549 $p = 0.0132$, *$p < 0.05$. (G) and MDA-MB-231 $p = 0.0180$, *$p < 0.05$. (H) cells treated with DMSO (CIN$^{LOW}$) or REV (CIN$^{HIGH}$) using trans-well assays. Statistical significance was determined using two-sided T-tests (*$p < 0.05$, **$p < 0.01$, ***$p < 0.001$, ****$p < 0.0001$). Experiments were performed as biological triplicates. Scale bar: 100 μm. Error bars represent the standard deviation (SD) of the mean. $N = 3$ independent experiments; each data point represents the mean of the replicates; Statistical significance was determined with a paired two-tailed t-test. (I) Western blot quantification of RAB27A knockdown efficacy in BT549 cells. (J) qPCR quantification of RAB27A knockdown efficacy in BT549 cells: Statistical tests were done using two-way ANOVA ($n = 3$; ***, **$p < 0.01$, ***, $p < 0.001$, ****$p < 0.0001$). (K, L) Representative images (K) and quantification (L) of siRNA-mediated RAB27A knockdown in BT549 recipient cells reduces the increases in migration induced by reversine-derived (CIN$^{HIGH}$) EVs. Scale bar: 100 μm. Error bars represent the standard deviation (SD) of the mean. $N = 3$ independent experiments; each data point represents the mean of the replicates; statistical significance was determined with a two-way ANOVA. Source data are available online for this figure.

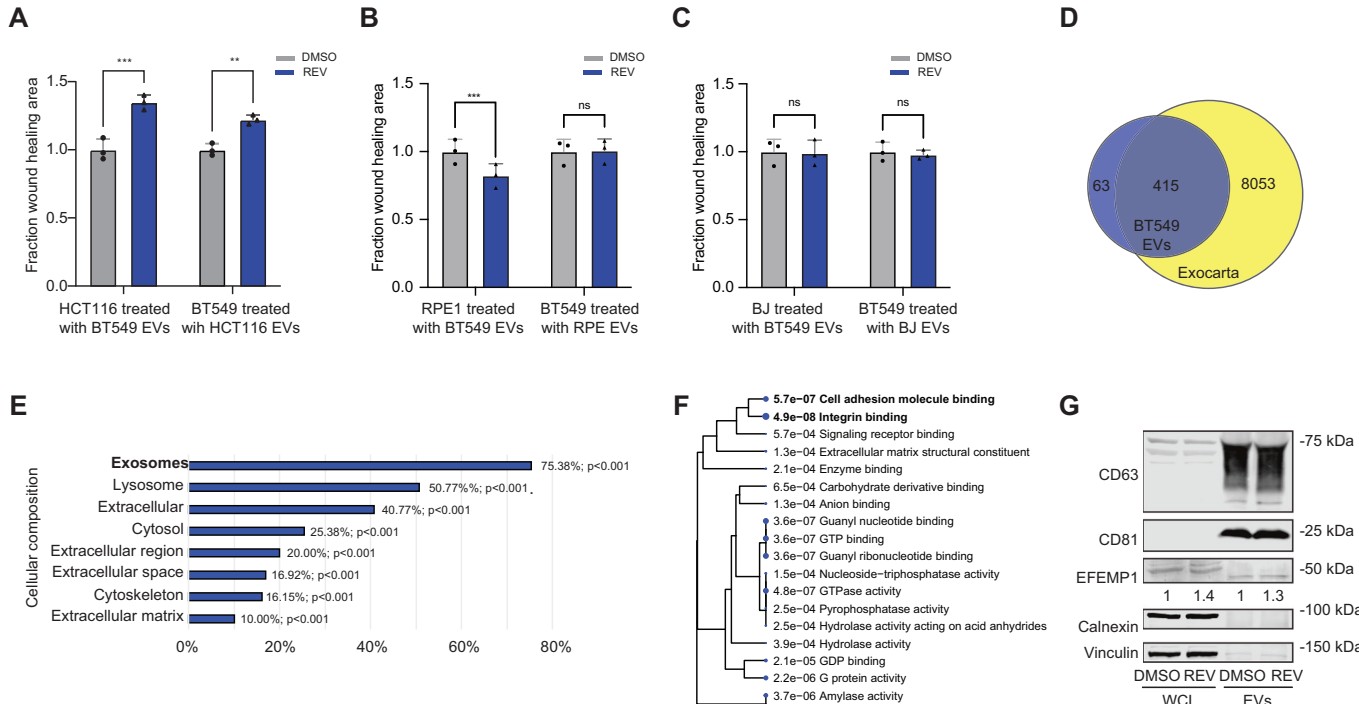

Figure EV3. Reciprocal EV transfer assays, EV proteomics annotation, and EFEMP1 enrichment in CIN^HIGH EVs.

(A) Quantification of scratch-wound migration of HCT116 colon cancer cells treated with BT549-derived EVs generated under 250 nM REV, and the reciprocal condition in which BT549 cells were treated with HCT116-derived EVs at the same doses. Statistical significance was assessed by two-sided t-tests ($n = 3$ biological replicates); $p < 0.01$. Error bars represent the standard deviation (SD) of the mean. (B) Quantification of scratch-wound migration in RPE1 fibroblasts treated with BT549-derived EVs generated under 250 nM REV, and the reciprocal condition in which BT549 cells were treated with RPE1-derived EVs at the same doses. Statistical significance was assessed by two-sided t-tests ($n = 3$ biological replicates); $p < 0.01$. Error bars represent the standard deviation (SD) of the mean. (C) Quantification of scratch-wound migration in BJ fibroblasts treated with BT549-derived EVs generated under 250 nM REV, and the reciprocal condition in which BT549 cells were treated with BJ-derived EVs at the same doses. Statistical significance was assessed by two-sided t-tests ($n = 3$ biological replicates); $p < 0.01$. Error bars represent the standard deviation (SD) of the mean. (D) Venn diagram of peptides enriched in EVs isolated from BT549 cells compared to Exocarta database. (E) Cellular component analysis of differentially expressed proteins using Funrich 3.1.4. software. (F) Gene Ontology (GO) analysis on the proteins enriched in CIN^HIGH EVs compared to CIN- EVs. (G) EFEMP1 levels in BT549 cell lysates and BT549 EVs treated with DMSO (CIN^LOW) or REV (CIN^HIGH) detected by Western blot. Source data are available online for this figure.

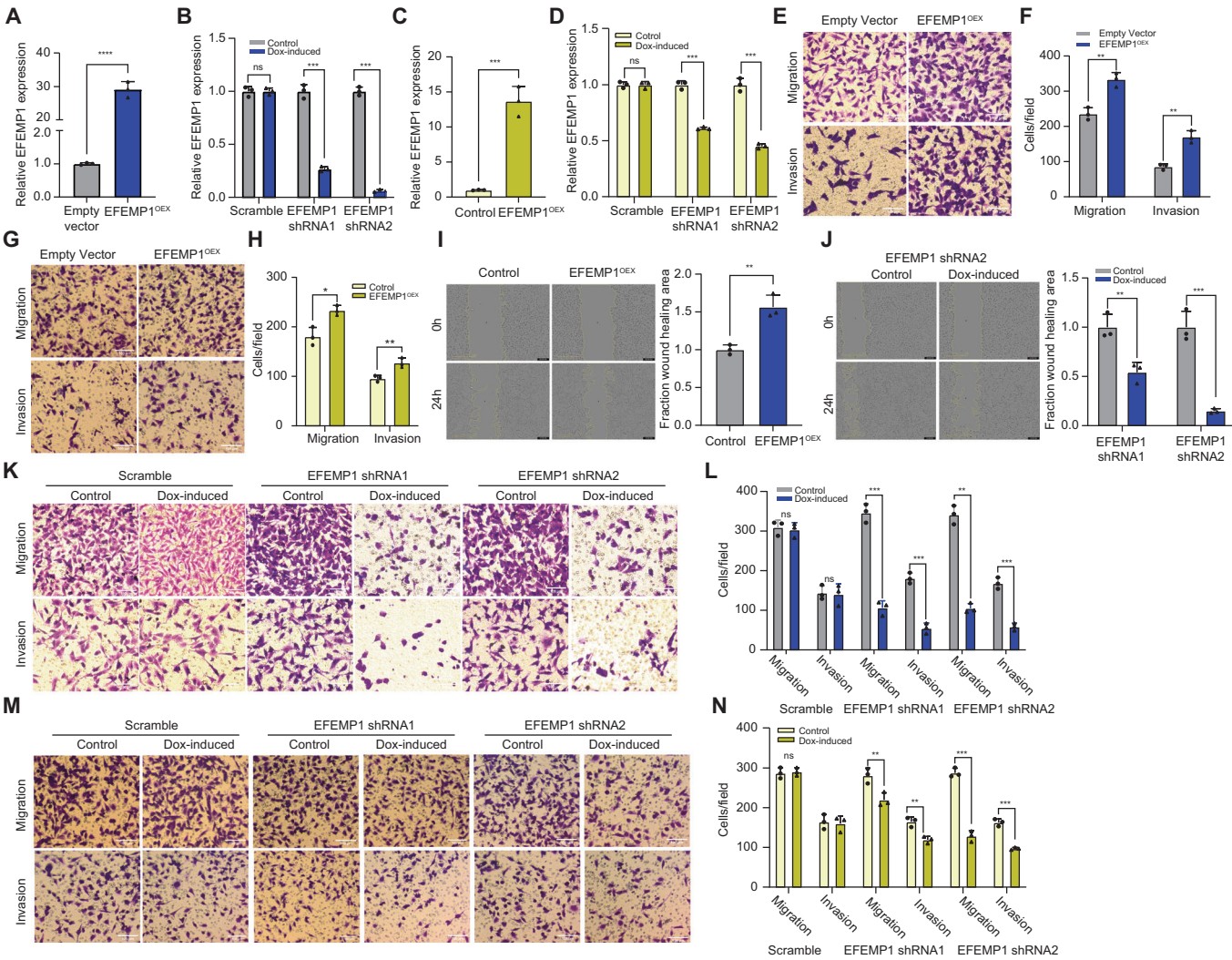

**Figure EV4. EFEMP1 enhances migration and invasion of BT549 and MDA-MB-231 cells.**

(A) qPCR quantification of EFEMP1 knockdown efficacy in BT549 cells. Statistical tests were done using a two-sided T-test ($n = 3$; ***, $p < 0.001$). (B) qPCR quantification of EFEMP1 overexpression in BT549 cells. Statistical significance was assessed by a two-sided T-test ($n = 3$; ****, $p < 0.0001$). (C) qPCR quantification of EFEMP1 knockdown efficiency in MDA-MB-231 cells. Statistical analysis using a two-sided T-test ($n = 3$; ***, $p < 0.001$). (D) qPCR quantification of EFEMP1 overexpression. Statistical significance assessed by a two-sided T-test ($n = 3$; ****, $p < 0.0001$). (E, F) Representative images (E) and quantification (F) of cell migration and invasion of BT549 cells with EFEMP1 overexpression assessed by trans-well assays. Statistical significance was determined by a two-sided T-test ($n = 3$ biological replicates), *$p < 0.05$, **$p < 0.01$, ***$p < 0.001$, ****$p < 0.0001$. Scale bar: 100 µm. (G, H) Representative images (G) and quantification (H) of cell migration and invasion of MDA-MB-231 cells with EFEMP1 overexpression assessed by trans-well assays. Statistical significance was determined by a two-sided T-test ($n = 3$ biological replicates), *$p < 0.05$, **$p < 0.01$, ***$p < 0.001$, ****$p < 0.0001$. Scale bar: 100 µm. (I) Representative images (left panel) and quantification (right panel) of scratch assays to quantify migration of BT549 cells with EFEMP1 knockdown. Statistical significance was determined by a two-sided T-test ($n = 3$ biological replicates), *$p < 0.05$, **$p < 0.01$, ***$p < 0.001$, ****$p < 0.0001$. (J) Representative images (left panel) and quantification (right panel) of scratch assays to quantify migration of BT549 cells with EFEMP1 overexpression. Statistical significance was determined by a two-sided T-test ($n = 3$ biological replicates), *$p < 0.05$, **$p < 0.01$, ***$p < 0.001$, ****$p < 0.0001$. (K, L) Representative images (K) and quantification (L) of cell migration and invasion of BT549 cells with EFEMP1 knockdown assessed by trans-well assays. Statistical significance was determined by a two-sided T-test ($n = 3$ biological replicates), *$p < 0.05$, **$p < 0.01$, ***$p < 0.001$, ****$p < 0.0001$. Scale bar: 100 µm. (M, N) Representative images (M) and quantification (N) of cell migration and invasion of MDA-MB-231 cells with EFEMP1 knockdown (two shRNAs) assessed by trans-well assays. Statistical significance was determined by a two-sided T-test ($n = 3$ biological replicates), *$p < 0.05$, **$p < 0.01$, ***$p < 0.001$, ****$p < 0.0001$. Source data are available online for this figure.

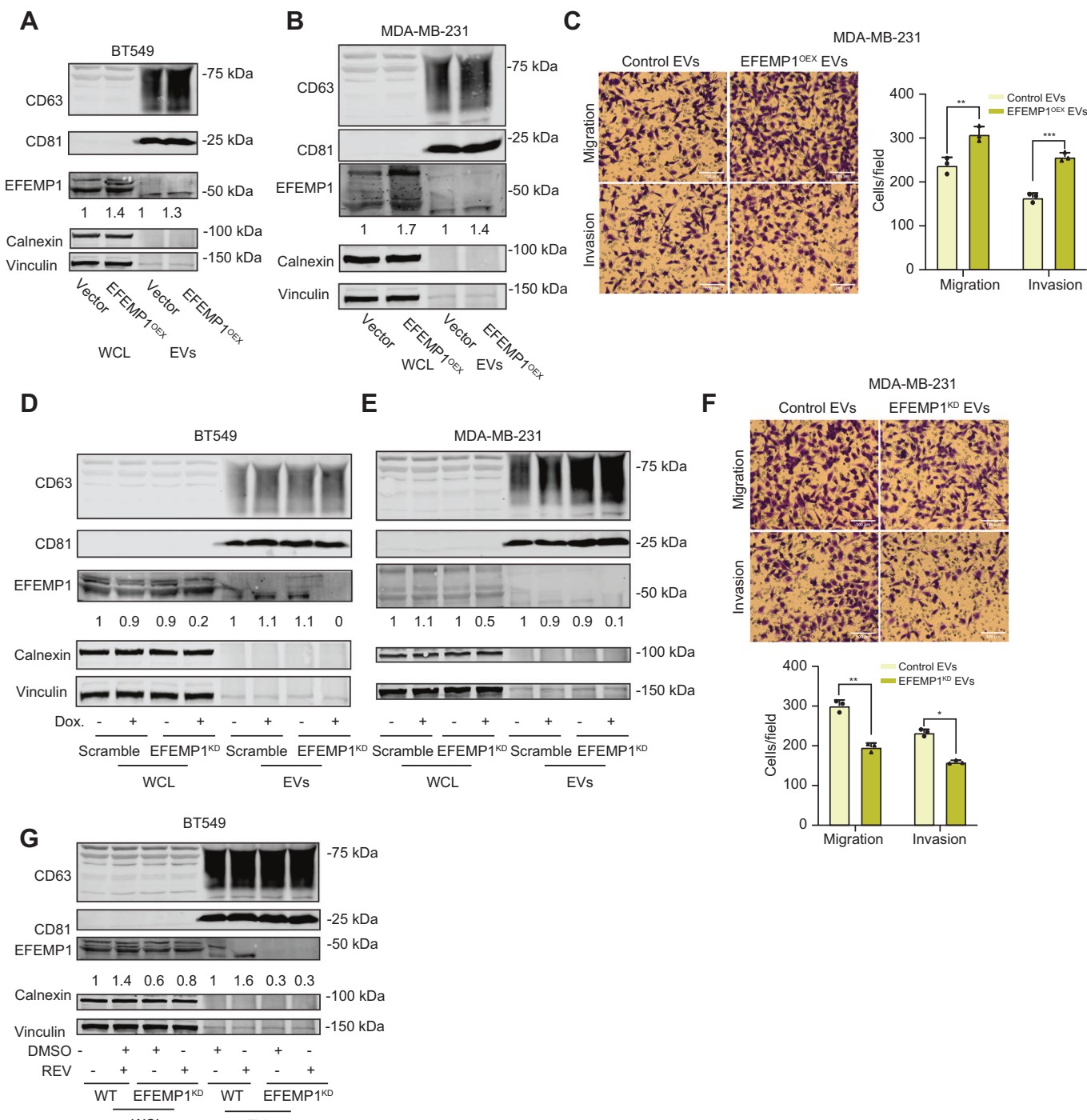

**Figure EV5. EFEMP1 and EV marker expression and EV-mediated migration/invasion in BT549 and MDA-MB-231 cells.**

(A) CD63, CD81, EFEMP1, Calnexin and Vinculin protein levels in whole cell lysates and EVs from BT549 and BT549 EFEMP1^OEX cells determined by Western blot. (B) CD63, CD81, EFEMP1, Calnexin and Vinculin protein levels in whole cell lysates and EVs from MDA-MB-231 cells with or without EFEMP1 overexpression determined by Western blot. (C) Representative images (left panel) and quantification (right panel) of cell migration and invasion of MDA-MB-231 cells treated with EFEMP1^OEX EVs assessed by trans-well assays. Statistical significance was determined by a two-sided T-test (n = 3 biological replicates). Scale bar: 100 μm. (D) CD63, CD81, EFEMP1, Calnexin and Vinculin protein levels in whole cell lysates and EVs from BT549 scramble and EFEMP1 knockdown (EFEMP1^KD) cells determined by Western blot. (E) CD63, CD81, EFEMP1, Calnexin and Vinculin protein levels in whole cell lysates and EVs from MDA-MB-231 control and EFEMP1^KD cells determined by Western blot. (F) Representative images (upper panel) and quantification (bottom panel) of cell migration and invasion of MDA-MB-231 cells treated with EFEMP1^KD EVs assessed by transwell assays. Statistical significance was determined by a two-sided T-test (n = 3 biological replicates), *p < 0.05, **p < 0.01, ***p < 0.001, ****p < 0.0001. Scale bar: 100 μm. (G) CD63, CD81, EFEMP1, Calnexin and vinculin protein levels in whole cell lysates and EVs from DMSO- or REV-treated BT549 scramble and EFEMP1^KD cells determined by Western blot. Source data are available online for this figure.

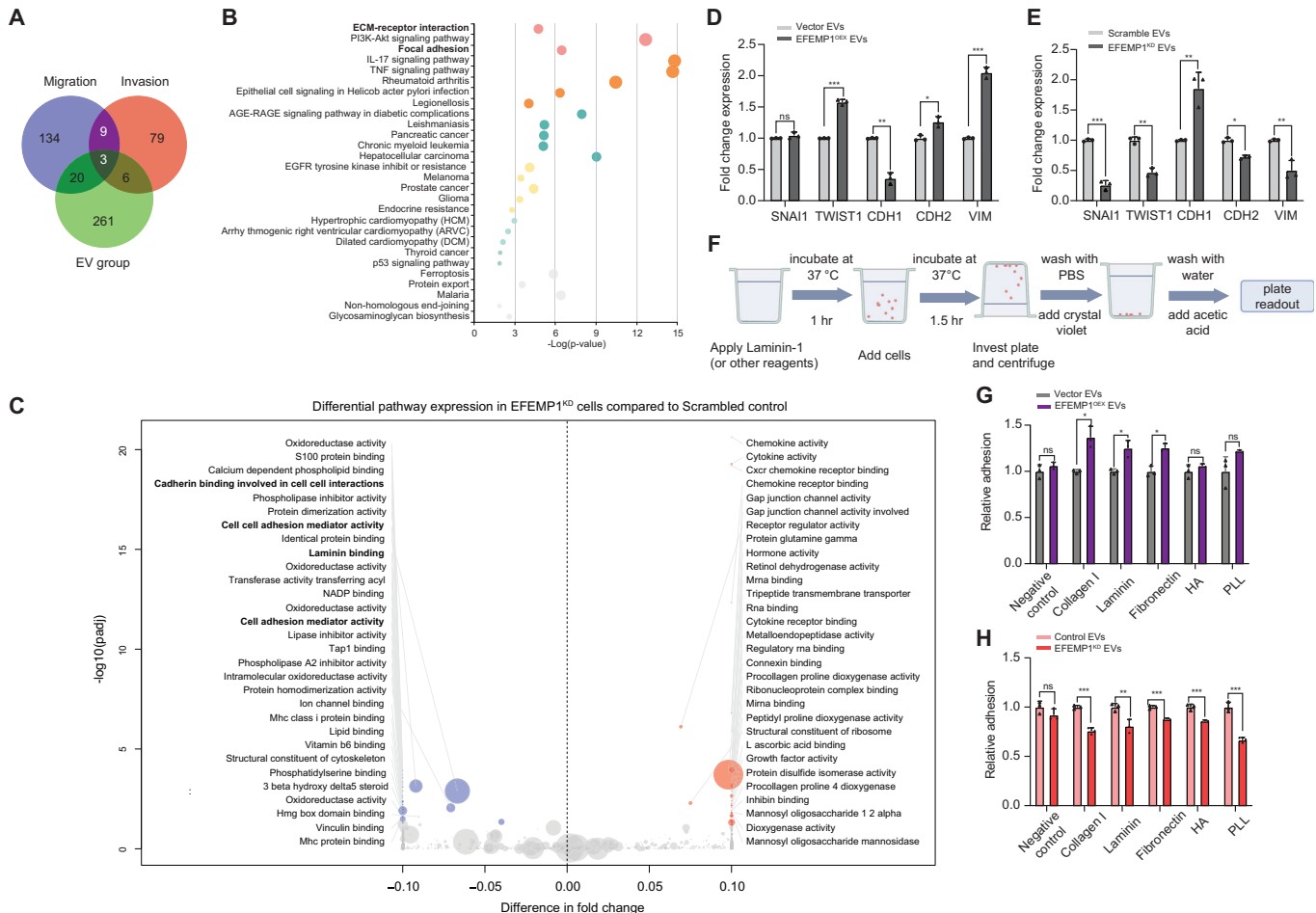

**Figure EV6. EFEMP1 modulates cell adhesion in a paracrine manner.**

(A) Venn diagram of RNA sequencing analysis illustrating the downregulation of 29 genes associated with migration and invasion in BT549 cells treated with EFEMP1[KD] EVs. The gene sets for migration (166 genes) and invasion (97 genes) were derived from the Cancer Single-cell State Atlas, underscoring the significant role of EFEMP1 in regulating cellular migration and invasion patterns (http://biocc.hrbmu.edu.cn/CancerSEA/goDownload). (B) KEGG pathway analysis of the 290 deregulated genes in EFEMP1[KD] cells. (C) Molecular Functions (MF) pathway analysis of the 290 deregulated genes in EFEMP1[KD] cells. (D) qPCR quantification of SNAI1, TWIST, CHD1, CDH2, VIM in vector EVs and EFEMP1[OEX] EVs treated BT549 cells: Statistical tests were done using a two-sided T-test ($n = 3$; ***, **$p < 0.01$, ***$p < 0.001$). (E) qPCR quantification of SNAI1, TWIST, CHD1, CDH2, VIM in scramble EVs and EFEMP1[KD] EVs treated BT549 cells: Statistical tests were done using a two-sided T-test ($n = 3$; ***, **$p < 0.01$, ***$p < 0.001$). (F) Schematic outline of the cell adhesion assay. Image created using a licenced BioRender account. (G) Quantification of cell adhesion of BT549 cells treated with EVs-derived from EFEMP1[KD] cells. (H) Quantification of cell adhesion of BT549 cells treated with EVs-derived from EFEMP1[OEX] cells. Source data are available online for this figure.

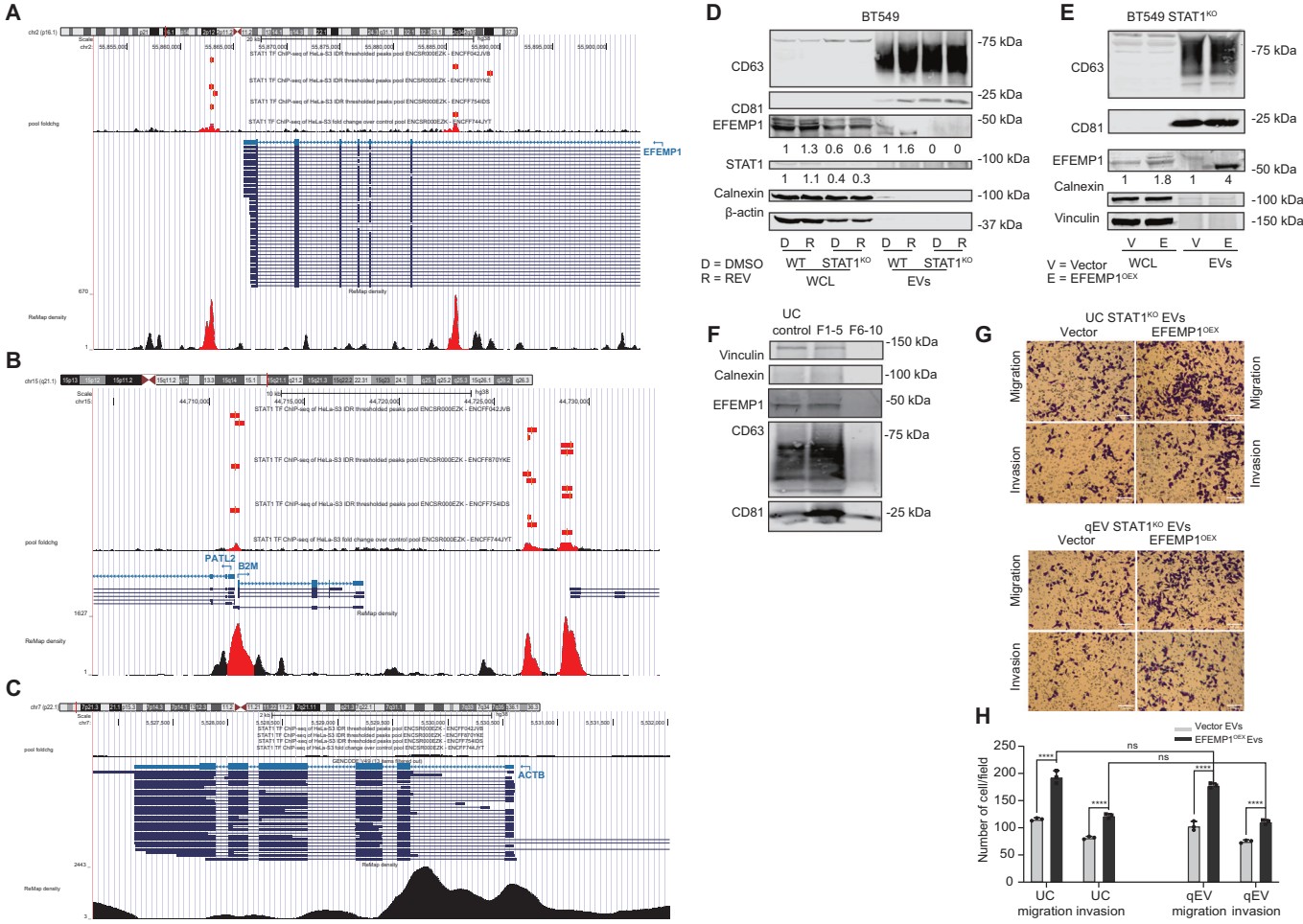

**Figure EV7. EFEMP1 and EV marker expression in BT549 WT and BT549 STAT1^KO cells.**

(A) ChIP-seq-identified STAT1 binding motifs at the human EFEMP1 locus from public ENCODE data. (B) ChIP-seq-identified STAT1 binding motifs at the human B2M locus from public ENCODE data. (C) ChIP-seq-identified STAT1 binding motifs at the human ACTB locus from public ENCODE data. (D) CD63, CD81, EFEMP1, Calnexin and beta-Actin protein levels in whole cell lysates and EVs from BT549 and STAT^KO BT549 cells treated with DMSO or REV determined by Western blot. (E) CD63, CD81, EFEMP1, Calnexin and Vinculin protein levels in whole cell lysates and EVs from STAT^KO BT549 cells treated with or without EFEMP1 overexpression determined by Western blot. (F) Western blots of EV-protein isolates from STAT^KO BT549 cells treated with or without EFEMP1 overexpression fractionated by SEC with fractions 1–5 and fractions 6–10 pooled. (G, H) Representative images (G) and quantification (H) of cell migration and invasion of BT549 cells co-cultured with EVs isolated from STAT1^KO BT549 cells by ultracentrifugation (UC) or size-exclusion chromatography (qEV), treated with vector or EFEMP1^OEX, assessed by transwell assays. Statistical significance was determined by a two-way ANOVA ($n = 3$ biological replicates), *$p < 0.05$, **$p < 0.01$, ***$p < 0.001$, ****$p < 0.0001$. Scale bar: 100 µm. Source data are available online for this figure.

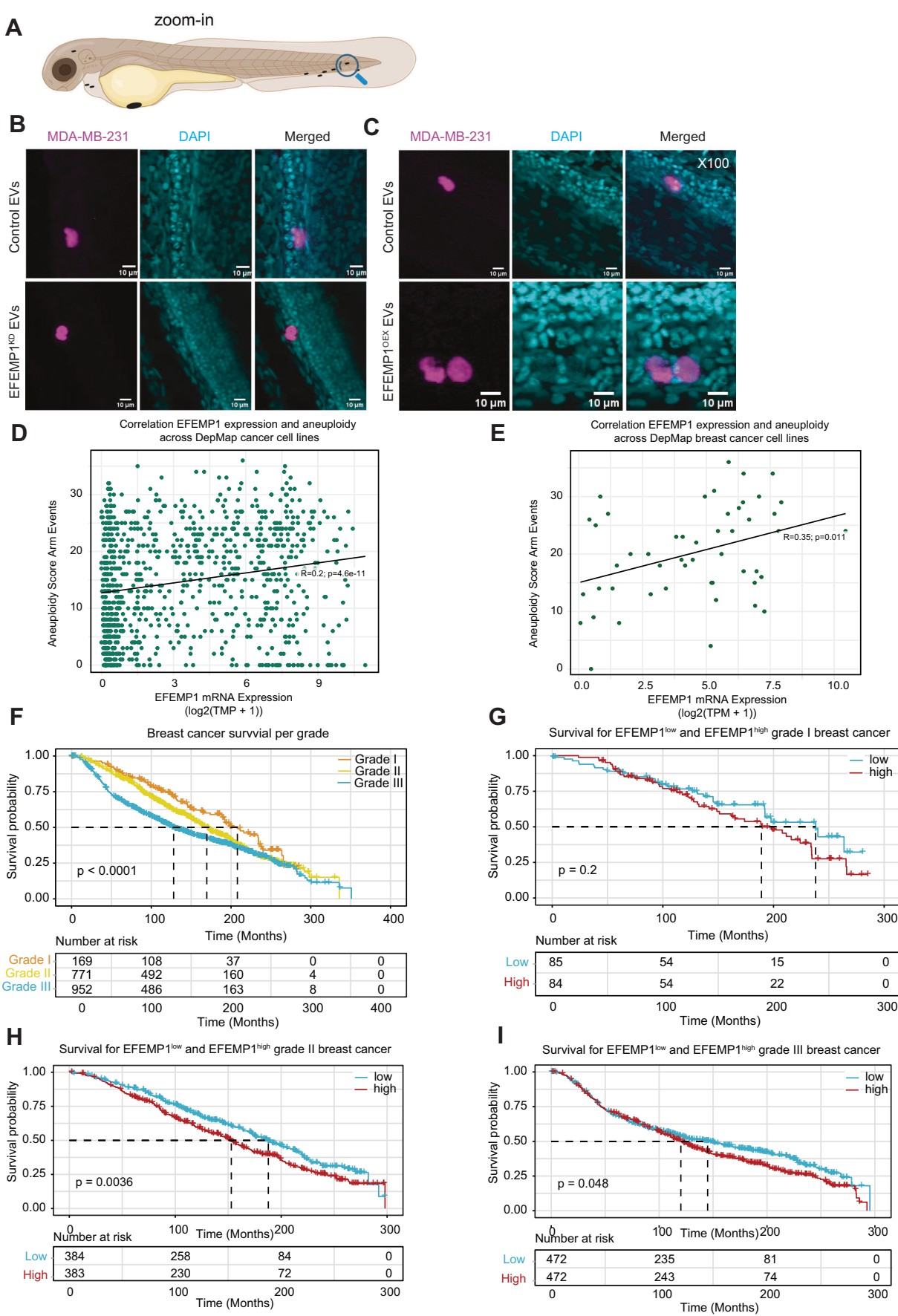

◀

**Figure EV8.  High resolution imaging confirms MDA-MB-231 spreading to the tail of xenografted zebrafish embryos, and EFEMP1 and STAT1 expression patterns in breast cancer progression and prognosis.**

(**A**) Schematic overview of a zebrafish embryo showing the region where migrated MDA-MB-231 cells were monitored. Image created using a licenced BioRender account. (**B, C**) Representative single optical section images of zebrafish embryo tails with mCherry-H2B labelled MDA-MB-231 cells 1 dpt. All nuclei were labelled with DAPI Scale bar: 10 μm. (**D, E**) Scatter plots showing the correlation between EFEMP1 expression and aneuploidy score in DepMap included cancer cell lines (**D**) or DepMap included breast cancer cell lines (**E**). (**F**) Kaplan–Meier survival curves for TGCA-included breast cancer patients, showing overall survival for grade 3 (blue) ($n = 952$), grade 2 (yellow) ($n = 771$) and grade 1 (tangerine) ($n = 169$) breast cancer. (**G–I**) Kaplan–Meier survival curves for TGCA-included breast cancer patients stratified for low or high EFEMP1 expression per grade (grade I, **G**; grade II, **H**, grade III, **I**). Significant differences between EFEMP1 expression groups were tested using a Log-rank Test. Source data are available online for this figure.

