## [Peer Review File · The EMBO Journal]

Chromosomal instability promotes cell migration and invasion via EFEMP1 in extracellular vesicles

Siqi Zheng, Ruifang Tian, Marsudi Siburian, Anna Haider-Rubio, Yuanyuan Liu, René Wardenaar, Marjan Shirzai, Laura Kempe, Emma Dijkstra, Eliza Warszawik, Maria Suarez Peredo Rodriguez, Klaas Sjollema, Petra Bakker, Patrick Rijn, Michaela Borghesan, Judith Paridaen, Stefano Santaguida, and Floris Foijer

Corresponding author(s): Floris Foijer (f.foijer@umcg.nl)

Review Timeline:

Submission Date:	19th Oct 24
Editorial Decision:	3rd Dec 24
Revision Received:	5th Feb 26
Editorial Decision:	2nd Mar 26
Revision Received:	13th Mar 26
Accepted:	17th Mar 26

Editor: Hartmut Vodermaier

Transaction Report:

Dr. Floris Fojjer
University Medical Center Groningen
European Institute for the Biology of Aging
Antonius Deusinglaan 1
Groningen 9713AV
Netherlands

3rd Dec 2024

Re: EMBOJ-2024-119021
Chromosomal instability promotes cell migration and invasion via EFEMP1 in extracellular vesicles

Dear Floris,

Thank you for submitting your study on EV-mediated paracrine effects of CIN for our consideration. I apologize for the delay in getting back to you with a decision, but we have only now finally received a complete set of referee reports, copied below for your information. As you will see, the three reviewers appreciate the overall technical quality of the work as well as the interest and potential importance of your findings. Nevertheless, especially referees 2 and 3 raise several substantive concerns that would need to be satisfactorily clarified, before publication may be warranted; and referee 1 also recommends some experiments to strengthen the insight into underlying mechanisms.

In this situation, I would on balance still give you an opportunity to respond to the reports by way of a revised version of the manuscript, but should emphasize that we can only pursue the study further for The EMBO Journal in case you should be able to strengthen the evidence for the paracrine effects of EFEMP1 being mediated by EV's, as requested by referee 2, and to decisively address the main points raised by referee 3. Furthermore, our single-major-revision-round policy would make it important to diligently respond to all referee points at the time of resubmission. I would therefore encourage you to contact me with a revision plan and preliminary point-by-point response already during the early stages of your revision work, so that we could discuss (via email or Zoom call) if and how key issues raised in the reports may be solved. We would also be open to extension of the default three-months revision period if needed; our 'scooping protection' (meaning that competing work appearing elsewhere in the meantime will not affect our considerations of your study) would of course remain valid also throughout such an extension.

Detailed information on preparing, formatting and uploading a revised manuscript can be found below and in our Guide to Authors. Thank you again for the opportunity to consider this work for The EMBO Journal, and I look forward to hearing from you in due time.

With kind regards,

Hartmut

- size of the scale bars that are mandatory for all micrograph panels
- the statistical test used to generate error bars and P-values
- the type error bars (e.g., S.E.M., S.D.)
- the number (n) and nature (biological or technical replicate) of independent experiments underlying each data point
- Figures may not include error bars for experiments with $n < 3$; scatter plots showing individual data points should be used

instead.

9) To facilitate reproducibility and cross-laboratory adoption of methodologies, please structure the Materials & Methods section as outlined in our guide to authors, including a completed Reagents and Tools Table that can be downloaded from our author guidelines as well (<https://www.embopress.org/page/journal/14602075/authorguide#structuredmethods>).

10) Digital image enhancement is acceptable practice, as long as it accurately represents the original data and conforms to community standards. If a figure has been subjected to significant electronic manipulation, this must be clearly noted in the figure legend and/or the 'Materials and Methods' section. The editors reserve the right to request original versions of figures and the original images that were used to assemble the figure. Finally, we generally encourage uploading of numerical as well as gel/blot image source data; for details see: embopress.org/page/journal/14602075/authorguide#sourcedata

At EMBO Press, we ask authors to provide source data for the main manuscript figures. Our source data coordinator will contact you to discuss which figure panels we would need source data for and will also provide you with helpful tips on how to upload and organize the files.

Further information is available in our Guide For Authors:

In the interest of ensuring the conceptual advance provided by the work, we recommend submitting a revision within 3 months (3rd Mar 2025). Please discuss the revision progress ahead of this time with the editor if you require more time to complete the revisions. Use the link below to submit your revision:

Link Not Available

Referee #1:

The manuscript by Zheng et al. presents a compelling study on the role of chromosomal instability (CIN) in enhancing the migratory and invasive capacities of triple-negative breast cancer (TNBC) cells via EFEMP1 secretion in extracellular vesicles (EVs). This research reveals critical insights into the paracrine effects of CIN within the tumor microenvironment (TME), emphasizing EFEMP1's role as a CIN-induced, STAT1-regulated factor that promotes cellular migration and invasion. Overall, this is an impactful contribution to the understanding of metastasis in TNBC, and the manuscript is well-structured, with a comprehensive set of experiments and analysis.

This study is well-executed and presents a cohesive narrative and will undoubtedly be of interest to a broad audience of cell and cancer biologists. Nonetheless, certain areas require clarification, additional controls, or further experiments to strengthen the claims and broaden the scope of findings. Below are several suggestions that would add significant value to the work.

Major Comments for Improvement:

Broader Functional Validation of EFEMP1-Containing EVs: The study demonstrates that EFEMP1-enriched EVs from CIN-high TNBC cells promote migration and invasion. However, it would be beneficial to include an experiment showing if EVs isolated from non-TNBC cell lines (or a non-cancerous cell line) with artificially induced CIN similarly affect TNBC cells' migration. This would help confirm that the migratory effect is driven by CIN and EFEMP1, rather than by intrinsic properties of the TNBC cell lines used in this study.

Mechanistic Exploration of EFEMP1's Role in Cell Adhesion and Migration: The results suggest that EFEMP1 impacts cell adhesion, yet this connection could be explored further to link adhesion dynamics with migration and invasion. A detailed investigation of specific adhesion-related molecules (e.g., integrins) influenced by EFEMP1-enriched EVs would provide mechanistic depth to these findings and strengthen the link between EFEMP1's secretion and cell adhesion alterations.

STAT1's Downstream Effects on EFEMP1 Expression: While the STAT1-EFEMP1 pathway is suggested as a regulatory mechanism in CIN-high TNBC cells, further validation of STAT1's direct interaction with the EFEMP1 promoter could be insightful. Chromatin immunoprecipitation (ChIP) analysis could confirm STAT1 binding to the EFEMP1 promoter, adding a layer of mechanistic validation.

Additional Experimental Control for EV Uptake: While the uptake of EVs by recipient TNBC cells is demonstrated using fluorescence labeling, adding a control experiment where EV uptake is inhibited (e.g., with heparin or specific uptake inhibitors) could validate that the observed effects on migration are indeed mediated by EV internalization rather than indirect signaling.

In conclusion, this manuscript presents a significant and well-supported investigation into CIN-driven metastasis mechanisms in TNBC, focusing on EFEMP1 as a promising therapeutic target. Implementing the above suggestions will enhance the mechanistic understanding and translational potential of these findings, thereby strengthening the manuscript's impact.

Referee #2:

The manuscript by Zheng et al describes a series of experiments, which suggest that induction of chromosomal instability in triple negative breast cancer (TNBC) cell lines leads to an increase in expression of the extracellular matrix protein EFEMP1. Through knockdown and overexpression experiments, the authors suggest that EFEMP1 promotes cell migration and invasion, and that it can be secreted to stimulate these effects in cell culture potentially in association with extracellular vesicles (EVs). This work is backed up by xenograft experiments in zebrafish and patient studies that suggest high EFEMP1 expression correlates with poorer outcome in grade 1 breast cancer.

As the authors explain, the link between EFEMP1 and cancer metastasis has already been made for TNBC, but the link to chromosomal instability seems to be new, as does the important role of STAT1 in controlling EFEMP1 expression. The association with EVs is perhaps the most significant aspect, but I think this is also the least robust part of the study based on current data.

I have the following concerns:

Major

1. Are the paracrine effects of EFEMP1 mediated by EVs? The authors use a standard ultracentrifugation preparation method (combined with some concentration by filtration) to isolate small EVs (sEVs). As best shown in the study by Jeppesen et al., 2019 (Reassessment of Exosome Composition. Cell. doi: 10.1016/j.cell.2019.02.029), this approach also co-isolates non-vesicular protein aggregates and these can contain high levels of extracellular proteins. These aggregates might also be mediating the effects described and without assessing this, the core conclusions regarding a role for EVs cannot be made. The authors should do the following experiments to clarify this:

- Density gradient (iodixanol) separation of EVs from aggregates and western blots of fractions to identify which particles contain EFEMP1.
- The vesicle fractions could then be tested for activity versus protein aggregates.
- Ideally, it would also be informative try knockdown experiments in secreting cells, which reduce EV secretion. One problem with this approach is that treatments are different for blocking exosomes (Rab27a kd?) versus small ectosomes (ARRDC1 inhibition), but showing that either of these suppressed the effects of paracrine EFEMP1 would strengthen the conclusion that EVs are involved.

2. Several experiments in which the effects of different EV preparations are tested do not have a no-EV control, eg. Figs. 2C/D, 3C/E/G, S5E/F and 5D/E. It is therefore unclear whether EV preps under any condition promote functions like invasion and migration, or whether control EVs preps have no effect or even negative effects. This seems an important point to clarify.

Minor

1. Page 3, line 65 - 'the role of EVs in a chromosomal instability background remains unexplored' - some of the authors have reviewed this topic (Zheng et al., 2023, doi: 10.3390/cells12232712) and should include some of the studies they mention in this review, because they suggest a link between EVs and CIN.

2. Some sEV isolations have contaminants from proteins that should not be present, eg. Fig. 1F - calnexin; S4D - actin; S4F - calnexin and actin; S4G - calnexin. A key reason for doing these experiments is to eliminate contaminating material produced by cell lysis, and this suggests the preparations have additional contaminants, as well as secreted aggregates.

3. Statistics - at the very least, multiple comparisons should use an ANOVA test, assuming parametric data, not a t-test, eg. Fig. 3G and 4E, and this will apply to other data where a no-EV control is included.
4. In Fig, 4C, shouldn't the blue and purple bars also be compared to determine whether STAT1 controls some of the other genes?
5. For Fig. S4G and S6B, the EFEMP1 westerns appear overexposed or too much protein has been loaded for the overexpressing EFEMP1 cells.
6. Have the proteomics data been deposited in a public repository? Since EFEMP1 is an ECM protein that has been shown to influence other ECM protein organisation, I wondered whether there were any of these proteins in the proteomics list, which could have been tested by western in cells/EVs where EFEMP1 is knocked down or overexpressed to determine whether these are co-regulated on EVs or extracellular non-vesicular particles. This is relevant in the context that other ECM proteins have already been suggested to regulate migration on EVs, eg fibronectin, Sung BH, et al. 2015, (Directional cell movement through tissues is controlled by exosome secretion. Nat Commun. 2015 doi: 10.1038/ncomms8164), a study that might be mentioned in the introduction.

Text

Page 3, line 58 - 'microvesicles' are now generally called 'ectosomes' in the literature

Page 4, line 100 - use 'isolated' or 'separated and concentrated', not 'purified', partly for the reasons discussed above

Line 103-4 - 'confirmed that the isolates contained EVs' - there are no assays presented to look for common contaminant secreted protein aggregates.

Referee #3:

In this manuscript, Zheng et al investigate the link between CIN and invasion in TNBC. In particular, the authors examined the paracrine effect of CIN cancer cells in the migration/invasion of the same cells. For that, they went on to test how CIN impacts EVs secretion, since EVs play roles in cell-cell communication in cancer. They found that inducing CIN by treating cancer cells with reversine promotes the secretion of EVs and that these EVs can promote migration and invasion of cancer cells. Proteomics analyses of these EVs suggest that these phenotypes could be induced by the extracellular matrix protein EFEMP1. Depletion of EFEMP1 from breast cancer cell lines diminishes significantly the ability of EVs to promote migration/invasion. The authors propose that STAT1 expression is linked to EFEMP1 expression.

Overall, the work presented here is novel and experiments well conducted. However, below concerns are raised about some of the conclusions and lack of quantifications necessary to support some of the claims. The main concern is the induction of CIN phenotype and how specific what is described here is. The authors should address these points before publication.

Major comments:

1. The choice of cell lines was unclear, apart from the fact that these are TNBC. I assume that both cell lines already display CIN? Thus, it was unclear why the need of further inducing CIN and not choosing cells lines that have no CIN to compare? And induce CIN on those?
2. I could not find in the manuscript how the reversine treatments were performed (apart from concentration). This information needs to be added, including on the main text as it is critical for the interpretation of the data. The main reason I am bringing this as a major comment is related with the above comment and the graphs in Fig 1A where a variety of phenotypes, including some that will promote tetraploidy and cell arrest. How sure are the authors that what they identified here is a consequence of CIN and not other more significant phenotypes that culminate with severe cell arrest, mitotic failure, senescence, apoptosis and DNA damage?
3. Related to the comment above, the morphology of cells after reversine changes significantly, as seen for example in the migration and invasion assays in Fig 2C-D. are some of these cells on their way to become senescence? And hence the changes in the secretory phenotype the authors observed?
4. Uptake of vesicles by the cancer cells needs to be quantified by different methods. The movie shows vesicles on cells but there is no indication that these vesicles enter the cells. Could the authors quantify the % of cells that uptake vesicles and quantify vesicle uptake for example using confocal microscopy and assessing vesicles inside cells? (z-stacks)
5. The heatmap provided in Fig 2E represents all proteins from the mass spec data that are significantly changed between control and reversine treated EVs? How many of those have been shown to be associated with EVs?
6. Fibulin 3 (EFEMP1) was identified in the label-free proteomics samples as increased upon reversine treatment. However, as seen in on the WB in Suppl Fig 2F there is little to know difference observed between DMSO and reversine treated EVs. It would be great if the authors could show quantifications of the WB data that is key for their conclusions as not only that is not provided but significant variability is observed between WBs. The WB for CD63 and CD81 are not ideal and might be difficult to use as loading controls, which is necessary to quantify the amount of EFEMP1 associated with EVs.

7. the WB on suppl figure 4 are not conclusive. This is a key part of the work. The levels of EFEMP1 vary significantly and no quantifications are provided. As an example, on panel F, EFEMP1 is observed both on lysates and EVs from MDA-231 cells but that is not the same on panel G where EFEMP1 is barely detected. This variability makes it hard to conclude that the overexpression construct is working and increased EFEMP1 is observed in the lysates and extracts because on that WB it is barely seen in the control conditions (which is different from other WB). On panel D regarding overexpression of EFEMP1 in BT-549 cells, the authors also claim that EFEMP1 can be detected at higher levels in EVs, however CD63 levels are also increase and thus difficult to conclude if that is the case without quantifications.

8. The authors conclude that EFEMP1 associated with EVs is important for the paracrine EV-mediated migration/invasion. However, loss of EFEMP1 also affects the same phenotypes in DMSO treated cells. Could the authors comment on the specificity of the phenotype? The same is observed for STAT1 KD.

9. I find the correlation plots on Fig 4A-B and Fig 6A-B very difficult to interpret as the correlations are not obvious and how significant these data is is unclear. For example, on Fig 4A, when STAT1 is expressed from log2 4 to 8 where most data points are there is no correlation that is obvious. The interpretations should be toned down.

10. Data on patient survival does not support a strong association of EFEMP1 and STAT1 with progression and it is unclear whether it could be used as prognostic as differences are very minor. In my opinion, this is not needed to validate the findings in patients.

Minor comments:

11. In the abstract the authors state that the TME is modulated by EVs. However, soluble secreted proteins have been shown to play key roles in this process (cytokines, chemokines for example). It is important that it is clear that EVs are just one type of modulators and the authors should write the abstract and the manuscript with this in mind. Also, this manuscript does not investigate TME, the initial part of the abstract is misleading by those reasons.

12. Graphs on Figure 1A are better placed in the supplementary data.

13. It is very difficult to extract the data from the proteomics analyses on Table 1. The table should provide a list of protein names, IDs, all separated in different columns so that anyone can access the data. The authors should also compare their data sets with publicly available datasets for EVs proteomics. It is important to know how much overlap there is to understand how well represented this data set is.

14. First long paragraph on page 10 needs clarity as it is difficult to read and understand.

15. I could not find information about how the treatments with EV were mediated. This needs to be explained, were EVs normalised for number? How long were cells treated with EVs?

16. Figure 5 is not easy to follow. At what time invasion is quantified? In all time points on the scheme in 5A? if so, could the authors then provide the data by time point?

17. How the authors envision that EFEMP1-loaded EVs could promote migration/invasion? Is this protein inside the EVs? Outside? How much of it needs to be incorporated in EVs to make an impact on cells? Some of this should be discussed as it is not immediately clear mechanistically how this operates.

European Research Institute for the Biology of Ageing
University Medical Center Groningen, University Groningen
P.O. Box 196, 9700 AD, Groningen, the Netherlands

Groningen, 5 Feb 2026

Re: EMBOJ-2024-119021R

Floris Foijer, PhD
Full professor
Tel: +31 (0)6 5272 4864
Email: f.foijer@umcg.nl

Dear Hartmut,

Many thanks for your assessment of our manuscript and the reviewers' comments.

We have now prepared a revised manuscript in which we have addressed all comments of all the reviewers carefully. We have described this in the point-by-point response letter below and copied in the relevant new data into this letter to facilitate review. We think that the manuscript is significantly improved and hope that our revisions take away the concerns that were raised by the reviewers. We also hope that you now find our manuscript suitable for publication in EMBO Journal.

We also want to take this opportunity to thank both you and the reviewers for the efforts made to improve our manuscripts and bring it up to the high standards of EMBO journal.

We look forward to the outcome of your and the reviewers' assessments of this revised version.

Best wishes,

Floris

We want to start out by thanking the reviewers for the comprehensive and constructive review of our study. We have worked hard to address all comments raised by the three reviewers comprehensively as described point by point below and copied in the relevant data to facilitate review. We also added a PDF version of the manuscript with the figures copied in throughout the text and a version in which all changes are tracked compared to the previous version.

Referee #1:

The manuscript by Zheng et al. presents a compelling study on the role of chromosomal instability (CIN) in enhancing the migratory and invasive capacities of triple-negative breast cancer (TNBC) cells via EFEMP1 secretion in extracellular vesicles (EVs). This research reveals critical insights into the paracrine effects of CIN within the tumor microenvironment (TME), emphasizing EFEMP1's role as a CIN-induced, STAT1-regulated factor that promotes cellular migration and invasion. Overall, this is an impactful contribution to the understanding of metastasis in TNBC, and the manuscript is well-structured, with a comprehensive set of experiments and analysis.

This study is well-executed and presents a cohesive narrative and will undoubtedly be of interest to a broad audience of cell and cancer biologists. Nonetheless, certain areas require clarification, additional controls, or further experiments to strengthen the claims and broaden the scope of findings. Below are several suggestions that would add significant value to the work.

Many thanks for this kind assessment.

Major Comments for Improvement:

1.1 Broader Functional Validation of EFEMP1-Containing EVs: The study demonstrates that EFEMP1-enriched EVs from CIN-high TNBC cells promote migration and invasion. However, it would be beneficial to include an experiment showing if EVs isolated from non-TNBC cell lines (or a non-cancerous cell line) with artificially induced CIN similarly affect TNBC cells' migration. This would help confirm that the migratory effect is driven by CIN and EFEMP1, rather than by intrinsic properties of the TNBC cell lines used in this study.

This is an excellent point. To address this, we performed additional experiments in another cancer cell line (different cancer type; colon cancer) and two non-transformed cell lines. 1) We induced CIN in HCT116 (diploid colon cancer), BJ cells (non-transformed diploid fibroblasts) and RPE1 cells (non-transformed near diploid-retinal epithelial cells) using 250 nM of REV, isolated EVs secreted by these cells and added these to BT459 TNBC cells. We then measured migration of these EV-recipient BT459 cells using scratch assays. We find that EVs isolated from CIN^{HIGH} HCT116 cells promote the migration of BT549 recipient cells and *vice versa*, confirming that migration induced by EVs isolated from CIN^{HIGH} cancer cells is not restricted to TNBC (Sup. Fig. 3A). Conversely, when isolating EVs from non-transformed cells with induced CIN, we find that BT549 migration is modestly decreased (EVs isolated from CIN^{HIGH} RPE1 cells on BT549) or no effect (EVs from CIN^{HIGH} BJ cells on BT549). We also fail to see an effect of EVs isolated from CIN^{HIGH} BT549 on these untransformed cell lines. These data are now shown in Sup. Fig. 3B and C. Together, these results indicate that while the phenotype might also occur in other cancer cell types, it cannot be universally extrapolated

to non-transformed cell lines.

Supplementary Figure 3. Reciprocal EV transfer assays, EV proteomics annotation, and EFEMP1 enrichment in CIN^{HIGH} EVs. (A) Quantification of scratch-wound migration of HCT116 colon cancer cells treated with BT549-derived EVs generated under 250 nM reversine, and the reciprocal condition in which BT549 cells were treated with HCT116-derived EVs at the same doses. Statistical significance was assessed by two-sided t-tests (n = 3 biological replicates); p < 0.01. (B) Quantification of scratch-wound migration in RPE1 fibroblasts treated with BT549-derived EVs generated under 250 nM reversine, and the reciprocal condition in which BT549 cells were treated with RPE1-derived EVs at the same doses. Statistical significance was assessed by two-sided t-tests (n = 3 biological replicates); p < 0.01. (C) Quantification of scratch-wound migration in BJ fibroblasts treated with BT549-derived EVs generated under 250 nM reversine, and the reciprocal condition in which BT549 cells were treated with BJ-derived EVs at the same doses. Statistical significance was assessed by two-sided t-tests (n = 3 biological replicates); p < 0.01.

1.2 Mechanistic Exploration of EFEMP1's Role in Cell Adhesion and Migration: The results suggest that EFEMP1 impacts cell adhesion, yet this connection could be explored further to link adhesion dynamics with migration and invasion. A detailed investigation of specific adhesion-related molecules (e.g., integrins) influenced by EFEMP1-enriched EVs would provide mechanistic depth to these findings and strengthen the link between EFEMP1's secretion and cell adhesion alterations.

Many thanks for this suggestion. We have extended our work to better understand how EFEMP1 and cell adhesion are related. To this end, we selected a number of genes that have a known role in cell adhesion (SNAI1, TWIST1, CDH2 (N-cadherin), VIM and CDH1 (E-cadherin)) and tested whether expression of these genes is indeed affected when cells are treated with EVs isolated from EFEMP1^{OEX} or EFEMP1^{KD} cells. Indeed, in BT549 cells exposed to EFEMP1-enriched EVs, TWIST1, CDH2 (N-cadherin), and VIM were upregulated, whereas CDH1 (E-cadherin) was downregulated (Sup. Fig. 6D), a canonical “cadherin switch” associated with vimentin-dependent cytoskeletal reorganization, integrin activation, and increased focal-adhesion turnover, facilitating traction and motility (Hałas-Wiśniewska *et al*, 2025; Ostrowska-Podhorodecka *et al*, 2021). Conversely, EFEMP1-knockdown EVs yielded the opposite effect (Sup. Fig. 6E), indicating that EFEMP1 in EVs can drive EMT-like transcriptional reprogramming and the associated adhesion dynamics.

Supplementary Figure 6 EFEMP1 modulates cell adhesion in a paracrine manner. (D) qPCR quantification of SNAI1, TWIST, CHD1, CDH2, VIM in vector EVs and EFEMP1^{OEX} EVs treated BT549 cells: Statistical tests were done using a two-sided T-test (n=3; ***, **p < 0.01, ***p < 0.001). (E) qPCR quantification of SNAI1, TWIST, CHD1, CDH2, VIM in scramble EVs and EFEMP1^{KO} EVs treated BT549 cells: Statistical tests were done using a two-sided T-test (n=3; ***, **p < 0.01, ***p < 0.001).

1.3 STAT1's Downstream Effects on EFEMP1 Expression: While the STAT1-EFEMP1 pathway is suggested as a regulatory mechanism in CIN-high TNBC cells, further validation of STAT1's direct interaction with the EFEMP1 promoter could be insightful. Chromatin immunoprecipitation (ChIP) analysis could confirm STAT1 binding to the EFEMP1 promoter, adding a layer of mechanistic validation.

Thank you for bringing this up. Indeed, EFEMP1 was previously reported as a STAT1 target gene in a ChIP-seq study (Satoh & Tabunoki, 2013). To confirm this, we set out to confirm STAT1 binding to the EFEMP1 locus. Unfortunately, setting up CUT&TAG (the approach we chose) is taking more time than anticipated and we therefore so far not yet succeeded in establishing our own STAT1-binding dataset. As an alternative, we mined public STAT1 ChIP-seq datasets, which show STAT1 occupancy at multiple sites across the EFEMP1 locus in human cells (Sup. Fig. 7A). As positive and negative controls, we show (lack of) binding to the B2M locus (Sup. Fig 7B, an established STAT1-target gene, (Satoh & Tabunoki, 2013; Neerinx *et al*, 2013)) and Actin(Satoh & Tabunoki, 2013; Robertson *et al*, 2016) (Sup. Fig 7C). These results are in line with our finding that EFEMP1 expression is steeply decreased in STAT1^{KO} cells (Fig. 4C).

Supplementary Figure 7 EFEMP1 and EV marker expression in BT549 WT and BT549 STAT1^{KO} cells. **(A)** ChIP-seq-identified STAT1 binding motifs at the human EFEMP1 locus from public ENCODE data. **(B)** ChIP-seq-identified STAT1 binding motifs at the human B2M locus from public ENCODE data. **(C)** ChIP-seq-identified STAT1 binding motifs at the human ACTB locus from public ENCODE data.

Figure 4 STAT1 is required for CIN-induced EFEMP1 expression and its downstream effects on migration and invasion. (C) qPCR quantification of IL6, IL8, CXCL1, CXCL10, EFEMP1, STAT1, STAT3, CD63 and CYLD in BT549 WT and STAT1 KO cells. Statistical analysis was done using two-way ANOVA (N = 3; *p < 0.05, **p < 0.01, ***p < 0.001, ****p < 0.0001).

1.4 Additional Experimental Control for EV Uptake: While the uptake of EVs by recipient TNBC cells is demonstrated using fluorescence labeling, adding a control experiment where EV uptake is inhibited (e.g., with heparin or specific uptake inhibitors) could validate that the observed effects on migration are indeed mediated by EV internalization rather than indirect signaling.

Excellent point! To address this comment, we have:

1) Labelled EVs using PKH26 staining, which confirms that EVs are taken up by recipient cells at 37°C much more efficiently compared to when cells are at 4°C which is known to block EVs uptake. Furthermore, this labelling confirms increased uptake of EVs when EVs are isolated from CIN^{HIGH} cells, in line with their increased secretion of EVs. Data for these experiments are shown in Sup. Fig 1E, F.

Zheng *et al*, Sup. Figure 1

Supplementary Figure 1. Biophysical characterization of EVs. (E) Immunofluorescence images of BT549 cells labelled for PKH26 and F-actin incubated for 4 h with indicated EV preparations, compared to DMSO controls. Nuclei were counterstained with DAPI. (F) Quantification of PKH26 fluorescence intensity (RFU) in BT549 cells in the presence or absence of 500 nM REV-induced EVs. * $p < 0.05$. RFU, relative fluorescence units.

2) Tested the effect of modulating RAB27. RAB27 GTPases are well-established regulators of multivesicular endosome docking and exosome release, and perturbing RAB27A is a standard strategy to attenuate EV pathway activity and test EV dependence of phenotypes in cancer models (Ostrowski *et al*, 2010; Wang *et al*, 2025; Jamshidiha *et al*, 2022). To test whether the observed pro-migratory effects of CIN^{HIGH} EVs indeed require EV-dependent trafficking in recipient cells, we transiently silenced RAB27A in BT549 recipient cells (Sup. Fig. 2I, J). While EVs isolated from REV-treated BT549 cells still increased migration of DMSO- or control-siRNA-treated recipient BT549 cells compared to treatment with EVs from mock-treated BT549 cells, recipient BT549 cells in which RAB27A was knocked down showed significantly decreased migration and a much smaller increase in migration when treated with EVs isolated from REV-treated BT549 cells (Sup. Fig. 2K-L). These findings further underscore the importance of efficient EV uptake by the recipient cells, particularly for EVs secreted by REV-treated cells and confirm the paracrine nature of our phenotype.

Supplementary Figure 2. CIN drives pro-migratory EVs and requires RAB27A in recipient cells. (I) Western blot quantification of RAB27A knockdown efficacy in BT549 cells. (J) qPCR quantification of RAB27A knockdown efficacy in BT549 cells: Statistical tests were done using two-way ANOVA (n=3; ***, **p < 0.01, ***p < 0.001, ****p < 0.0001). (K, L) Representative images (K) and quantification (L) of siRNA-mediated RAB27A knockdown in BT549 recipient cells reduces the increases in migration induced by reversine-derived (CIN^{HIGH}) EVs. Two-way ANOVA (n = 3), *p < 0.05, **p < 0.01, ***p < 0.001, ****p < 0.0001. Scale bar: 100 μ m.

In conclusion, this manuscript presents a significant and well-supported investigation into CIN-driven metastasis mechanisms in TNBC, focusing on EFEMP1 as a promising therapeutic target. Implementing the above suggestions will enhance the mechanistic understanding and translational potential of these findings, thereby strengthening the manuscript's impact.

Many thanks for your kind and constructive assessment!

Referee #2:

The manuscript by Zheng et al describes a series of experiments, which suggest that induction of chromosomal instability in triple negative breast cancer (TNBC) cell lines leads to an increase in expression of the extracellular matrix protein EFEMP1. Through knockdown and overexpression experiments, the authors suggest that EFEMP1 promotes cell migration and invasion, and that it can be secreted to stimulate these effects in cell culture potentially in association with extracellular vesicles (EVs). This work is backed up by xenograft experiments in zebrafish and patient studies that suggest high EFEMP1 expression correlates with poorer outcome in grade 1 breast cancer.

As the authors explain, the link between EFEMP1 and cancer metastasis has already been made for TNBC, but the link to chromosomal instability seems to be new, as does the important role of STAT1 in controlling EFEMP1 expression. The association with EVs is perhaps the most significant aspect, but I think this is also the least robust part of the study based on current data.

Many thanks for your constructive assessment.

I have the following concerns:

Major

2.1 Are the paracrine effects of EFEMP1 mediated by EVs? The authors use a standard ultracentrifugation preparation method (combined with some concentration by filtration) to isolate small EVs (sEVs). As best shown in the study by Jeppesen *et al.*, 2019 (Reassessment of Exosome Composition. *Cell*. doi: 10.1016/j.cell.2019.02.029), this approach also co-isolates non-vesicular protein aggregates and these can contain high levels of extracellular proteins. These aggregates might also be mediating the effects described and without assessing this, the core conclusions regarding a role for EVs cannot be made.

The authors should do the following experiments to clarify this:

- Density gradient (iodixanol) separation of EVs from aggregates and western blots of fractions to identify which particles contain EFEMP1.
- The vesicle fractions could then be tested for activity versus protein aggregates.
- Ideally, it would also be informative try knockdown experiments in secreting cells, which reduce EV secretion. One problem with this approach is that treatments are different for blocking exosomes (Rab27a kd²) versus small ectosomes (ARRDC1 inhibition) but showing that either of these suppressed the effects of paracrine EFEMP1 would strengthen the conclusion that EVs are involved.

Many thanks for proposing these important control experiments to show that our effects are indeed EV-mediated. To address points a and b, we have:

- as a complementary method to ultracentrifugation, purified EVs using Size Exclusion Chromatography (SEC) another established assay in the field to purify EVs without aggregates (Tkach *et al*, 2022). For this, EVs from BT549 cells were concentrated by ultracentrifugation and then further purified by SEC using qEV columns. Ponceau S staining of ultracentrifuge inputs and SEC outputs revealed low levels of bulk protein across fractions and controls (Sup. Fig. 1B), consistent with an efficient separation of vesicular and soluble components rather than bulk co-isolation of extracellular protein aggregates. We next profiled all SEC fractions by Western blot for the canonical EV markers CD63 and CD81. Both EV markers were strongly enriched in fractions 1–5, whereas fractions 6–10 were essentially devoid of EV markers (Fig. 1G).

Supplementary Figure 1: (B) Ponceau S staining of ultracentrifuge inputs and SEC fractions.

Figure 1: (G) Western blots of SEC fractions show EV markers (CD63, CD81) enriched in fractions 1–5 and absent in fractions 6–10.

- isolated EVs using both methods from STAT1^{KO} cells that overexpress EFEMP1 to test whether the functional impact of the EVs was affected by the concentration method. We then either took the ultracentrifuge-purified EVs (method 1) or the first five fractions from the SEC purification that contain EVs and EFEMP1 (method 2; Sup. Fig. 7F) and incubated these EVs with BT549 recipient cells. Using either purification method, EVs isolated from EFEMP1^{OEX} cells increased migration, and to a similar extent (Sup. Fig. 7G, H), indicating that the migration effects we observe are indeed driven by EVs.

Supplementary Figure 7 EFEMP1 and EV marker expression in BT549 WT and BT549 STAT1^{KO} cells. (F) Western blots of EV-protein isolates from STAT^{KO} BT549 cells treated with or without EFEMP1 overexpression fractionated by SEC with fractions 1–5 and fractions 6–10 pooled. (G, H) Representative images (G) and quantification (H) of cell migration and invasion of BT549 cells co-cultured with EVs isolated from STAT1^{KO} BT549 cells by ultracentrifugation (UC) or size-exclusion chromatography (qEV), treated with vector or EFEMP1^{OEX}, assessed by transwell assays. Statistical

significance was determined by Two-way ANOVA (n= 3 biological replicates), *p < 0.05, **p < 0.01, ***p < 0.001, ****p < 0.0001. Scale bar: 100 µm.

To address point c and confirm that vesicle uptake mechanisms are required for the EV phenotype to manifest itself, we tested the effect of modulating RAB27 as suggested by this reviewer. RAB27 GTPases are well-established regulators of multivesicular endosome docking and exosome release, and perturbing RAB27A is a standard strategy to attenuate EV pathway activity and test EV dependence of phenotypes in cancer models (Ostrowski *et al*, 2010; Wang *et al*, 2025; Jamshidiha *et al*, 2022). To test whether the observed pro-migratory effects of CIN^{HIGH} EVs indeed require EV-dependent trafficking in recipient cells, we transiently silenced RAB27A in BT549 recipient cells (Sup. Fig. 2I, J). We then isolated EVs using SEC purification. While EVs isolated from REV-treated BT549 cells still increased migration of DMSO- or control siRNA-treated recipient BT549 cells compared to treatment with EVs from mock-treated BT549 cells, recipient BT549 cells in which RAB27A was knocked down showed significantly decreased migration and a much smaller increase in migration when treated with EVs isolated from REV-treated BT549 cells (Sup. Fig 2K, L). These findings further underscore the importance of efficient EV uptake by the recipient cells, particularly for EVs secreted by REV-treated cells and confirm the paracrine nature of our phenotype.

Supplementary Figure 2. CIN drives pro-migratory EVs and requires RAB27A in recipient cells. (I) Western blot quantification of RAB27A knockdown efficacy in BT549 cells. (J) qPCR quantification of RAB27A knockdown efficacy in BT549 cells: Statistical tests were done using two-way ANOVA (n=3; ***, **p < 0.01, ***p < 0.001, ****p < 0.0001). (K, L) Representative images (K) and quantification (L) of siRNA-mediated RAB27A knockdown in BT549 recipient cells reduces the increases in migration induced by reversine-derived (CIN^{HIGH}) EVs. Two-way ANOVA (n = 3), *p < 0.05, **p < 0.01, ***p < 0.001, ****p < 0.0001. Scale bar: 100 µm.

2.2 Several experiments in which the effects of different EV preparations are tested do not have a no-EV control, eg. Figs. 2C/D, 3C/E/G, S5E/F and 5D/E. It is therefore unclear whether EV preps

under any condition promote functions like invasion and migration, or whether control EVs preps have no effect or even negative effects. This seems an important point to clarify.

Indeed, this is a very important control that we missed. In the revised manuscript, we include no-EV controls for key migration and invasion assays, and for the EV uptake and proliferation assay. More specifically, we now added PBS controls (*i.e.* no-EV controls) to Fig. 2A,2B,2D,2E, 2F, 2G, Sup2A,2B,2K, 2L. As we observed no differences between PBS-treated cells or cells treated with EVs isolated from DMSO-treated cells, we conclude that the control EVs have a negligible effect on the migration effect of recipient cells. Therefore, the increased migration instigated by the EVs isolated from CIN^{HIGH} cells seems to primarily be driven by the increase of STAT1-induced EFEMP1 in EVs.

Figure 2 Paracrine effects of EVs isolated from CIN⁺ BT549 and MDA-MB-231 cells on cell proliferation, migration, and invasion. (A) Representative images of EdU incorporation in BT549 cells treated with EVs from DMSO or REV treated cells. Scale bar: 100 μ m. **(B)** Quantification of EdU incorporation as shown in (A) for three biological replicates. Statistical analysis was performed using Two-way ANOVAS (* $p < 0.05$; ** $p < 0.01$; *** $p < 0.001$; **** $p < 0.0001$). **(D, E)** Representative images (D) and quantification (E) of cell migration and invasion of BT549 cells treated with PBS, DMSO (CIN^{LOW}) or REV (CIN^{HIGH}) using trans-well assays. Scale bar: 100 μ m. Statistical significance was determined using one-way ANOVA (* $p < 0.05$, ** $p < 0.01$, *** $p < 0.001$, **** $p < 0.0001$). Experiments were performed as biological triplicates. **(F, G)** Representative images (F) and quantification (G) of cell migration and invasion of MDA-MB-231 cells treated with PBS, DMSO (CIN^{LOW}) or REV (CIN^{HIGH}) using trans-well assays. Scale bar: 100 μ m. Statistical significance was determined using one-way ANOVA (* $p < 0.05$, ** $p < 0.01$, *** $p < 0.001$, **** $p < 0.0001$). Experiments were performed as biological triplicates.

Supplementary Figure 2. CIN drives pro-migratory EVs and requires RAB27A in recipient cells (A, B) Representative images of EdU incorporation in MDA-MB-231 cells treated with EVs from PBS, DMSO or REV treated cells (A) and quantification for three biological replicates (B). Statistical analysis was performed using One-way ANOVA. **(K, L)** Representative images (K) and quantification (L) of siRNA-mediated RAB27A knockdown in BT549 recipient cells reduces the increases in migration induced by reversine-derived (CIN^{HIGH}) EVs. Two-way ANOVA (n = 3), *p < 0.05, **p < 0.01, ***p < 0.001, ****p < 0.0001. Scale bar: 100 μ m.

Minor

2.3 Page 3, line 65 - 'the role of EVs in a chromosomal instability background remains unexplored' - some of the authors have reviewed this topic (Zheng *et al*, 2023, doi: 10.3390/cells12232712) and should include some of the studies they mention in this review, because they suggest a link between EVs and CIN.

Many thanks for bringing this up. We have now added some more of the primary references as well, *i.e.* (Zheng *et al*, 2023; Adams *et al*, 2021; Bao *et al*, 2021; Martins *et al*, 2023; Fordjour *et al*, 2019), also see page 3, line 68, 69.

2.4 Some EV isolations have contaminants from proteins that should not be present, eg. Fig. 1F - calnexin; S4D - actin; S4F - calnexin and actin; S4G - calnexin. A key reason for doing these experiments is to eliminate contaminating material produced by cell lysis, and this suggests the preparations have additional contaminants, as well as secreted aggregates.

Many thanks for bringing up this point. We improved our EV isolation protocols and

performed new Western blot analyses that confirm this improved isolation. The corresponding blots are shown in Figs. 1F, 1G and Sup. Figs. 3G, 5A, 5B, 5D, 5E, 5G, 7D, 7E, and 7F.

2.5 Statistics - at the very least, multiple comparisons should use an ANOVA test, assuming parametric data, not a t-test, eg. Fig. 3G and 4E, and this will apply to other data where a no-EV control is included.

Thank you for this. We performed ANOVAs for all multiple comparisons. Please find the results in Figs. 2B, 2E, 2G, 3F, 4C, 4E, Sup. Figs. 2L, 2J, and 7H with all relevant controls in Figs. 2A, 2B, 2D, 2E, 2F, 2G, Sup. Figs. 2A, 2B, 4K, and 4L.

Figure 2 Paracrine effects of EVs isolated from CIN+ BT549 and MDA-MB-231 cells on cell proliferation, migration, and invasion. **(A)** Representative images of EdU incorporation in BT549 cells treated with EVs from DMSO or REV treated cells. Scale bar: 100 μm. **(B)** Quantification of EdU incorporation as shown in (A) for three biological replicates. Statistical analysis was performed using Two-way ANOVAS (*p < 0.05; **p < 0.01; ***p < 0.001; ****p < 0.0001). **(D, E)** Representative images (D) and quantification (E) of cell migration and invasion of BT549 cells treated with PBS, DMSO (CIN^{LOW}) or REV (CIN^{HIGH}) using trans-well assays. Scale bar: 100 μm. Statistical significance was determined using one-way ANOVA (*p < 0.05, **p < 0.01, ***p < 0.001, ****p < 0.0001). Experiments were performed as biological triplicates. **(F, G)** Representative images (F)

and quantification (G) of cell migration and invasion of MDA-MB-231 cells treated with PBS, DMSO (CIN^{LOW}) or REV (CIN^{HIGH}) using trans-well assays. Scale bar: 100 μ m. Statistical significance was determined using one-way ANOVA (* p < 0.05, ** p < 0.01, *** p < 0.001, **** p < 0.0001). Experiments were performed as biological triplicates.

Supplementary Figure 2. CIN drives pro-migratory EVs and requires RAB27A in recipient cells (A, B) Representative images of EdU incorporation in MDA-MB-231 cells treated with EVs from PBS, DMSO or REV treated cells (A) and quantification for three biological replicates (B). Statistical analysis was performed using One-way ANOVA. **(K, L)** Representative images (K) and quantification (L) of siRNA-mediated RAB27A knockdown in BT549 recipient cells reduces the increases in migration induced by reversine-derived (CIN^{HIGH}) EVs. Two-way ANOVA ($n = 3$), * p < 0.05, ** p < 0.01, *** p < 0.001, **** p < 0.0001. Scale bar: 100 μ m.

2.6 In Fig. 4C, shouldn't the blue and purple bars also be compared to determine whether STAT1 controls some of the other genes?

Absolutely. In the revised Fig. 4C, we added these comparisons between WT-REV and STAT1^{KO} REV to determine which other gene products are upregulated in a STAT1-dependent manner under induction of CIN.

Figure 4 STAT1 is required for CIN-induced EFEMP1 expression and its downstream effects on migration and invasion. (C) qPCR quantification of IL6, IL8, CXCL1, CXCL10, EFEMP1, STAT1, STAT3, CD63 and CYLD in BT549 WT and STAT1 KO cells. Statistical analysis was done using two-way ANOVA (N = 3; *p < 0.05, ** p < 0.01, ***p < 0.001, ****p < 0.0001).

2.7 For Fig. S4G and S6B, the EFEMP1 westerns appear overexposed or too much protein has been loaded for the overexpressing EFEMP1 cells.

Thank you for bringing this up. The new Western blots can be found in Sup. Figs. 5B and 5E. Please also see point 2.4.

2.8 Have the proteomics data been deposited in a public repository? Since EFEMP1 is an ECM protein that has been shown to influence other ECM protein organisation, I wondered whether there were any of these proteins in the proteomics list, which could have been tested by western in cells/EVs where EFEMP1 is knocked down or overexpressed to determine whether these are co-regulated on EVs or extracellular non-vesicular particles. This is relevant in the context that other ECM proteins have already been suggested to regulate migration on EVs, eg fibronectin, Sung BH, et al. 2015, (Directional cell movement through tissues is controlled by exosome secretion. Nat Commun. 2015 doi: 10.1038/ncomms8164), a study that might be mentioned in the introduction.

Thank you for this suggestion. Our mass spec data has been publicly made available via the PRIDE repository under accession PXD073786. We refer to this study in our discussion on Page 27, line 574.

Text

2.9 Page 3, line 58 - 'microvesicles' are now generally called 'ectosomes' in the literature

Thanks, we corrected this. Also see page 3, line 59.

2.10 Page 4, line 100 - use 'isolated' or 'separated and concentrated', not 'purified', partly for the reasons discussed above

We corrected this, also see see page 4, line 107 and throughout the text at other places where we originally used purified.

2.11 Line 103-4 - 'confirmed that the isolates contained EVs' - there are no assays presented to look for common contaminant secreted protein aggregates.

We blotted all fractions from the SEC concentration protocol (Fig. 1G, Sup. Fig. 1B, and 7F) and found no evidence for contaminants.

Sup. Fig. (1B) Ponceau S staining of ultracentrifuge inputs and SEC fractions.

(1G) Western blots of SEC fractions show EV markers (CD63, CD81) enriched in fractions 1–5 and absent in fractions 6–10.

Sup. Fig. (7F) Western blots of EV-protein isolates from STAT^{KO} BT549 cells treated with or without EFEMP1 overexpression fractionated by SEC with fractions 1–5 and fractions 6-10 pooled

Referee #3:

In this manuscript, Zheng et al investigate the link between CIN and invasion in TNBC. In particular, the authors examined the paracrine effect of CIN cancer cells in the migration/invasion of the same cells. For that, they went on to test how CIN impacts EVs secretion, since EVs play roles in cell-cell communication in cancer. They found that inducing CIN by treating cancer cells with reversine promotes the secretion of EVs and that these EVs can promote migration and invasion of cancer cells. Proteomics analyses of these EVs suggest that these phenotypes could be induced by the extracellular matrix protein EFEMP1. Depletion of EFEMP1 from breast cancer cell lines diminishes significantly the ability of EVs to promote migration/invasion. The authors propose that STAT1 expression is linked to EFEMP1 expression.

Overall, the work presented here is novel and experiments well conducted. However, below concerns are raised about some of the conclusions and lack of quantifications necessary to support some of the claims. The main concern is the induction of CIN phenotype and how specific what is described here is. The authors should address these points before publication.

Many thanks for this kind summary and underscoring the relevance of our work.

Major comments:

3.1 The choice of cell lines was unclear, apart from the fact that these are TNBC. I assume that both cell lines already display CIN? Thus, it was unclear why the need of further inducing CIN and not choosing cells lines that have no CIN to compare? And induce CIN on those?

Indeed, we chose to study the effect of increased CIN rates in TNBC cells as we previously identified strong inflammation phenotypes in TNBC cells with induced CIN (Hong *et al*, 2022). As cell-intrinsic inflammation might instigate paracrine effects as well, in this study we tested whether induced CIN promotes the secretion of EVs and if so, what the functional of these EVs are for neighbouring cells. We now better explain this in the introduction. While both cell lines display a modest intrinsic CIN phenotype (also see Fig. 1A, the cell-intrinsic CIN rate is ~20%), reversine (REV) treatment increases the CIN rate to 100%, an effect size required to measure the effects of CIN effectively in tissue culture models. Therefore, to

answer your question: for these experiments, we need a population of cells in which all cells display CIN. Therefore, as a model system for CIN, we used drug-mediated CIN in cell lines that have low intrinsic CIN rates so that we can compare CIN^{LOW} to CIN^{HIGH} with substantial differences in CIN rates between low and high CIN conditions.

However, your point is well-taken and we therefore also induced CIN in two non-transformed (near)-diploid cell lines and measured whether EVs isolated from these cells would also induce increased migration of BT549 TNBC recipient cells. We find that EVs isolated from non-transformed cell lines with induced CIN fail to promote migration of recipient TNBC cells (Sup. Fig. 3C, D, left bars). This implies that the migration phenotype instigated by EVs isolated from CIN^{HIGH} TNBC cells is not universal and that in addition to CIN some other rewiring needs to take place in cells to start secreting EFEMP1-loaded EVs that promote migration in a paracrine manner. We also bring this forward in the revised discussion section.

Fig. 1 (A) Quantitative analysis of mitotic aberrations in BT549 and MDA-MB-231 cells after exposure to 250 nM and 500 nM reversine (REV).

Sup. Fig 3 (B) Quantification of scratch-wound migration in RPE1 fibroblasts treated with BT549-derived EVs generated under 250 nM reversine, and the reciprocal condition in which BT549 cells were treated with RPE1-derived EVs at the same doses. Statistical significance was assessed by two-sided t-tests (n = 3 biological replicates); p < 0.01.

Sup. Fig. 3 (C) Quantification of scratch-wound migration in BJ fibroblasts treated with BT549-derived EVs generated under 250 nM reversine, and the reciprocal condition in which BT549 cells were treated with BJ-derived EVs at the same doses. Statistical significance was assessed by two-sided t-tests (n = 3 biological replicates); p < 0.01.

3.2 I could not find in the manuscript how the reversine treatments were performed apart from concentration). This information needs to be added, including on the main text as it is critical for the interpretation of the data. The main reason I am bringing this as a major comment is related with the above comment and the graphs in Fig 1A where a variety of phenotypes, including some that will promote tetraploidy and cell arrest. How sure are the authors that what they identified here is a consequence of CIN and not other more significant phenotypes that culminate with severe cell arrest, mitotic failure, senescence, apoptosis and DNA damage?

We now better describe our approach in the results section: *“To induce CIN^{HIGH} phenotypes, we use the MPS1 inhibitor reversine (REV) (Santaguida et al, 2010), a widely used compound for this purpose (Bosco et al, 2018; Hiruma et al, 2016; Garribba et al, 2023; Ippolito et al, 2021). CIN phenotypes were quantified by time-lapse imaging according to established protocols (Crozier et al, 2022; Thu et al, 2018; Huis In ’t Veld et al, 2019) and confirmed that MPS1 inhibition increased the rate of mitotic abnormalities in a dose-dependent manner in both BT549 and MDA-MB-231 TNBC*

cells (Fig. 1A). Cells were treated with 500 nM of REV for 72 h prior to EV isolation or harvesting of target cells, unless indicated otherwise. Inhibition of Mps1 is a very common strategy to induce CIN in cell lines, and indeed Mps1 inhibitor treatment can lead to a variety of abnormalities during mitosis ranging from individual lagging chromosomes to tetraploidization. All of the abnormalities quantified in Fig. 1A are indeed different manifestations of CIN and while the types of CIN are slightly different between cell lines, the EVs from both TNBC cells with induced CIN yield similar migration effects on recipient cells. We now also show that that REV treatment decreases EdU incorporation (as a proxy of proliferation) in both BT549 cells and MDA-MB-231 cells (Sup Fig 2C-F). However, we fully agree that as a result of CIN part of the cells will display cell cycle arrest or cell death and that we cannot exclude that other instigators of cell cycle arrest do the same. This was also not the purpose of our study. However, we agree that this is an important limitation, which we now do emphasize in the first paragraph of the discussion: *“However, CIN triggers various downstream responses including cell cycle arrest, cell death, and inflammation (Zheng et al, 2023). While our data indicate that ongoing CIN leads to increased secretion of EVs and that these EVs modulate the TME, but that this CIN also leads to decreased proliferation of the EV-secreting cells, future work should reveal whether other instigators of cell cycle arrest also yield secretion of EVs that promote migration of recipient cells.”*

Supplementary Figure 2. CIN drives pro-migratory EVs and requires RAB27A in recipient cells. (C, D) Representative immunofluorescence images of EdU incorporation in BT549 EV-donor cells treated with DMSO or REV prior to EV isolation (C), quantified for three biological replicates (D). Statistical analysis was performed using a two-sided t-test. (E, F) Representative immunofluorescence images of EdU incorporation in MDA-MB-231 EV-donor cells treated with DMSO or REV prior to EV isolation (E), quantified for three biological replicates (F).

3.3 Related to the comment above, the morphology of cells after reversine changes significantly, as seen for example in the migration and invasion assays in Fig 2C-D. are some of these cells on their way to become senescence? And hence the changes in the secretory phenotype the authors observed?

Thanks for bringing this up. We agree that some cells now shown in Fig. 2D and 2F look enlarged. However, the images of the migration and invasion assays that the reviewer refers to are in fact CIN^{LOW} cells that are exposed to EVs secreted by CIN^{HIGH} cells. Therefore, morphology changes are the result of EV exposure and not CIN. While we observed changes in migration behaviour of these EV-treated cells, we did not find any changes in their proliferation rate (also see Sup. Fig. 2A and 2B). Therefore, in our view, it is unlikely that these cells are on their way to senescence, also as the cells with similar morphologies can be seen in the control conditions, even in cells that are treated with PBS instead of EVs (new controls that we added, also see point 2, reviewer 2).

Supplementary Figure 2. CIN drives pro-migratory EVs and requires RAB27A in recipient cells (A, B) Representative images of EdU incorporation in MDA-MB-231 cells treated with EVs from PBS, DMSO or REV treated cells (A) and quantification for three biological replicates (B). Statistical analysis was performed using One-way ANOVA.

3.4 Uptake of vesicles by the cancer cells needs to be quantified by different methods. The movie shows vesicles on cells but there is no indication that these vesicles enter the cells. Could the authors quantify the % of cells that uptake vesicles and quantify vesicle uptake for example using confocal microscopy and assessing vesicles inside cells? (z-stacks)

Many thanks for this suggestion. This is an important control that we now added. To address this issue, we have:

1) Labelled EVs using PKH26 staining, which confirms that EVs are taken up by recipient cells at 37°C much more efficiently compared to when cells are at 4°C which is known to block EVs uptake. Furthermore, this labelling confirms increased uptake of EVs when EVs are isolated from CIN^{HIGH} cells, in line with their increased secretion of EVs. Data for these

experiments are shown in Sup. Figs. 1E, F.

Zheng *et al*, Sup. Figure 1

Supplementary Figure 1. Biophysical characterization of EVs. (E) Immunofluorescence images of BT549 cells labelled for PKH26 and F-actin incubated for 4 h with indicated EV preparations, compared to DMSO controls. Nuclei were counterstained with DAPI. (F) Quantification of PKH26 fluorescence intensity (RFU) in BT549 cells in the presence or absence of 500 nM REV-induced EVs. * $p < 0.05$. RFU, relative fluorescence units.

2) Tested the effect of modulating RAB27. RAB27 GTPases are well-established regulators of multivesicular endosome docking and exosome release, and perturbing RAB27A is a standard strategy to attenuate EV pathway activity and test EV dependence of phenotypes in cancer models (Ostrowski *et al*, 2010; Wang *et al*, 2025; Jamshidiha *et al*, 2022). To test whether the observed pro-migratory effects of CIN^{HIGH} EVs indeed require EV-dependent trafficking in recipient cells, we transiently silenced RAB27A in BT549 recipient cells (Sup. Fig. 2E, F). While EVs isolated from REV-treated BT549 cells still increased migration of DMSO- or control siRNA-treated recipient BT549 cells compared to treatment with EVs from mock-treated BT549 cells, recipient BT549 cells in which RAB27A was knocked down showed significantly decreased migration and a much smaller increase in migration when treated with EVs isolated from REV-treated BT549 cells (Sup. Fig 2G-I). These findings further underscore the importance of efficient EV uptake by the recipient cells, particularly for EVs secreted by REV-treated cells and confirm the paracrine nature of our phenotype.

Supplementary Figure 2. CIN drives pro-migratory EVs and requires RAB27A in recipient cells. (I) Western blot quantification of RAB27A knockdown efficacy in BT549 cells. (J) qPCR quantification of RAB27A knockdown efficacy in BT549 cells: Statistical tests were done using two-way ANOVA (n=3; ***, **p < 0.01, ***p < 0.001, ****p < 0.0001). (K, L) Representative images (K) and quantification (L) of siRNA-mediated RAB27A knockdown in BT549 recipient cells reduces the increases in migration induced by reversine-derived (CIN^{HIGH}) EVs. Two-way ANOVA (n = 3), *p < 0.05, **p < 0.01, ***p < 0.001, ****p < 0.0001. Scale bar: 100 μ m.

3.5 The heatmap provided in Fig 2E represents all proteins from the mass spec data that are significantly changed between control and reversine treated EVs? How many of those have been shown to be associated with EVs?

Excellent points. The heatmap (now Fig. 2H) shows a selection of EV-enriched proteins. The full list can be found in Sup. Table 1. We now better indicate this in the main text and legend of Fig. 2H. To address the second point, we compared all proteins isolated from our EV purifications to proteins annotated in Exocarta (Keerthikumar *et al*, 2016). We observed a substantial overlap between the proteins we identified and Exocarta: 415 out of the 478 identified peptides are of proteins included in Exocarta (Sup. Fig. 3D), underscoring a high concentration of EVs in our EVs isolates.

2H

(2H) Heatmap of a selection of peptides identified in the EVs isolates from BT549 cells treated with REV compared to controls. The full list can be found in **Sup. Table 1**.

Sup 3D

(Sup 3D) Venn diagram of peptides enriched in EVs isolated from BT549 cells compared to Exocarta database.

3.6 Fibulin 3 (EFEMP1) was identified in the label-free proteomics samples as increased upon reversine treatment. However, as seen in on the WB in Suppl Fig 2F there is little to know difference observed between DMSO and reversine treated EVs. It would be great if the authors could show quantifications of the WB data that is key for their conclusions as not only that is not provided but significant variability is observed between WBs. The WB for CD63 and CD81 are not ideal and might be difficult to use as loading controls, which is necessary to quantify the mount of EFEMP1 associated with EVs.

This is a fair point, and we apologize for the poor quality of the blots in our first submission. We worked very hard to improve the quality of the Western blots, particularly those of EVs, which are challenging due to the low protein concentration in EV isolates. Taking advantage of our improved protocols, we repeated these experiments and blotted whole cell lysates and the EVs for CD63, CD81, EFEMP1, Calnexin and Vinculin. The blots now are much cleaner. We quantified EFEMP1 in these blots with Vinculin as a control for whole cell lysates and CD81 as a control for EVs, which shows that EFEMP protein levels are increased by ~ 40%. This is well in line with the increase of ~40% of EFEMP1 in CIN^{HIGH} EVs observed in the mass spectrometry results. While this increase might seem subtle, the maximum increase we could accomplish with overexpression of EFEM1 in whole cell lysates was ~70% (Sup. Fig 5B), and EVs isolated from these cells yielded comparable effects on migration as EVs isolated from CIN^{HIGH} cells. We therefore conclude that this 40-70% increase in the physiological range required for the observed migration phenotypes.

Sup 5B

Sup. Fig. 5 (D) CD63, CD81, EFEMP1, Calnexin and vinculin protein levels in whole cell lysates and EVs from BT549 scramble and EFEMP1KD cells determined by Western blot.

3. 7 The WB on suppl figure 4 are not conclusive. This is a key part of the work. The levels of EFEMP1 vary significantly and no quantifications are provided. As an example, on panel F, EFEMP1 is observed both on lysates and EVs from MDA-231 cells but that is not the same on panel G where EFEMP1 is barely detected. This variability makes it hard to conclude that the overexpression construct is working and increased EFEMP1 is observed in the lysates and extracts because on that WB it is barely seen in the control conditions (which is different from other WB). On panel D regarding overexpression of EFEMP1 in BT-549 cells, the authors also claim that EFEMP1 can be detected at higher levels in EVs, however CD63 levels are also increase and thus difficult to conclude if that is the case without quantifications.

This is in line with the previous point, and we fully agree better blots are required. Taking advantage of our improved protocols, we repeated all EV isolations and Western blots. The new results, including quantification for EFEMP1 are shown in Figs. 1F, 1G and Sup. Figs. 3G, 5A, 5B, 5D, 5E, 5G, 7D, 7E, 7F. Legends can be found in the revised manuscript.

3.8 The authors conclude that EFEMP1 associated with EVs is important for the paracrine EV-mediated migration/invasion. However, loss of EFEMP1 also affects the same phenotypes in DMSO treated cells. Could the authors comment on the specificity of the phenotype? The same is observed for STAT1 KD.

Thank you for bringing up this point. Our working hypothesis is that CIN leads to increased expression of EFEMP1, which is then secreted via EVs to induce migration of recipient cells. However, we also find that DMSO-treated cells already express some EFEMP1. Therefore, EVs isolated from DMSO-treated cells will also promote migration, but much less than EVs isolated from cells with a CIN phenotype. The same is true for STAT1^{KO} cells: as EFEMP1 is a STAT1 target, its expression is reduced in STAT1^{KO} cells. As a result, EVs isolated from STAT1^{KO} cells promote migration less than EVs isolated from STAT1^{WT} cells. We now better explained this throughout the manuscript.

3.9 I find the correlation plots on Fig 4A-B and Fig 6A-B very difficult to interpret as the correlations are not obvious and how significant these data is unclear. For example, on Fig 4A, when STAT1 is expressed from log₂ 4 to 8 where most data points are there is no correlation that is obvious. The interpretations should be toned down.

Apologies that this was confusing. Fig. 4 A, B show the correlation between EFEMP1 and STAT1 mRNA expression across ~1,000 cancer cell lines in DepMap. We find that the correlation coefficient across all cancer cell lines, and across breast cancer cell lines is 0.3 and 0.25, respectively. Correlation coefficients R in the range of 0.2 - 0.4 indicate a weak positive correlation. Similarly, Sup. Fig. 8 D, E (Sup. Fig. 6 in original submission) now show the correlation between the aneuploidy score and EFEMP1 expression in TCGA included (breast) cancers with R values 0.2 (all cancers) and 0.35 (breast cancer), also indicating a weak positive correlation. We toned down our phrasing by stating that these correlations are 'weakly positive' instead of just 'positive'.

Supplementary Figure 8. High resolution imaging confirms MDA-MB-231 spreading to the tail of xenografted zebrafish embryos, and EFEMP1 and STAT1 expression patterns in breast cancer progression and prognosis. (D-E) Scatter plots showing the correlation between EFEMP1 expression and aneuploidy score in DepMap included cancer cell lines (D) or DepMap included breast cancer cell lines (E).

3.10 Data on patient survival does not support a strong association of EFEMP1 and STAT1 with progression and it is unclear whether it could be used as prognostic as differences are very minor. In my opinion, this is not needed to validate the findings in patients.

Thank you for this comment. We do not show a correlation between STAT1 expression and patient survival, only between EFEMP1 expression and survival. While EFEMP1 expression is associated with patient survival, we do acknowledge that the differences observed are small. We made this more explicit in the text as *‘Furthermore, stratifying breast cancer survival for EFEMP1 expression levels revealed that high EFEMP1 expression associated with a more adverse outcome, particularly in higher-grade tumours (Sup. Fig. 8F–I), although these effects are modest’,* and *‘However, together these data confirm that EFEMP1 expression is associated with aneuploidy across cancers and with modestly decreased survival in breast cancer, also in a human setting.’* We do still show these data as it does show that high EFEMP1 expression is associated with decreased survival in higher grade breast cancers, not to position high EFEMP1 expression as a *bona fide* prognostic marker, but rather as a potential future clinical target to inhibit metastasis.

Supplementary Figure 8. High resolution imaging confirms MDA-MB-231 spreading to the tail of xenografted zebrafish embryos, and EFEMP1 and STAT1 expression patterns in breast cancer progression and prognosis. (F) Kaplan-Meier survival curves for TCGA-included breast cancer patients, showing overall survival for grade 3 (blue) (n = 952), grade 2 (yellow) (n=771) and grade 1 (tangerine) (n=169) breast cancer. (G-I) Kaplan-Meier survival curves for TCGA-included breast cancer patients stratified for low or high EFEMP1 expression per grade (grade I, G; grade II, H, grade III, I). Significant differences between EFEMP1 expression groups were tested using a Log-rank Test.

Minor comments:

3.11 In the abstract the authors state that the TME is modulated by EVs. However, soluble secreted proteins have been shown to play key roles in this process (cytokines, chemokines for example). It is important that it is clear that EVs are just one type of modulators and the authors should write the abstract and the manuscript with this in mind. Also, this manuscript does not investigate TME, the initial part of the abstract is misleading by those reasons.

We apologize if our abstract promised more than we delivered. We removed any reference to the TME in the abstract and results section and only discuss this briefly in the introduction as background knowledge and the discussion section. With reference to our own finding, we only refer to the role of EVs and their role in paracrine signalling in the revised manuscript.

3.12 Graphs on Figure 1A are better placed in the supplementary data.

Thanks for this suggestion. As Fig. 1A in our view is the validation of our CIN models and thus central to how we induced CIN in our cell models throughout the paper, we have chosen to leave this panel in the main figure.

3.13 It is very difficult to extract the data from the proteomics analyses on Table 1. The table should provide a list of protein names, IDs, all separated in different columns so that anyone can access the data. The authors should also compare their data sets with publicly available datasets for EVs proteomics. It is important to know how much overlap there is to understand how well represented this data set is.

This is a good point. We made Sup. Table 1 more intuitive and uploaded the raw mass spec data to the public repository PRIDE (project number: PXD073786). As explained under 3.5, we also mapped overlap of the proteins we identified with Exopedia and find that most proteins that we identified in our EV preparations are indeed annotated in Exopedia (Sup. Fig. 3C).

Sup 3D

Sup. Fig. 3 (D) Venn diagram of peptides enriched in EVs isolated from BT549 cells compared to Exocarta database.

3.14 First long paragraph on page 10 needs clarity as it is difficult to read and understand.

Many thanks for noting this. We rewrote and restructured this section to improve clarity.

3.15 I could not find information about how the treatments with EV were mediated. This needs to be explained, were EVs normalised for number? How long were cells treated with EVs?

Apologies this was not clear. The treatment regimen is shown in Fig. 2C. EVs were normalized towards the same concentration of $\sim 10^8$ particles/ml and cells were treated with EVs for 72h.

3.16 Figure 5 is not easy to follow. At what time invasion is quantified? In all time points on the scheme in 5A? if so, could the authors then provide the data by time point?

We apologize that this was unclear. We quantified cell migration 24 hours following PVS injection and now clarified this in the schematic overview (Fig. 5A) and in the text describing this experiment.

3.17 How the authors envision that EFEMP1-loaded EVs could promote migration/invasion? Is this protein inside the EVs? Outside? How much of it needs to be incorporated in EVs to make an impact on cells? Some of this should be discussed as it is not immediately clear mechanistically how this operates.

We agree this was not clear in the original version. In the revised version, we have added several experiments that show that EFEMP1 promotes migration via EVs. While we do not show where EFEMP1 resides in/on EVs, we do not make any claims on this in our manuscript. With the much-improved Western blots (also see this reviewer points 3.6 and 3.7) we now show that an increase of $\sim 40-70\%$ of EFEMP1 protein in/on EVs is sufficient to promote migration in recipient cells. We also refer to these points in the second paragraph of the discussion: *'We find that CIN increases EFEMP1 protein levels up to ~ 1.5 fold and that this is sufficient to promote the migration of cancer cells in tissue culture. While our findings show that EFEMP1 is associated*

with EVs, further work should clarify whether EFEMP1 resides inside or on the surface of the CIN-induced EVs. However, our findings do reveal EFEMP1 as a CIN-induced factor and a paracrine modulator of cell migration and invasion.’

References in rebuttal letter

- Adams SD, Csere J, D’angelo G, Carter EP, Romao M, Arnandis T, Dodel M, Kocher HM, Grose R, Raposo G, *et al* (2021) Centrosome amplification mediates small extracellular vesicle secretion via lysosome disruption. *Curr Biol* 31: 1403-1416.e7
- Bao S, Hu T, Liu J, Su J, Sun J, Ming Y, Li J, Wu N, Chen H & Zhou M (2021) Genomic instability-derived plasma extracellular vesicle-microRNA signature as a minimally invasive predictor of risk and unfavorable prognosis in breast cancer. *J Nanobiotechnology* 19: 22
- Bosco B, Defant A, Messina A, Incitti T, Sighel D, Bozza A, Ciribilli Y, Inga A, Casarosa S & Mancini I (2018) Synthesis of 2,6-Diamino-Substituted Purine Derivatives and Evaluation of Cell Cycle Arrest in Breast and Colorectal Cancer Cells. *Molecules* 23
- Crozier L, Foy R, Mouery BL, Whitaker RH, Corno A, Spanos C, Ly T, Gowen Cook J & Saurin AT (2022) CDK4/6 inhibitors induce replication stress to cause long-term cell cycle withdrawal. *EMBO J* 41: e108599
- Fordjour FK, Daaboul GG & Gould SJ (2019) A shared pathway of exosome biogenesis operates at plasma and endosome membranes. *bioRxiv*
- Garribba L, De Feudis G, Martis V, Galli M, Dumont M, Eliezer Y, Wardenaar R, Ippolito MR, Iyer DR, Tijhuis AE, *et al* (2023) Short-term molecular consequences of chromosome mis-segregation for genome stability. *Nat Commun* 14: 1353
- Halas-Wiśniewska M, Zawadka P, Arendt W & Izdebska M (2025) From Adhesion to Invasion: Integrins, Focal Adhesion Signaling, and Actin Binding Proteins in Cervical Cancer Progression—A Scoping Review. *Cells* 14
- Hiruma Y, Koch A, Dharadhar S, Joosten RP & Perrakis A (2016) Structural basis of reversine selectivity in inhibiting Mps1 more potently than aurora B kinase. *Proteins* 84: 1761–1766
- Hong C, Schubert M, Tijhuis AE, Requesens M, Roorda M, van den Brink A, Ruiz LA, Bakker PL, van der Sluis T, Pieters W, *et al* (2022) cGAS-STING drives the IL-6-dependent survival of chromosomally unstable cancers. *Nature* 607
- Huis In ’t Veld PJ, Volkov VA, Stender ID, Musacchio A & Dogterom M (2019) Molecular determinants of the Ska-Ndc80 interaction and their influence on microtubule tracking and force-coupling. *Elife* 8
- Ippolito MR, Martis V, Martin S, Tijhuis AE, Hong C, Wardenaar R, Dumont M, Zerbib J, Spierings DCJ, Fachinetti D, *et al* (2021) Gene copy-number changes and chromosomal instability induced by aneuploidy confer resistance to chemotherapy. *Dev Cell* 56: 2440-2454.e6
- Jamshidiha M, Lanyon-Hogg T, Sutherland CL, Craven GB, Tessa M, De Vita E, Brustur D, Pérez-Dorado I, Hassan S, Petracca R, *et al* (2022) Identification of the first structurally validated covalent ligands of the small GTPase RAB27A. *RSC Med Chem* 13: 150–155
- Keerthikumar S, Chisanga D, Ariyaratne D, Al Saffar H, Anand S, Zhao K, Samuel M, Pathan M, Jois M, Chilamkurti N, *et al* (2016) ExoCarta: A Web-Based Compendium of Exosomal Cargo. *J Mol Biol* 428: 688–692

- Martins ÁM, Lopes TM, Diniz F, Pires J, Osório H, Pinto F, Freitas D & Reis CA (2023) Differential Protein and Glycan Packaging into Extracellular Vesicles in Response to 3D Gastric Cancer Cellular Organization. *Adv Sci (Weinb)*: e2300588
- Neerincx A, Castro W, Guarda G & Kufer TA (2013) NLRC5, at the Heart of Antigen Presentation. *Front Immunol* Volume 4-2013
- Ostrowska-Podhorodecka Z, Ding I, Lee W, Tanic J, Abbasi S, Arora PD, Liu RS, Patteson AE, Janmey PA & McCulloch CA (2021) Vimentin tunes cell migration on collagen by controlling $\beta 1$ integrin activation and clustering. *J Cell Sci* 134: jcs254359
- Ostrowski M, Carmo NB, Krumeich S, Fanget I, Raposo G, Savina A, Moita CF, Schauer K, Hume AN, Freitas RP, *et al* (2010) Rab27a and Rab27b control different steps of the exosome secretion pathway. *Nat Cell Biol* 12: 19–30
- Robertson KA, Hsieh WY, Forster T, Blanc M, Lu H, Crick PJ, Yutuc E, Watterson S, Martin K, Griffiths SJ, *et al* (2016) An Interferon Regulated MicroRNA Provides Broad Cell-Intrinsic Antiviral Immunity through Multihit Host-Directed Targeting of the Sterol Pathway. *PLoS Biol* 14: e1002364
- Santaguida S, Tighe A, D’Alise AM, Taylor SS & Musacchio A (2010) Dissecting the role of MPS1 in chromosome biorientation and the spindle checkpoint through the small molecule inhibitor reversine. *J Cell Biol* 190: 73–87
- Satoh J-I & Tabunoki H (2013) A Comprehensive Profile of ChIP-Seq-Based STAT1 Target Genes Suggests the Complexity of STAT1-Mediated Gene Regulatory Mechanisms. *Gene Regul Syst Bio* 7: 41–56
- Thu KL, Silvester J, Elliott MJ, Ba-Alawi W, Duncan MH, Elia AC, Mer AS, Smirnov P, Safikhani Z, Haibe-Kains B, *et al* (2018) Disruption of the anaphase-promoting complex confers resistance to TTK inhibitors in triple-negative breast cancer. *Proc Natl Acad Sci U S A* 115: E1570–E1577
- Tkach M, Thalmensi J, Timperi E, Gueguen P, Névo N, Grisard E, Sirven P, Coccozza F, Gouronnet A, Martin-Jaular L, *et al* (2022) Extracellular vesicles from triple negative breast cancer promote pro-inflammatory macrophages associated with better clinical outcome. *Proceedings of the National Academy of Sciences* 119: e2107394119
- Wang L, Chen T, Bai H, Wei T & Zhang J (2025) Effect of the PTPN4/TRAM/TLR4 Signaling Pathway on Angiogenesis Mediated by Rab27a-regulated miR-17 Secretion in Breast Cancer Exosomes. *Am J Med Sci* 370: 291–304
- Zheng S, Guerrero-Haughton E & Fojier F (2023) Chromosomal Instability-Driven Cancer Progression: Interplay with the Tumour Microenvironment and Therapeutic Strategies. *Cells* 12

Dr. Floris Fojjer
University Medical Center Groningen
European Institute for the Biology of Aging
Antonius Deusinglaan 1
Groningen 9713AV
Netherlands

2nd Mar 2026

Re: EMBOJ-2024-119021R
Chromosomal instability promotes cell migration and invasion via EFEMP1 in extracellular vesicles

Dear Floris,

Thank you for submitting your revised manuscript to The EMBO Journal. It has now been re-reviewed by referee 3, and since the other two original referees were not available at this time, I have also carefully gone through all responses and revisions myself. I am pleased to say that from the scientific side, there appear to be no more concerns at this stage. Before proceeding further with the manuscript, we however need to clarify several issues that came up during our routine pre-acceptance image checks. From our analyses, it looks like full or partial duplicate images are displayed within Figure S4K, and between Figures S5B and S7E - despite being labelled as showing distinct conditions. We will require conclusive clarification of these issues, including provision of supporting source data, and careful revision of these figures as applicable.

In addition, there are also a number of other editorial points that would still need to be addressed at this stage:

- Some authors are listed on the manuscript with names differing from the author profiles in our system (Y. Liu, M. Borghesan) - please check and correct as appropriate. Please also provide a valid email address for K. Sjollema, as our acknowledgement could not be delivered to their currently entered address.
- Please carefully go through the reference list and make sure that each reference is complete with citation year, volume, and page/eLocator numbers - this information is currently missing for many of them.
- As we are switching from a free-text author contribution statement towards a more formal statement based on Contributor Role Taxonomy (CRediT) terms, please remove the present Author Contribution section and instead specify each author's contribution(s) directly in the Author Information page of our submission system during upload of the final manuscript. See <https://casrai.org/credit/> for more information
- The 'supplementary figures' S1-S8 should be turned into Expanded View Figures, and renamed 'Figure EV1-8' on all occasions (in-text call-outs, Expanded View Figure Legends, and within the respective figure files).
- The 'supplementary datasets' S1 and S2 should be turned into Expanded View Dataset EV1 and Expanded View Table EV1, respectively. Their legends should be moved from the main text and instead incorporated in a separate 'legends' tab of each spreadsheet file. Finally, again make sure to update all naming and referencing to 'Dataset EV1' and 'Table EV1', respectively.
- Please rename the supplementary movie as Expanded View movie (in-text callouts "Movie EV1"), and provide its legend in an individual text file, which should be combined with the respective movie file into a ZIP file and uploaded as such.
- Please ensure that externally deposited datasets become publicly accessible at this stage, and update the Data Availability statement by removing referee tokens and inclusions of final URLs.
- For the provided Figure Source Data, please make sure to upload them as one folder/archive file per MAIN figure, with all data contained in such an archive clearly labelled and attributed to particular panels. Source Data for Appendix or Expanded View Figures can be uploaded in a single archive file (combining respective subfolders).
- During routine pre-acceptance checks, our data editors have raised the following queries regarding figures, data, and legends; I would appreciate if you briefly answered to them in the cover letter of your final submission, and made the requested text modifications with changes/additions highlighted via the "Track changes" option, to facilitate our final checking"
 1. Please define the annotated p values ****/**/**/* as well as provide the exact p-values for the same in the legend of figure 1H as appropriate.
 - 2. Please note that the exact p values are not provided in the legends of figures 1D, E; 2E, G; 3B, D, F; 4A, C, E, G;
 3. Please indicate the statistical test used for data analysis in the legends of figures 1D, E, H; 4A, B"
 4. Please note that information related to n is missing in the legends of figures 1D, E, H

5. Please note that the error bars are not defined in the legends of figures 1D, E, H; 2E, G; 3B, D, F; 4C, E, G"

- Finally, please provide suggestions for a short 'blurb' text prefacing and summing up the conceptual aspect of the study in two sentences (max. 250 characters), followed by 3-5 one-sentence 'bullet points' with brief factual statements of key results of the paper; they will form the basis of an editor-written 'Synopsis' accompanying the online version of the article. Please also upload a synopsis image, which can be used as a "visual title" for the synopsis section of your paper. The image should ideally be in JPG format, and please make sure that it remains in the modest dimensions of (exactly) 550 pixels wide and 300-600 pixels high.

I am therefore returning the manuscript to you once more for revision, hoping that you will be able to satisfactorily address these remaining issues. Please do not hesitate to contact me should you have any questions in this regard.

With kind regards,

Hartmut

*** PLEASE NOTE: All revised manuscript are subject to initial checks for completeness and adherence to our formatting guidelines. Revisions may be returned to the authors and delayed in their editorial re-evaluation if they fail to comply to the following requirements. As a first step please read our guidelines for revised submissions:
<https://link.springer.com/journal/44318/submission-guidelines#cms-Revised-submissions>

1) Every manuscript requires a Data Availability section (even if only stating that no deposited datasets are included). Primary datasets or computer code produced in the current study have to be deposited in appropriate public repositories prior to resubmission, and reviewer access details provided in case that public access is not yet allowed.

4) Each main and each Expanded View (EV) figure should be uploaded as individual production-quality files (preferably in .eps, .tif, .jpg formats). For suggestions on figure preparation/layout, please refer to our Figure Preparation Guidelines:
<https://media.springernature.com/original/springer-cms/rest/v1/content/27825798/data/v1>

6) Please complete our Author Checklist, and make sure that information entered into the checklist is also reflected in the manuscript; the checklist will be available to readers as part of the Review Process File.

8) Please note that supplementary information at EMBO Press has been superseded by the 'Expanded View' for inclusion of additional figures, tables, movies or datasets; with up to five EV Figures being typeset and directly accessible in the HTML version of the article.

9) To facilitate reproducibility and cross-laboratory adoption of methodologies, please structure the Materials & Methods section as outlined in our guide to authors, including a completed Reagents and Tools Table.

10) Digital image enhancement is acceptable practice, as long as it accurately represents the original data and conforms to community standards. If a figure has been subjected to significant electronic manipulation, this must be clearly noted in the figure

legend and/or the 'Materials and Methods' section. The editors reserve the right to request original versions of figures and the original images that were used to assemble the figure. Finally, we generally encourage uploading of numerical as well as gel/blot image source data.

In the interest of ensuring the conceptual advance provided by the work, we recommend submitting a revision within 3 months (31st May 2026). Please discuss the revision progress ahead of this time with the editor if you require more time to complete the revisions. Use the link below to submit your revision:

Link Not Available

Referee #3:

The authors did a great job in addressing my concerns. The additional data and text changes really helps conveying the message. I have no further comments and consider this work ready for publication.